# *AR* alterations inform circulating tumor DNA detection in metastatic castration resistant prostate cancer patients

Todd P. Knutson [1,13], Bin Luo [2,13], Anna Kobilka[3], Jacqueline Lyman[3,4], Siyuan Guo[2], Sarah A. Munro[1], Yingming Li[3], Rakesh Heer[5,6], Luke Gaughan[6], Michael J. Morris [7], Himisha Beltran [8], Charles J. Ryan[3,9], Emmanuel S. Antonarakis [3,9], Andrew J. Armstrong [10], Susan Halabi [2] & Scott M. Dehm [3,11,12] ✉

Circulating tumor DNA (ctDNA) in plasma cell free DNA (cfDNA) of cancer patients is associated with poor prognosis, but is challenging to detect from low plasma volumes. In metastatic castration-resistant prostate cancer (mCRPC), ctDNA assays are needed to prognosticate outcomes of patients treated with androgen receptor (AR) inhibitors. We develop a custom targeted cfDNA sequencing assay, named *AR*-ctDETECT, to detect ctDNA in limiting plasma cfDNA available from mCRPC patients in the Alliance A031201 randomized phase 3 trial of enzalutamide with or without abiraterone. Of 776 patients, 59% are ctDNA-positive, with 26% having high ctDNA aneuploidy and 33% having low ctDNA aneuploidy but displaying *AR* gain or structural rearrangement, *MYC*/*MYCN* gain, or a pathogenic mutation. ctDNA-positive patients have significantly worse median overall survival than ctDNA-negative patients (29.0 months vs. 47.4 months, respectively). Here, we show that mCRPC patients identified as ctDNA-positive using the *AR*-ctDETECT assay have poor survival despite treatment with potent AR inhibitors in a phase 3 trial.

Standard of care treatments for patients with advanced prostate cancer include therapies that inhibit the androgen receptor (AR), a transcription factor essential for the homeostasis and survival of prostate cancer cells[1,2]. However, the disease will eventually progress to metastatic castration resistant prostate cancer (mCRPC), which is a disease state responsible for nearly all prostate cancer deaths. Because the majority of mCRPC remains AR dependent[3], first-line treatments for mCRPC are potent AR-targeted therapies such as enzalutamide and abiraterone, which bind and inhibit the AR directly (enzalutamide) or prevent androgen synthesis (abiraterone)[4,5]. However, approximately 20–30% of mCRPC patients have primary resistance to these agents and may instead benefit from AR-independent therapies. For instance,

[1]Minnesota Supercomputing Institute, University of Minnesota, Minneapolis, MN, USA. [2]Department of Biostatistics and Bioinformatics, Duke University, Durham, NC, USA. [3]Masonic Cancer Center, University of Minnesota, Minneapolis, MN, USA. [4]Graduate Program in Molecular, Cellular, and Developmental Biology and Genetics, University of Minnesota, Minneapolis, MN, USA. [5]Newcastle upon Tyne Hospitals NHS Foundation Trust, Newcastle upon Tyne, Tyne and Wear, UK. [6]Translational and Clinical Research Institute, NU Cancer, Newcastle upon Tyne, Tyne and Wear, UK. [7]Genitourinary Oncology Service, Department of Medicine, Memorial Sloan Kettering Cancer Center, New York, NY, USA. [8]Department of Medical Oncology, Dana Farber Cancer Institute and Harvard Medical School, Boston, MA, USA. [9]Division of Hematology, Oncology and Transplantation, Department of Medicine, University of Minnesota, Minneapolis, MN, USA. [10]Department of Medicine, Division of Medical Oncology, Duke Cancer Institute Center for Prostate and Urologic Cancers, Duke University, Durham, NC, USA. [11]Department of Laboratory Medicine and Pathology, University of Minnesota, Minneapolis, MN, USA. [12]Department of Urology, University of Minnesota, Minneapolis, MN, USA. [13]These authors contributed equally: Todd P. Knutson, Bin Luo. ✉e-mail: dehm@umn.edu

taxane chemotherapy or radium-223 have proven survival benefits in mCRPC patients[6–9]. Additionally, specific genomic features of mCRPC tumors can be used to match patients to treatment with targeted therapies, such as poly (ADP-ribose) polymerase (PARP) inhibitors for mCRPC with alterations in genes encoding homologous recombination DNA repair machinery[10–12], or the immune checkpoint inhibitor pembrolizumab for mCRPC with high microsatellite instability or DNA mismatch repair deficiency[13]. Optimizing treatment regimens with these and other AR-independent agents, as well as designing efficient clinical trials to test emerging new therapies, may be aided by prognostic biomarkers that improve on currently available clinical prognostic models in mCRPC patients[14,15].

Previous studies have provided evidence for clinical utility of analysis of cell-free DNA (cfDNA) isolated from plasma of men with mCRPC. For instance, high plasma concentrations of cfDNA are associated with worse survival outcomes in mCRPC patients treated with taxane chemotherapy[16]. Detection of circulating tumor DNA (ctDNA) in the cfDNA is associated with poor clinical outcomes[17–20], and evaluating changes in ctDNA levels may be useful for monitoring therapy response over time[21,22]. Detection of specific genomic alterations in ctDNA may also have clinical utility for prognostication, such as gain-of-function AR alterations, loss of tumor suppressor genes such as TP53, RB1, or BRCA2, or mutations in SPOP[17,18,22–27]. Commercial cfDNA assays are available for identifying patients with ctDNA, and detecting a subset of the genomic alterations that underlie mCRPC-specific resistance mechanisms[28–30]. However, there is an unmet need for cfDNA assays able to prognosticate outcomes in mCRPC patients, particularly in the context of AR-targeted therapies.

Based on these considerations, we developed AR-ctDETECT, a targeted cfDNA sequencing (cfDNA-seq) assay to interrogate alterations in AR and other known actionable alterations that occur in mCRPC patients, using low input volumes of plasma. The determination of ctDNA positivity is typically accomplished through detection of aneuploidy and/or mutations present in cancer but absent in normal cells. A unique aspect of the AR-ctDETECT cfDNA-seq assay design is the ability to identify ctDNA uniquely in mCRPC patients via comprehensive profiling of AR alterations including mutations, amplification, enhancer amplification, and gene structural rearrangements (GSRs), as well as gains in MYC and/or MYCN, due to the high sensitivity for detecting these alterations relative to normal cells. To evaluate the clinical utility of this cfDNA-seq assay for prognosticating outcomes in mCRPC patients, we studied plasma collected prior to treatment in the phase 3 Alliance A031201 trial, which randomized men 1:1 to treatment with enzalutamide or enzalutamide plus abiraterone[31]. While this trial did not demonstrate improved survival with the combination therapy, it represented an important opportunity to evaluate the AR-ctDETECT assay in both treatment arms for prognostic utility.

In this work, we detect mCRPC-specific genomic alterations at high frequency in cfDNA specimens classified as having high ctDNA aneuploidy. We assign these patients to a ctDNA aneuploidy-high Group 1 and demonstrate they have a short duration of radiographic progression-free survival (rPFS) and overall survival (OS) compared with patients assigned to a ctDNA-negative Group 3. In patients with low ctDNA aneuploidy, we identify a second ctDNA-positive group on the basis of detectable pathogenic tumor-derived mutations, AR-GSRs, and/or copy gains in AR, MYC or MYCN. We assign these patients to a ctDNA aneuploidy-low Group 2, and demonstrate they also have shorter rPFS and OS compared with patients in ctDNA-negative Group 3. These data validate the importance of evaluating these genomic alterations in patients having low ctDNA aneuploidy in a phase 3 trial context, as well as the prognostic utility of detecting ctDNA in mCRPC patients being treated with contemporary AR-targeted therapies.

## Results

### Baseline cfDNA correlates with ctDNA aneuploidy fraction

The Alliance A031201 trial was a randomized study of enzalutamide compared with enzalutamide plus abiraterone in first-line mCRPC patients[31]. The treatment combination did not demonstrate a benefit in OS, although it demonstrated a modest 3 month delay on the secondary endpoint of rPFS. Blood specimens were collected and banked for all 1311 patients evaluated in the trial, which we used for exploratory liquid biopsy analysis of plasma cell free DNA (cfDNA). Of the 1059 patients that consented to baseline blood studies, there were 790 patients with at least 2 mL of plasma collected prior to treatment that was used for cfDNA isolation (Fig. 1a). Patients with less than or equal to 2 mL of plasma were excluded to avoid exhausting any patient plasma in the biorepository. A detectable level of cfDNA was isolated from 789/790 plasma specimens. These cfDNA specimens were analyzed using a research-grade paired-end DNA sequencing (DNA-seq) assay named AR-ctDETECT, which was designed to target 820,324 bp of genomic DNA representing control regions and 69 genes displaying recurrent alterations in mCRPC tumors (Fig. 1b, Supplementary Data 1).

We performed targeted DNA-seq on 789 cfDNA samples, of which 776 passed quality control checks during library preparation and DNA-seq (Fig. 1a). These 776 patients had similar clinical baseline characteristics, treatment assignments, and outcomes to the 535 patients from the Alliance A031201 trial for which cfDNA-seq data were not available for analysis (Supplementary Table 1). Samples having very low cfDNA concentrations displayed higher percentages of DNA-seq reads attributed to PCR duplicates than those having high cfDNA concentrations (Supplementary Fig. 1A). Accordingly, the cfDNA concentration of a sample was positively related to the mean unique read coverage of that sample after removal of PCR duplicates, which averaged 401X across all 776 samples (Supplementary Fig. 1B). Comparable depths of coverage were achieved for all targeted genes (Supplementary Fig. 1C).

The ichorCNA algorithm was developed to estimate circulating tumor DNA (ctDNA) fraction from aneuploidy detected in low-pass whole genome sequencing of cfDNA[32]. Previous studies demonstrated the off-target reads from targeted DNA-seq assays can be used to approximate low-pass whole genome sequencing and be leveraged for estimating ctDNA fraction[33,34]. Therefore, we removed the on-target reads from each sample and used the remaining off-target reads as input in ichorCNA. The median estimate from this approach was 9.4% ctDNA aneuploidy, with a range of 0.4 to 84.6% ctDNA aneuploidy (Fig. 1c, Supplementary Data 2). The ctDNA aneuploidy fraction was positively correlated with the yield of cfDNA from the corresponding plasma specimens (Fig. 1c, Supplementary Data 2). A test of the association of ctDNA aneuploidy fraction with clinical outcomes revealed that the hazard ratio (HR) for rPFS was 1.3 (95% CI: 1.2–1.4, P < 0.0001) and the HR for OS was 1.5 (95% CI: 1.4–1.6, P < 0.0001) for each 0.1 unit (10%) increase in ctDNA aneuploidy fraction.

Because ichorCNA can provide a ctDNA aneuploidy fraction estimate for any cfDNA sample, including those from healthy donors[32], we also used the AR-ctDETECT assay to analyze control plasma pooled from healthy donors. In these control plasma samples, ichorCNA provided estimates ranging from 4.7–12.2% ctDNA aneuploidy (mean = 7.0% and standard deviation = 2.4%), which we used to derive an assay-specific threshold of 14.2% to distinguish cfDNA samples with high ctDNA aneuploidy (3 standard deviations beyond the control sample mean, Fig. 1d).

### Baseline cfDNA displays a high burden of AR gene alterations

Based on this assay-specific ctDNA aneuploidy threshold, we classified 26% of the plasma cfDNA specimens above the threshold as ctDNA aneuploidy-high (Fig. 2a). An additional method used to identify ctDNA in cfDNA is to leverage the detection of somatic mutations in genes targeted by DNA-seq[17,35,36]. We identified 309 mutations that were pathogenic and classified as likely somatic (Supplementary Data 3). As

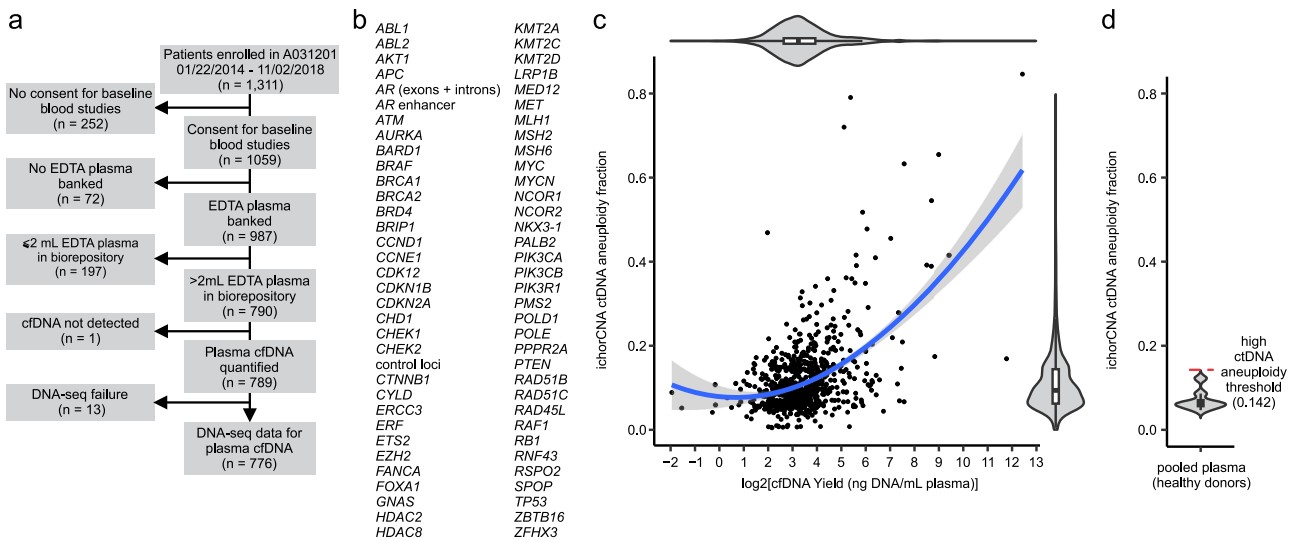

**Fig. 1 | Cell free DNA (cfDNA) yield and circulating tumor DNA (ctDNA) fraction in A031201 plasma specimens. a** CONSORT flow diagram of patients enrolled in A031201 and plasma cfDNA specimens analyzed by targeted DNA-seq. 'n' refers to the number of patients. **b** AR-ctDETECT targeted DNA-seq assay design. **c** Scatterplot of ctDNA aneuploidy fraction estimated by ichorCNA using off-target DNA-seq reads vs. cfDNA yield from 776 plasma specimens analyzed by targeted DNA-seq. Loess trendline and 95% confidence intervals are shown. Violin plots illustrate data density from min to max. Boxes in violin plots illustrate median and interquartile range. Whiskers illustrate 1.5X interquartile range. **d** ctDNA aneuploidy fraction estimate as in (**c**) using cfDNA isolated from pooled plasma from healthy donors. Violin plots illustrate data density from min to max. Boxes in violin plots illustrate median and interquartile range. Whiskers illustrate 1.5X interquartile range. An assay cutoff of mean + 3 standard deviations (0.142) distinguishes ctDNA aneuploidy-high specimens.

expected, the majority of these pathogenic mutations occurred in ctDNA aneuploidy-high specimens. However, there were an additional 97 specimens below the ctDNA aneuploidy cutoff that harbored one or more of these pathogenic mutations, which we used to classify them as ctDNA-positive (Fig. 2a). We noted that the patients with detected pathogenic mutations had a higher average yield of cfDNA isolated from their plasma specimens compared to those without detected pathogenic mutations (Supplementary Fig. 2). This finding may indicate a higher sensitivity for detecting ctDNA-specific mutations in plasma with greater cfDNA amounts, or higher levels of ctDNA in cfDNA with detected pathogenic mutations.

Few oncogenes and tumor suppressor genes are altered at high frequency by pathogenic mutations in mCRPC[37–41]. Instead, mCRPC harbors a high degree of structural and copy number variation, and AR is frequently impacted by these events[38,42]. Therefore, we asked whether such AR alterations were detectable in cfDNA. The AR gene body displayed copy number gain in 198 cfDNA samples and the AR upstream enhancer displayed copy number gain in 207 cfDNA samples (Fig. 2b, c, Supplementary Data 4). As expected, these AR copy gains were detected in many of the 200 ctDNA aneuploidy-high samples and the 97 ctDNA aneuploidy-low samples having a pathogenic SNV. However, these alterations were also apparent in many of the 479 cfDNA samples in our assay that would otherwise be classified as ctDNA-negative. Similarly, AR-GSRs, defined as structural variants with at least one breakpoint transecting the AR gene body[42–47], were detected in 175 cfDNA samples spanning those that were ctDNA-positive and otherwise ctDNA-negative (Fig. 2d, Supplementary Data 5). AR mutations (or single nucleotide variants, SNVs) were the least frequent AR alteration, occurring in 20 cfDNA samples (Fig. 2e, Supplementary Data 3). The most common AR mutation encoded a H875Y substitution in the AR protein.

### AR-GSRs in baseline cfDNA associate with poor clinical outcomes

Functionally, certain AR-GSRs have been shown to promote synthesis of truncated, constitutively active AR variant (AR-V) proteins lacking the AR ligand binding domain (LBD). There were a total of 457 AR-GSRs detected, with 57% of the AR-GSR-positive samples having 2 or more AR-GSRs (Supplementary Data 6). By drawing on previous examples of how certain AR-GSRs alter AR mRNA splicing to drive AR-V expression[42–47], we classified 54 AR-GSRs occurring in 22 cfDNA samples as likely LBD-truncating events (Fig. 2d, Supplementary Fig. 3, Supplementary Data 6). Additionally, AR being amplified on extra-chromosomal circular DNA (ecDNA) was recently identified as a mechanism that promotes high AR copy number and accumulation of complex AR-GSRs in individual CPRC cells[40]. Therefore, we classified 60 cfDNA samples with gain of the AR gene body and 2 or more AR-GSRs as likely harboring AR ecDNA (Fig. 2d, f, Supplementary Data 5). Noteworthy, cfDNA samples harboring AR-GSRs that were likely LBD-truncating and/or AR ecDNA-associated occurred in ctDNA-positive samples as well in cfDNA samples that were otherwise ctDNA-negative.

A prospective hypothesis of this study was that patients with AR-GSRs would have worse rPFS and OS. In univariate analysis and multivariable analysis corrected for ctDNA aneuploidy fraction, patients harboring AR-GSRs had worse rPFS and OS compared with patients that lacked AR-GSRs (Fig. 2g, h). In univariate analysis, the hazard ratios for worse rPFS and OS were even higher in patients harboring AR-GSRs that were likely LBD-truncating or likely associated with AR ecDNA (Fig. 2g, h). In multivariable analysis corrected for ctDNA aneuploidy fraction, these associations for AR-GSRs that were likely LBD-truncating or likely associated with AR ecDNA remained significant, with the exception of the association between likely LBD-truncating AR-GSRs and OS. We noted that many of the samples with AR-GSRs were below the ctDNA aneuploidy fraction cutoff, which could impact the accuracy of using ctDNA aneuploidy fraction for multivariable correction. Therefore, we tested the association of ctDNA aneuploidy fraction as a continuous variable with clinical outcomes in the 479 patients with low ctDNA aneuploidy and lacking a detectable pathogenic mutation. Remarkably, even in this subgroup, ctDNA aneuploidy fraction remained prognostic for rPFS (HR = 1.05 for each 1% increase in ctDNA aneuploidy fraction, 95% CI: 1.01–1.09) and OS (HR = 1.05 for each 1% increase in ctDNA aneuploidy fraction, 95% CI: 1.01–1.09).

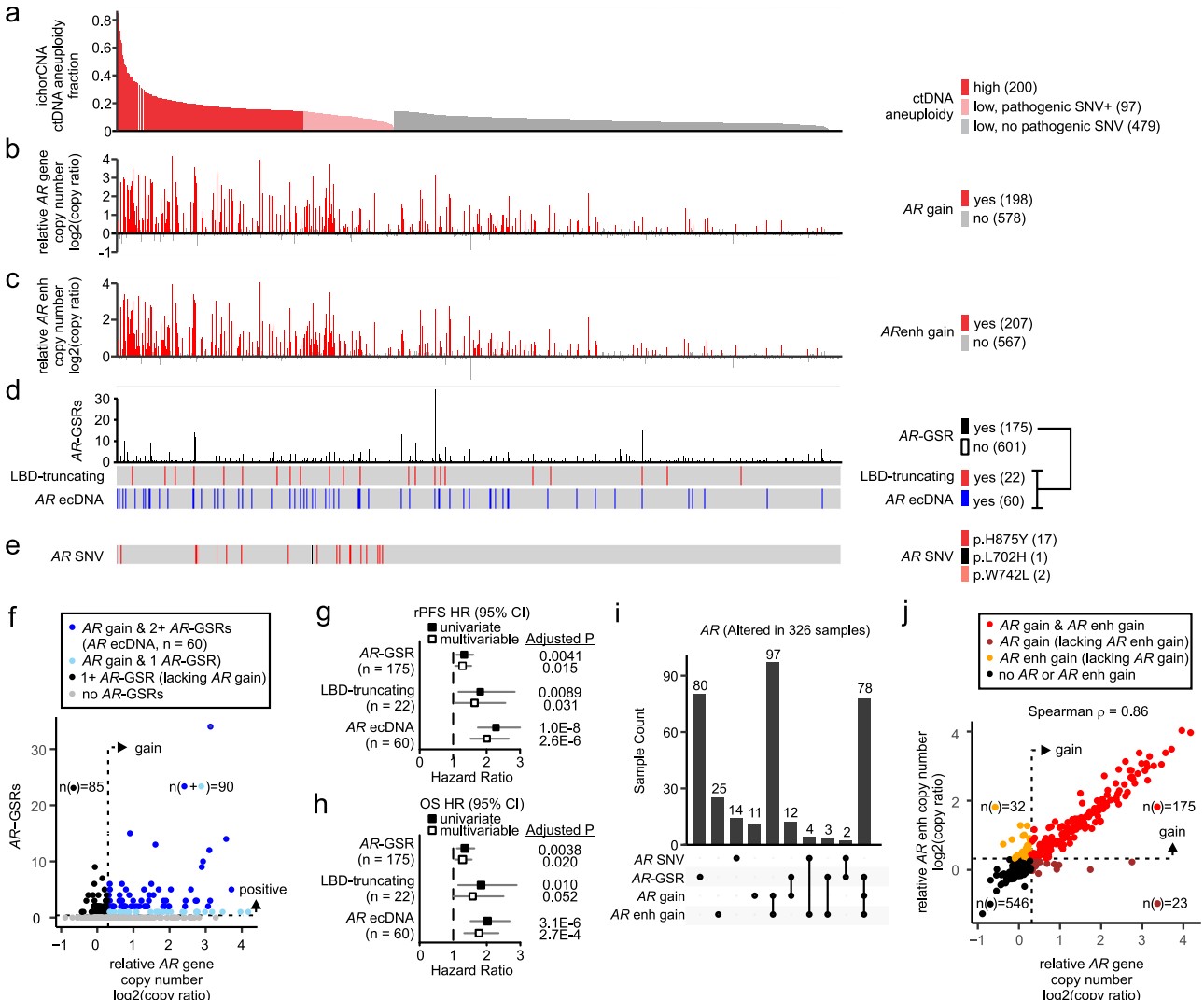

**Fig. 2 | Comprehensive profiling of *AR* genomic alterations in A031201 cfDNA.** **a** ctDNA aneuploidy fraction estimated by ichorCNA in targeted DNA-seq data from n = 776 cfDNA specimens. Samples below the ctDNA aneuploidy-high threshold of 0.142 are in gray/pink. **b** Relative *AR* gene copy number ratio in cfDNA specimens ordered left to right as in (**a**). A log2 ratio >= 0.3 is considered a gain. **c** Relative *AR* upstream enhancer copy number ratio in cfDNA specimens ordered left to right as in (**a**). A log2 ratio >= 0.3 is considered a gain. **d** Number of *AR* gene structural rearrangements (*AR*-GSRs) in cfDNA specimens ordered left to right as in (**a**). *AR*-GSRs predicted to encode AR variant proteins with a truncated ligand binding domain (LBD), and *AR*-GSRs predicted to be associated with *AR* amplification on extrachromosomal DNA (ecDNA), are shown. **e** *AR* single nucleotide variants (SNVs) detected in cfDNA specimens as in (**a**). **f** Scatterplot of relative *AR* gene copy number ratio and number of *AR*-GSRs. Dots are colored blue based on copy number gain of the *AR* gene body. Dark blue denotes samples with 2 or more *AR*-GSRs and

*AR* gene body copy gain, which is a signature of *AR* ecDNA (shown also in (**d**)). 'n' refers to the number of patient samples. Forest plots illustrating hazard ratio (squares) and 95% confidence intervals (horizontal lines) for (**g**) radiographic progression (rPFS) and (**h**) overall survival (OS) in patients demonstrating indicated cfDNA features. Multivariable analysis is adjusted for ctDNA aneuploidy fraction. P-values are from the Wald test from the Cox's proportional hazards model, adjusted for multiplicity using the Benjamini-Hochberg method (false discovery rate). A FDR < 0.05 is considered statistically significant. 'n' refers to the number of patients. **i** UpSet plot showing co-occurrence of indicated *AR* genomic alterations. **j** Scatterplot of relative *AR* gene copy number ratio and *AR* upstream enhancer copy number ratio. Dots are colored based on copy number gain of the *AR* gene body only (brown), *AR* upstream enhancer only (gold), or both the *AR* gene body and *AR* upstream enhancer (red). 'n' refers to the number of patient samples.

## *AR* alterations and *MYC*/*MYCN* gains identify ctDNA-positive patients

Collectively, 326 of all cfDNA samples analyzed by DNA-seq displayed at least one *AR* gene alteration (Fig. 2i). Consistent with previous studies of mCRPC tissue, the most frequent co-occurring *AR* gene alterations were copy number gains of the *AR* gene body and upstream enhancer[38,48,49]. Only 23 of the cfDNA samples with *AR* gene body gain lacked an accompanying *AR* upstream enhancer gain, and 32 of the cfDNA samples with *AR* upstream enhancer gain lacked an accompanying *AR* gene body gain (Fig. 2i, j). Indeed, there was very high correlation in the relative copy number of the *AR* gene body and upstream enhancer, which is consistent with the notion that these two

genomic regions are usually contained on the same amplicon in mCRPC tissues (Fig. 2j)[38,48,49].

Because mCRPC-specific *AR* amplification was apparent in cfDNA that might otherwise be classified as ctDNA-negative, we assessed whether additional amplification events could be leveraged for their high ctDNA-specific signal (Supplementary Data 4). Amplification of *MYC* and *MYCN* are somatic events that occur with high frequency in mCRPC genomes[38,39,50,51]. *MYC* displayed copy number gain in 90 cfDNA samples and *MYCN* gain displayed copy number gain in 39 cfDNA samples (Fig. 3a, Supplementary Data 4). Although most of these *MYC* and/or *MYCN* copy number gains occurred in ctDNA-positive samples, they were also detectable in cfDNA samples that

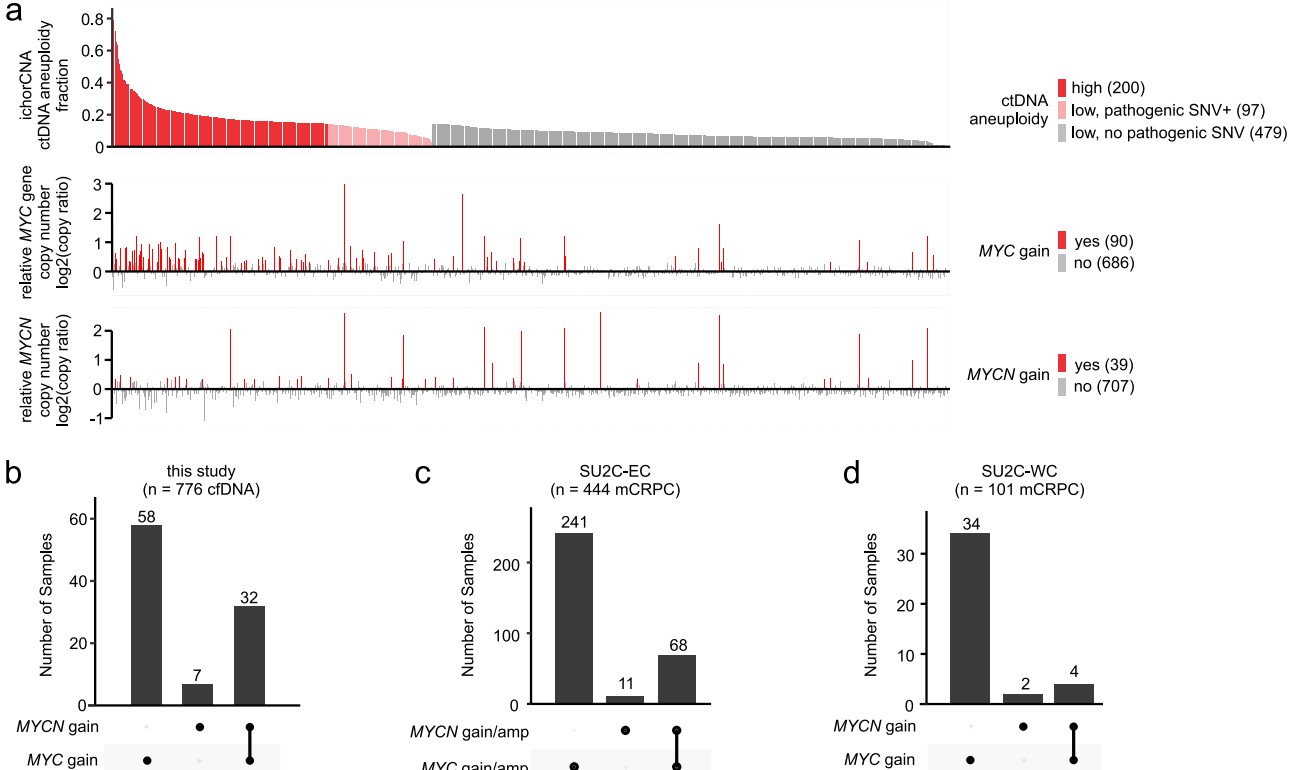

**Fig. 3 | *MYC* and *MYCN* copy number gains in A031201 cfDNA. a** Relative *MYC* and *MYCN* gene copy number ratios in cfDNA specimens based on ctDNA aneuploidy fraction estimated by ichorCNA (samples below the ctDNA aneupoidy-high threshold of 0.142 are in gray/pink). A log2 ratio >= 0.3 is considered a gain. **b** UpSet plot showing co-occurrence of *MYC* and *MYCN* copy number gains in cfDNA samples from (**a**). **c** UpSet plot showing co-occurrence of *MYC* and *MYCN* copy number gains in mCRPC biopsy specimens from 444 patients analyzed in the AACR-PCF Stand Up To Cancer East Coast study. **d** UpSet plot showing co-occurrence of *MYC* and *MYCN* copy number gains in mCRPC biopsy specimens from 101 patients analyzed in the AACR-PCF Stand Up To Cancer West Coast study.

were otherwise cfDNA-negative (Fig. 3a). Notably, 82% of the cfDNA samples that displayed *MYCN* gain also displayed *MYC* gain (Fig. 3b). To the best of our knowledge, concurrent gain of *MYC* and *MYCN* has not been reported in mCRPC previously. To confirm this was not an artifact of the *AR*-ctDETECT assay or analysis of cfDNA, we evaluated *MYC* and *MYCN* gain in two public datasets of whole exome or whole genome DNA-seq analysis of mCRPC tissue biopsies[38,52]. Both of these datasets confirmed this co-occurrence, with 86% or 67% of the mCRPC tissues with *MYCN* gains also having *MYC* gains (Fig. 3c, d).

Based on our observations that alterations of *AR*, *MYC*, or *MYCN* were detectable in cfDNA samples that would otherwise be classified as ctDNA-negative by aneuploidy or mutation-based methods, we evaluated the clinical significance of incorporating *AR*, *MYC* or *MYCN* alterations into ctDNA detection. Across the 479 patients that were ctDNA aneuploidy-low and lacked a pathogenic mutation, OS and rPFS were longer in the 320 patients that lacked detectable alterations in *AR*, *MYC* or *MYCN* compared with the 159 patients where these alterations were detected (Supplementary Fig. 4A, B). Based on these supporting results, we stratified the 776 cfDNA samples into 3 groups. The first group of 200 cfDNA samples were those originally classified by ichorCNA as ctDNA aneuploidy-high (Fig. 4a). The second group of 256 cfDNA samples were those classified by ichorCNA as ctDNA aneuploidy-low, but deemed to be ctDNA-positive on the basis of harboring one or more likely somatic pathogenic mutations in any targeted gene and/or detectable alterations in *AR*, *MYC* or *MYCN* (Fig. 4b, c, Supplementary Fig. 5). Overall, 59% of all cfDNA samples were ctDNA positive based on these classifications (Fig. 4c). *AR* alterations occurred in 71% of ctDNA-positive specimens, with the highest frequency in ctDNA aneuploidy-low Group 2 (Fig. 4d). In contrast, *MYC* and/or *MYCN* alterations occurred in 21% of ctDNA-

positive specimens, with the highest frequency in ctDNA aneuploidy-high Group 1 (Fig. 4d).

## Alterations in common mCRPC genes are detectable in ctDNA
We next assessed mutations in *TP53*, *PTEN*, and *RB1* in ctDNA aneuploidy-high Group 1 and ctDNA aneuploidy-low Group 2, as alterations in these tumor suppressor genes have been associated with worse outcomes and lineage plasticity in men with mCRPC[17,18,52]. *TP53* displayed the highest mutational frequency among all the genes targeted for DNA-seq analysis, with 15.3% of the ctDNA-positive samples harboring a *TP53* mutation (Fig. 4e, Supplementary Data 3). The mutational frequencies of *PTEN* (2.9% of ctDNA-positive samples) and *RB1* (0.7% of ctDNA-positive samples) were much lower.

In addition to mutations, substantial copy number loss was detectable for *TP53*, *PTEN*, and *RB1* in ctDNA-positive samples, although we were not able to distinguish whether these were 1- or 2-copy losses (Fig. 4f–h, Supplementary Data 4). By aggregating the data for mutations and copy number losses, we observed that *TP53* was altered in 25% of the ctDNA-positive samples (Fig. 4i), *PTEN* was altered in 29% of the ctDNA-positive samples (Fig. 4j), and *RB1* was altered in 31% of the ctDNA-positive samples (Fig. 4k). Comparing the proportions of samples belonging to ctDNA aneuploidy-high Group 1 or ctDNA aneuploidy-low Group 2 that harbored detectable mutations and/or copy number loss in *TP53*, *PTEN*, or *RB1* revealed 2.1–2.4 fold higher rates of alterations in these genes in samples belonging to ctDNA aneuploidy-high Group 1 (Fig. 4l). Overall, 46% of the ctDNA-positive samples displayed at least 1 alteration in *TP53*, *PTEN*, or *RB1* (Fig. 4m). Further, 13% of the ctDNA-positive samples displayed alterations in all 3 tumor suppressor genes, with all but 6 of these 60 samples belonging to ctDNA-high Group 1.

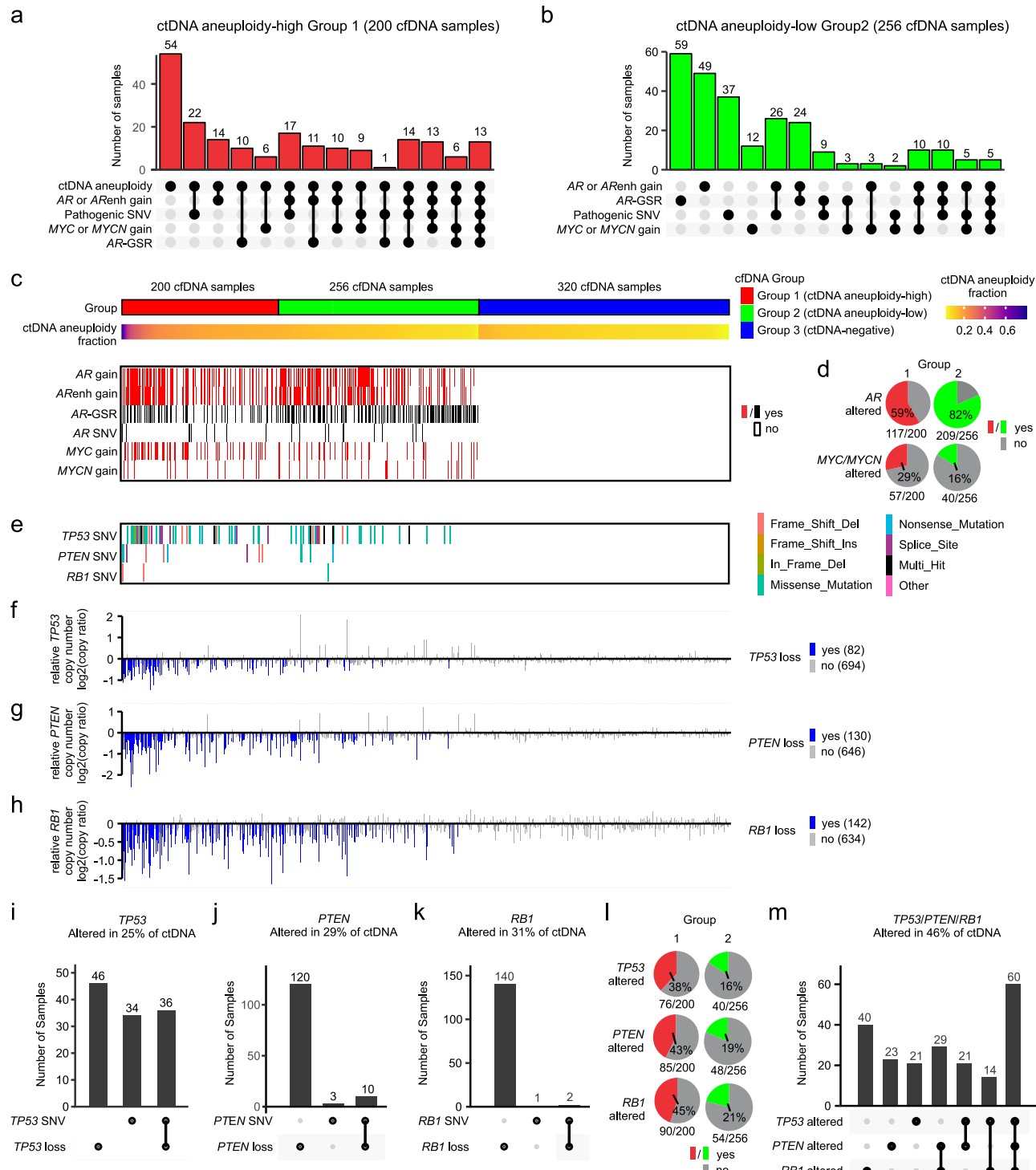

**Fig. 4 | Prevalent genomic alterations in ctDNA-positive A031201 cfDNA.** UpSet plots showing relationships between ctDNA features and genomic alterations in (**a**) ctDNA aneuploidy-high Group 1 and (**b**) ctDNA aneuploidy-low Group 2. **c** cfDNA specimens stratified into 3 groups based on samples exceeding a ctDNA aneuploidy threshold (ctDNA aneuploidy-high Group 1), samples below the ctDNA aneuploidy threshold but harboring a likely somatic pathogenic mutation, an *AR*-GSR, and/or copy number gain of *AR*, *MYC* or *MYCN* (ctDNA aneuploidy-low Group 2), or samples below the ctDNA aneuploidy threshold and lacking a likely somatic pathogenic mutation, an *AR*-GSR, or copy number gain of *AR*, *MYC* or *MYCN* (ctDNA-negative Group 3). **d** Pie charts illustrating frequency of *AR* or *MYC/MYCN* alterations detected in ctDNA aneuploidy-high Group 1 or ctDNA aneuploidy-low Group 2. **e** *TP53*, *PTEN*, and *RB1* single nucleotide variants (SNVs) detected in ctDNA-positive cfDNA specimens ordered left to right as in (**c**). Relative (**f**) *TP53*, (**g**) *PTEN*, or (**h**) *RB1* gene copy number ratios in cfDNA specimens ordered left to right as in (**c**). A log2 ratio <= −0.3 in a ctDNA-positive sample is considered a loss. UpSet plots showing relationships between (**i**) *TP53*, (**j**) *PTEN*, or (**k**) *RB1* SNVs and copy number losses in ctDNA-positive cfDNA samples. **l** Pie charts illustrating frequency of *TP53, PTEN, or RB1* alterations detected in ctDNA aneuploidy-high Group 1 or ctDNA aneuploidy-low Group 2. **m** UpSet plot showing relationships between alterations in *TP53, PTEN,* and *RB1* in ctDNA-positive cfDNA specimens.

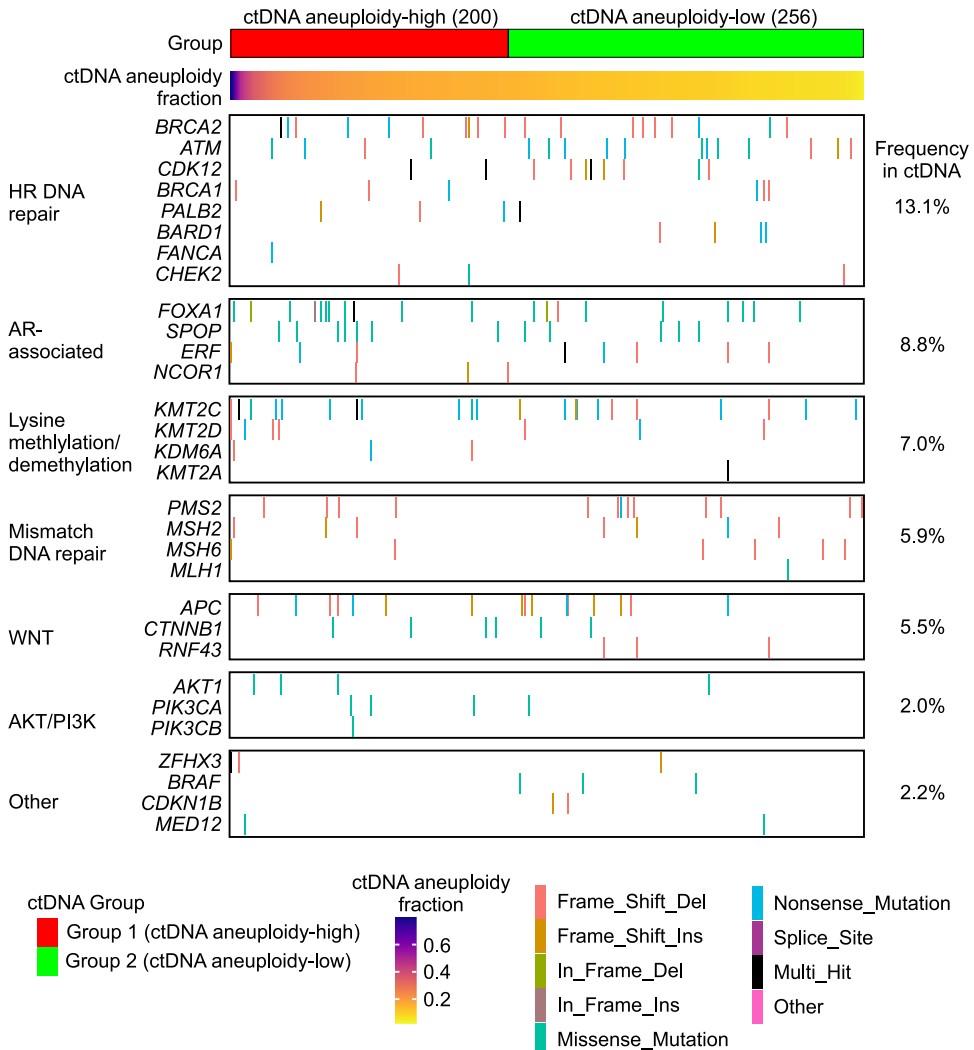

**Fig. 5 | Pathogenic mutations detected in A031201 ctDNA.** Oncoprint of single nucleotide variants (SNVs) occurring in indicated genes across ctDNA specimens in ctDNA aneuploidy-high Group 1 and ctDNA aneuploidy-low Group 2. The frequency of SNVs within these groups is indicated at the right. Samples are ordered left to right by descending ctDNA aneuploidy fraction estimate.

Extending the analysis to include all genes targeted by DNA-seq revealed an additional 30 genes that harbored detectable pathogenic mutations in ctDNA-positive samples (Fig. 5, Supplementary Data 3). Because the mutational frequency of any individual gene was low, we grouped them functionally to assess their prevalence among ctDNA-positive samples. The most frequently-altered pathways were those associated with homologous recombination DNA repair and AR regulation. The mutation frequency for genes in these and other pathways were similar between ctDNA aneuploidy-high Group 1 samples and ctDNA aneuploidy-low Group 2 samples.

We used the genetic alterations detected in ctDNA to benchmark the ctDNA aneuploidy fraction estimates derived from using off-target reads as input for ichorCNA. For instance, the variant allele fraction (VAF) of somatic tumor-derived mutations has been used for calculating ctDNA fraction in studies where high plasma volumes yielded high cfDNA concentrations, enabling high sequencing depth and sensitive mutation detection[17,28]. We found that the VAFs of mutations detected across 183 cfDNA specimens were higher in samples having high ctDNA aneuploidy than in samples having low ctDNA aneuploidy (Supplementary Fig. 6A, B). Further, the maximum VAF detected in each of these 183 cfDNA specimens displayed a positive correlation with the ctDNA aneuploidy fraction (Supplementary Fig. 6C). This positive correlation between maximum VAF and ctDNA aneuploidy fraction was highest for *TP53*, which was the most frequently-mutated gene in our study (Supplementary Fig. 6C). Another input that has been used to calculate ctDNA fraction is the magnitude of tumor-derived somatic copy number gains or losses of targeted genes[21,25,36]. We observed positive correlations between the ctDNA aneuploidy fraction and magnitude of copy number gain detected in the *AR* gene body or enhancer in cfDNA samples where these gains occurred (Supplementary Fig. 6D, E), and a negative correlation between the ctDNA aneuploidy fraction and magnitude of copy number loss in *TP53*, *PTEN*, or *RB1* in cfDNA samples where these losses occurred (Supplementary Fig. 6F–H). Finally, CNVkit is an algorithm to infer and visualize tumor-derived copy number alterations from targeted DNA-seq data[53]. Visual inspection of the CNVkit output for all 776 plasma specimens confirmed a high degree of genome-wide copy number gains and losses in samples predicted by ichorCNA to have high ctDNA aneuploidy, and a relative paucity of genome-wide copy number gains and losses in samples below the ichorCNA ctDNA aneuploidy threshold (Supplementary Fig. 6I). Collectively, these benchmarking data demonstrate that the ctDNA aneuploidy fraction estimated by ichorCNA correlates with inputs used to calculate ctDNA fraction in cfDNA.

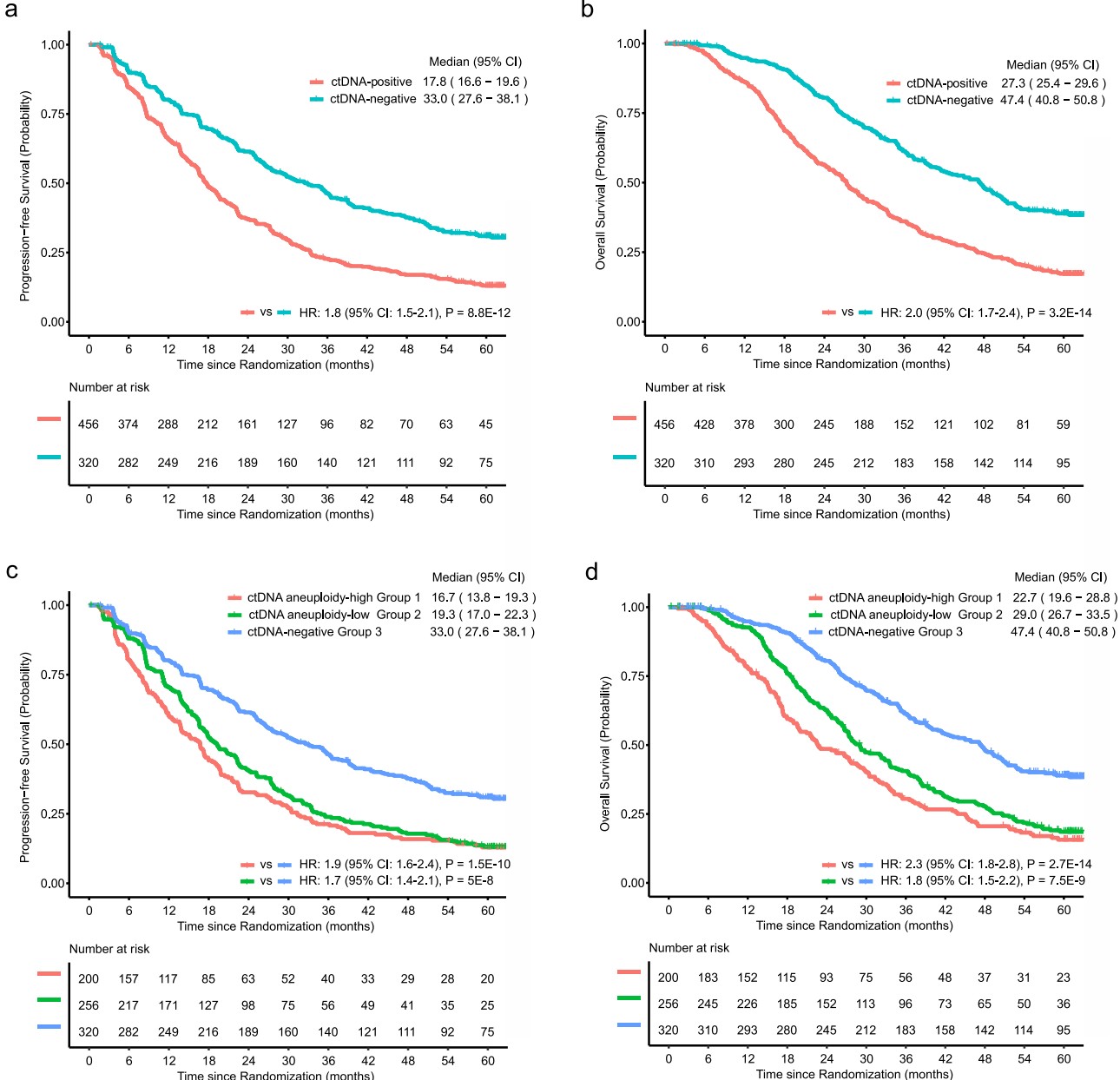

**Fig. 6 | Survival of ctDNA-positive A031201 patients.** Kaplan-Meier plots of (**a**) radiographic progression-free survival and (**b**) overall survival in patients classified as ctDNA-positive or ctDNA-negative. Kaplan-Meier plots of (**c**) radiographic progression-free survival and (**d**) overall survival in patients classified as belonging to ctDNA aneuploidy-high Group 1, ctDNA aneuploidy-low Group 2, or ctDNA-negative Group 3.

## ctDNA-positive patients have poor clinical outcomes

A prospective hypothesis of this study was that ctDNA-positive patients would have poor prognosis. High plasma concentrations of cfDNA are associated with worse survival outcomes in mCRPC patients treated with taxane chemotherapy[16], which is consistent with our observation that mCRPC patients in ctDNA aneuploidy-high Group 1 had the highest average plasma cfDNA concentration, patients in ctDNA-negative Group 3 had the lowest average plasma cfDNA concentration, and patients in ctDNA aneuploidy-low Group 2 had an intermediate average plasma cfDNA concentration (Supplementary Fig. 7). We next evaluated relationships with prognostic risk at baseline. Compared to patients in ctDNA-negative Group 3, patients in ctDNA aneuploidy-high Group 1 or ctDNA aneuploidy-low Group 2 were more likely to be classified as high risk and less likely to be classified as low risk, and had higher Halabi Risk scores, which are

based on a multivariate prognostic model of OS (Supplementary Table 2)[14,31,54].

To further test this prospective hypothesis, we performed survival analysis. Patients negative for ctDNA had median rPFS of 33.0 months (95% CI 27.6–38.1 months) and median OS of 47.4 months (95% CI 40.8–50.8 months). In contrast, ctDNA-positive patients had shorter median rPFS of 17.8 months (95% confidence interval 16.6–19.6 months, Fig. 6a) and shorter median OS of 27.3 months (95% confidence interval 25.4–29.6 months, Fig. 6b). The HR for rPFS was 1.8 (95% CI: 1.5–2.1, P = 8.8E–12) and the HR for OS was 2.0 (95% CI: 1.7-2.4, P = 3.2E–14) in ctDNA-positive vs. ctDNA-negative patients.

When ctDNA-positive patients were stratified into ctDNA aneuploidy-high Group 1 and ctDNA-aneuploidy-low Group 2, patients in Group 1 had median rPFS of 16.7 months (95% CI 13.8–19.3 months, Fig. 6c). The HR for rPFS was 1.9 (95% CI: 1.6–2.4, P = 1.5E–10) in Group 1

compared with Group 3. Patients in Group 2 had a median rPFS of 19.3 months (95% confidence interval 17.0–22.3 months). The HR for rPFS was 1.7 (95% CI: 1.4–2.1, P = 5E-8) for Group 2 compared with Group 3. There was no statistical difference in rPFS between Groups 1 and 2 (HR = 1.1, 95% CI: 0.9–1.4, P = 0.21).

Patients in ctDNA aneuploidy-high Group 1 had the shortest OS duration with a median of 22.7 months (95% CI 19.6–28.8 months, Fig. 6d). The HR for OS was 2.3 (95% CI: 1.8–2.8, P = 2.7E-14) in Group 1 compared with ctDNA-negative Group 3. In contrast, patients in ctDNA aneuploidy-low Group 2 had a median OS of 29.0 months (95% CI 26.7–33.5 months, Fig. 6d). The HR for OS was 1.8 (95% CI 1.5–2.2, P = 7.5E-9) in Group 2 compared with ctDNA-negative Group 3. The HR for OS was higher in Group 1 compared with Group 2 (HR = 1.3, 95% CI: 1.0–1.6, P = 0.03).

When the patients in ctDNA aneuploidy-low Group 2 were analyzed as two subgroups, the 97 patients in Group 2 that harbored detectable pathogenic mutations had worse rPFS and OS than patients in ctDNA-negative Group 3 (Supplementary Fig. 8A–D), but rPFS and OS that were indistinguishable from ctDNA aneuploidy-high Group 1 (Supplementary Fig. 8E, F). The 159 patients in Group 2 that lacked detectable pathogenic mutations but harbored AR-GSRs and/or a copy gain in AR, MYC, or MYCN had worse rPFS and OS than patients in ctDNA-negative Group 3 (Supplementary Fig. 8A–D), and better rPFS (but not OS) than patients in ctDNA aneuploidy-high Group 1 (Supplementary Fig. 8E, F). Collectively, these data demonstrate clinical utility of leveraging AR-GSRs and/or a copy gain in AR, MYC, or MYCN along with more traditional use of somatic mutations for detecting ctDNA in patients with low/negative ctDNA aneuploidy. Overall, patients classified as ctDNA-positive by the AR-ctDETECT cfDNA sequencing assay have poor clinical outcomes despite treatment with potent AR inhibition in a phase 3 trial.

## Discussion

In this study, we developed the AR-ctDETECT targeted DNA-seq assay to analyze cfDNA specimens from men with newly-diagnosed mCRPC enrolled in the randomized phase 3 Alliance A031201 trial testing enzalutamide vs. enzalutamide plus abiraterone[31]. We analyzed 776 cfDNA specimens for which at least 2 mL plasma was available, and stratified patients into 3 groups based on distinguishing characteristics in their cfDNA. There were 200 patients in ctDNA aneuploidy-high Group 1, which were ctDNA-positive based on a definition of exceeding an assay-specific ctDNA aneuploidy fraction greater than 14.2% estimated by the ichorCNA algorithm[32]. However, using a strategy of defining as ctDNA-positive those patients harboring a known pathogenic mutation, an AR-GSR, and/or a copy gain in AR, MYC or MYCN, we identified 256 patients in a ctDNA aneuploidy-low Group 2 that had a poor prognosis despite having an estimated ctDNA aneuploidy fraction at or below the assay-specific threshold and receiving potent AR inhibition. Over 80% of the patients in ctDNA aneuploidy-low Group 2 had at least 1 AR alteration These molecular alterations have a high signal to noise ratio in cfDNA given their high level of gain, and thus detection ability is enhanced as compared to copy losses or lower level gains. Overall survival was shorter in patients belonging to ctDNA aneuploidy-high Group 1 and ctDNA aneuploidy-low Group 2 compared with ctDNA-negative Group 3, which is an important finding of our work. This suggests that the ctDNA aneuploidy-low Group 2, which is mainly defined by AR alterations, has a poor prognosis despite having a low level of ctDNA aneuploidy. Interesting, the patients in ctDNA aneuploidy-high Group 1 had worse OS (but not rPFS) compared to patients in ctDNA aneuploidy-low Group 2. Given that ctDNA aneuploidy fraction correlates with other inputs used to calculate ctDNA fraction, this finding confirms previous work showing the prognostic significance of ctDNA fraction in patients[19]. Collectively, these clinical associations support our unique strategy of leveraging the high-signal copy number gains in AR, MYC, and MYCN, as well as AR-

GSRs and combining them with the more traditional use of detectable pathogenic mutations to classify otherwise ctDNA-low/negative cfDNA samples as being ctDNA-positive.

In a previous study of cfDNA from 3334 advanced prostate cancer patients using a commercial targeted DNA-seq assay, the fraction of ctDNA-positive patients was greater than 95%[28]. Similar to our work, this study also used a hybrid approach for determining ctDNA, first by using a proprietary algorithm similar to ichorCNA that assesses ctDNA aneuploidy fraction, and then by using the maximum VAF of tumor-specific mutations in samples where ctDNA aneuploidy fraction was below 10%. Approximately half of the cfDNA specimens analyzed in that prior study were isolated from pre-treatment plasma samples collected from patients enrolled in the phase 3 TRITON2 and TRITON3 trials of patients that had progressed on abiraterone and/or enzalutamide, meaning the patients were at a more advanced disease stage than patients enrolled in the current phase 3 A031201 trial. Conversely, other targeted DNA-seq studies of pre-treatment cfDNA isolated from patients at a disease stage and treatment regimen more comparable to A031201 patients reported ctDNA fractions of 47–59%, which is in-line with the 59% ctDNA-positivity reported in our study[17,22,55]. Because these prior studies were performed with samples collected in the context of phase II trials, the patient cohorts were smaller and were recruited from a limited number of sites. Nevertheless, they established that patients with high ctDNA had worse OS than patients with low ctDNA[17,22], or that patients who were positive for ctDNA had worse OS than patients negative for ctDNA[55]. However, to the best of our knowledge, our study is the first to demonstrate in a phase 3 cohort that ctDNA-positive mCRPC patients treated with first-line potent AR-targeted therapies had significantly worse OS compared to ctDNA-negative mCRPC patients.

When we combined ctDNA-positive patients in ctDNA aneuploidy-high Group 1 and ctDNA aneuploidy-low Group 2, we found they had a median OS of 27.3 months (95% confidence interval, 25.4–29.6 months), whereas ctDNA-negative patients in Group 3 had a median OS of 47.4 months (95% confidence interval, 40.8–50.8 months). Adverse clinical factors have also been used to stratify all 1311 patients in the A031201 trial into Halabi Risk Factor groups, with a 3-tier prognostic model of high, intermediate, and low risk groups stratifying patients by OS[14]. As expected, we noted associations between ctDNA status and these Halabi Risk Factor groups, including a higher frequency of the high-risk classification among patients positive for ctDNA, and a higher frequency of the low-risk classification among patients negative for ctDNA. However, there were many patients in the low- and intermediate-risk groups that were in ctDNA aneuploidy-high Group 1 and ctDNA aneuploidy-low Group 2, indicating a potential for success with our planned future study to combine adverse clinical factors and ctDNA detection into a unified clinical-genomic model for improving patient prognostication.

Our study also highlights a potential utility of using targeted DNA-seq of cfDNA to identify key alterations in mCRPC genomes. Among the 456 ctDNA-positive cfDNA samples analyzed in this study, AR was altered in 326 samples, nominating it as the most frequently-altered gene in this context. The three most common AR alterations were copy gain of the AR upstream enhancer, copy gain of the AR gene body, and AR-GSRs, which alter the structure of the AR gene[42–47]. The frequent co-occurrence of these AR alterations confirms previous studies showing that a substantial proportion of mCRPC is characterized by complex patterns of amplified and rearranged AR gene structures, which in certain cases can be explained by AR ecDNA[22,40,42,56]. The majority of the cfDNA samples in ctDNA aneuploidy-low Group 2 were classified as ctDNA-positive as a result of displaying these complex patterns of AR alterations. The fact that patients belonging to ctDNA aneuploidy-low Group 2 demonstrated a worse OS relative to ctDNA-negative Group 3 is consistent with the notion that these AR alterations drive broad resistance to AR-targeted therapies, including potent drugs like

enzalutamide and/or abiraterone. It should be noted that all patients in the phase 3 A031201 trial were treated with enzalutamide or enzalutamide plus abiraterone, so it will be important to determine whether patients in ctDNA aneuploidy-low Group 2 also have worse OS in trials of drugs that are independent of AR signaling.

The 200 cfDNA samples in ctDNA aneuploidy-high Group 1 harbored the majority of the non-*AR* genomic alterations that were detected by the *AR*-ctDETECT assay. This included frequent mutations and/or copy number losses in tumor suppressors *TPS3*, *PTEN*, and *RB1*, and copy number gains in *MYC* and/or *MYCN*. Conversely, samples in ctDNA aneuploidy-low Group 2 displayed lower rates of these alterations, which is likely due to the lower sensitivity for detecting mutations and/or copy number losses in samples with low ctDNA content. Many of the alterations in these genes are known to promote mCRPC resistance to therapies that inhibit AR and are negative prognostic biomarkers when detected in tumor tissue and/or cfDNA[2,18,22,26]. Therefore, there may be utility of incorporating specific gene alterations into prognostic models that include clinical factors and ctDNA fractions. This is further supported by our finding that *AR*-GSRs, especially those that are predicted to truncate the AR LBD and/or arise through *AR* ecDNA amplification, are associated with increased risk of radiographic progression and death.

A limitation of this study is the use of a research grade *AR*-ctDETECT assay for targeted DNA-seq of clinical specimens. Accordingly, associations reported in this study are exploratory and would require extensive validation and further development before incorporating into clinical use. Another limitation is that out of the 1311 patients treated in the phase 3 A031201 trial, plasma specimens were only available from 987 of these patients and our study was only approved to analyze 790 of them. This is because analysis of the remaining 197 plasma specimens would have exhausted them in the biorepository. Additionally, the low volumes of plasma available for analysis led to low yields of cfDNA in many samples, reducing the sensitivity for detecting ctDNA-specific alterations. These technical limitations likely contributed to false negatives among the patients categorized as ctDNA-negative. A final limitation was that targeted DNA-seq was performed on cfDNA without patient-matched germline controls, blood, or tumor tissue. This limits the ability to accurately distinguish between somatic or germline alterations in target genes, or between tumor-associated mutations and clonal hematopoiesis. To address this, we only considered high confidence mutations with well-annotated pathogenicity (from OncoKB[57]). This may have resulted in under-reporting of mutations in some genes targeted by the *AR*-ctDETECT assay and also restricts our ability to determine tumor mutational burden. Accurately detecting tumor-associated mutations in cfDNA and measuring their corresponding VAFs is important for determining ctDNA fractions in cfDNA[17,28]. Due to the lower sensitivity for detecting these mutations in our study, we calculated a ctDNA aneuploidy fraction for all samples using the ichorCNA algorithm. Although this ctDNA aneuploidy fraction calculation was prognostic for rPFS and OS, and correlated with other inputs used to calculate ctDNA fraction, it is important to note that these metrics are distinct and should not be directly compared.

In summary, analysis of this phase 3 study has demonstrated the prognostic utility of a mCRPC-specific *AR*-ctDETECT cfDNA-seq assay that leverages detection of tumor-derived copy gain and structural alterations in *AR*, as well as copy gain in *MYC* and/or *MYCN* to build on prior studies that have relied on detection of tumor-derived aneuploidy and/or somatic mutations to identify ctDNA. Detection of ctDNA in general and specific genomic alterations in particular may improve on established clinical factors for developing composite prognostic biomarkers.

## Methods
### Ethics statement
This study for analysis of patient blood plasma specimens was approved by the University of Minnesota Institutional Review Board on

09/29/2020 (STUDY00010929), the Duke University Institutional Review Board on 09/23/2020 (Pro00106740), and the National Cancer Institute (NCI) Clinical Trials Network (NCTN) for correlative science proposal CSC0159 on June 16, 2021.

### Patients and plasma samples
Alliance A031201 (Clinicaltrials.gov: NCT01949337) was a randomized phase 3 clinical trial conducted by the National Cancer Institute (NCI)-funded NCTN Alliance for Clinical Trials in Oncology to compare the anticancer effects of first-line enzalutamide with or without abiraterone acetate and prednisone in men with mCRPC[31]. Men with previously untreated mCRPC and progressive metastatic disease despite ongoing androgen deprivation therapy (ADT) were included. Prior docetaxel or first generation AR inhibitors were permitted in earlier settings. Patients were treated with standard of care doses of enzalutamide 160 mg/d with or without abiraterone acetate and prednisone (1:1 randomization, open label) until clinical or radiographic progression and patients were followed long-term for mortality. Details of the patients, inclusion criteria, study endpoints, and specimens collected for the correlative analysis have been published[31]. Approval was received by the NCTN for correlative science proposal CSC0159 on June 16, 2021, enabling 2 mL of banked pre-treatment EDTA plasma specimens collected from patients at baseline to be shipped on dry ice from the Alliance biorepository at Ohio State University to the University of Minnesota, only for those patients where at least 2 mL of plasma was available. This ensured that banked plasma specimens from A031201 would not be exhausted by this study. This study was approved by the University of Minnesota Institutional Review Board on 09/29/2020 (STUDY00010929) and the Duke University Institutional Review Board on 09/23/2020 (Pro00106740).

### Cell-free DNA (cfDNA) isolation and yield
Cell-free DNA (cfDNA) was isolated from 1 to 3 mL (mean = 2.12 mL, standard deviation = 0.49 mL) of input plasma using a Circulating Nucleic Acid kit (Qiagen catalog 55114) according to the manufacturer's recommendation. cfDNA samples were eluted in 30–50 µL of elution buffer. Concentration was determined from 3 µL input cfDNA using a Qubit 1X dsDNA HS Assay kit (Thermo Fisher catalog Q33231) and a Qubit Fluorometer (Thermos Fisher catalog Q33216). cfDNA yield was calculated by dividing the total mass of cfDNA eluted by the volume of plasma used as input.

### Targeted DNA-sequencing of cfDNA
DNA-seq library preparation and sequencing was performed by the University of Minnesota Genomics Center (UMGC). Concentrations of cfDNA samples were determined using Quant-iT PicoGreen dsDNA Assay (ThermoFisher catalog P7589) per manufacturer's recommendations. An initial 38 cfDNA samples (UMGC project ID: Dehm_Project_070) with masses of at least 1 ng were prepared as DNA-seq libraries using a ThruPLEX Tag-Seq kit (Takara, Catalog R400586) according to manufacturer's recommendations. A subsequent 95 cfDNA samples (UMGC project ID: Dehm_Project_076) with masses of at least 5 ng were prepared as DNA-seq libraries using a ThruPLEX Tag-Seq HV kit (Takara Catalog R400743) according to manufacturer's recommendations. All remaining cfDNA samples with masses of at least 1 ng were prepared as DNA-seq libraries using a ThruPLEX Plasma-Seq kit (Takara, Catalog R400681) with Takara Bio DNA unique dual index kits tubes A (Takara, Catalog R400665), B (Takara, Catalog R400666), C (Takara, Catalog R400667), and D (Takara, Catalog R400668) according to manufacturer's recommendations. Libraries were cleaned using 1X Ampure XP beads (Beckman Coulter, Catalog A63880) followed by DNA quantification. Targeted DNA-seq was performed using SureSelect (Agilent Technologies) capture protocol (Version B.3 June 2015) with minor modifications. Library input was between 750 ng and 1 µg. For libraries with masses below this range, a

second amplification step was performed using KAPA HiFi HotStart Ready Mix (Roche), following manufactured protocols and a second 1X Ampure XP bead clean-up to increase concentrations. Libraries were hybridized overnight (between 16 and 24 h) using a SureSelect XT Reagent Kit (Agilent, Catalog 930672) and SureSelect XT Custom Baits (Agilent, Catalog 5191–6908) described in Supplementary Data 1. Post hybridization libraries were amplified by PCR using KAPA HiFi HotStart Ready mix and library amplified primer mix (Roche, Catalog KK2621). Post capture, libraries were cleaned using 1X Ampure XP beads. Libraries were quantified and pooled in equimolar amounts for sequencing using an Illumina NovaSeq 6000 SP flow cell with 2 × 150 bp settings. Data are available via dbGaP (accession number phs003325.v1.p1).

## DNA-seq read mapping and duplicate removal

Adapters and poor-quality bases were trimmed from paired-end FASTQ files using Trimmomatic (v. 0.39)[58]. An initial pilot experiment with 1.3–30 ng of cfDNA extracted from 38 plasma specimens was used for library generation using unique molecular index (UMI) sequences (Supplementary Fig. 9A). These 38 samples were processed with umi-tools (v. 1.1.2)[59] to identify, correct, and remove UMIs from the reads and place them into the BAM RX tag. Reads were mapped against the human reference genome (GRCh37/hg19) using bwa mem (v. 0.7.17) (http://arxiv.org/abs/1303.3997) and samblaster (v. 0.1.24) was used to limit the number of split alignments for a read (max=2) and limit the number of non-overlapping base pairs between two alignments (min=20). Read duplicate marking and removal was explored using four strategies: Picard tools (v. 2.25.5) (https://broadinstitute.github.io/picard/) MarkDuplicates (defaults), Picard MarkDuplicates with the BARCODE_TAG option set to RX, Picard UmiAwareMarkDuplicatesWithMateCigar with the BARCODE_TAG option set to RX, or collapsing duplicates to a single consensus read via FgBio (v. 1.3.0) functions SetMateInformation, GroupReadsByUmi (using RX tag), CallMolecularConsensusReads, FilterConsensusReads, and remapping deduplicated consensus reads with bwa mem. Use of UMI barcodes for removal of PCR duplicates only improved the number of unique total, mapped, and properly paired DNA-seq reads by an average of 3-4% compared to removal of PCR duplicates using a UMI-independent method (Supplementary Fig. 9B–G). This average improvement in the number of unique DNA-seq reads was mainly attributed to 5 of the 38 pilot cfDNA samples with the highest input mass, indicating that use of UMIs did not enhance the analysis of cfDNA samples with low input masses to a meaningful extent. Based on these results, UMI-independent library preparation and PCR duplicate removal methods were used for analysis of subsequent samples. For these libraries that did not contain UMIs, Picard MarkDuplicates was used for read duplicate removal. Read sequencing, mapping, and capture metrics were collected from samtools (v 1.12) flagstat and Picard tools functions (CollectInsertSizeMetrics, CollectAlignmentSummaryMetrics, CollectHSMetrics, CollectGCBiasMetrics).

## Estimation of ctDNA aneuploidy fraction

The circulating tumor DNA (ctDNA) aneuploidy fraction was estimated by the ichorCNA R package (v.0.2.0)[32] with R (v.4.2.0) using off-target reads as input for ichorCNA[34] (https://github.com/GavinHaLab/ichorCNA_offtarget). Specifically, duplicates-removed bam files were filtered with bedtools (v. 2.3.0) intersect software to remove all reads overlapping the targeted regions, leaving only off-target reads (i.e. approximating low-pass whole genome DNA-seq) for coverage analysis. The number of mapped reads within non-overlapping 1 Mb windows was determined using the readCounter function from HMM Copy Utils software library (commit: 5911bf6, https://github.com/shahcompbio/hmmcopy_utils). Only reads with mapping quality scores >20 were included. These genome-wide binned read count values were used as input to the runIchorCNA.R script. It was executed

using custom initialization parameters for ploidy (2, 3) and normal fraction levels (0.5, 0.6, 0.7, 0.8, 0.9). Otherwise, the script used default settings (including the use of package extra data files: gc_hg19_1000kb.wig, map_hg19_1000kb.wig, GRCh37.p13_centromere_UCSC-gapTable.txt, and HD_ULP_PoN_1Mb_median_normAutosome_mapScoreFiltered_median.rds).

## SNV/indel calling and filtering

For each sample, Freebayes (v.1.3.1) (http://arxiv.org/abs/1207.3907) and GATK MuTect2 (v.4.1.6.0)[60] were used to identify SNV/indel variants using the duplicates-removed BAM file as input. Freebayes was used with default parameters, except the minimum allele frequency was set to 0.01, minimum mapping quality set to 20, and minimum number of alternate allele reads set to 3. MuTect2 was used with default parameters, except a gnomAD[61] germline resource file of known SNPs was used to filter common variants (gnomad.exomes.r2.0.2.sites), the MateOnSameContigOrNoMappedMateReadFilter filtering was disabled, and the minimum allele frequency was set to 0.0000025 for alleles not found in the germline reference file. Variant calling in Freebayes and MuTect2 was restricted to GRCh37/hg19 regions targeted by the capture probes. Output files were processed by the rtg-tools (v.3.11) vcfdecompose function designed to break large indels and multiple-nucleotide polymorphisms into smaller sized SNPs. Variants were normalized with vt software (v.0.57)[62] and rtg-tools vcffilter was used to retain only variants with 6 or more reads containing the alternative allele.

The rtg-tools vcfeval function, with decompose and squash-ploidy parameters enabled, was used to compare filtered Freebayes and MuTect2 VCFs and intersecting calls were retained. Variants were annotated with vcf2maf (v 1.6.21) (DOI: 10.5281/zenodo.1185418), which depends on the Ensembl Variant Effect Predictor (v. 104.3)[63], and OncoKB (v. 3.3.1)[57]. Variants from all intersecting and non-intersecting comparisons were imported into R where each was labeled, filtered, and counted using custom scripts. SNV/indels were considered true positives if they were identified by both Freebayes and MuTect2 and met the following criteria: (1) variant was curated in OncoKB and must be labeled Oncogenic or Likely Oncogenic; or (2) if variant was not curated in OncoKB, it must labeled Oncogenic or Likely Oncogenic, but must also not have a splice-site or splice-region classification in OncoKB. Additional filtering criteria was applied for certain genes based on known germline or non-pathogenic variant status: *PMS2* (remove p.Arg20Gln); *MET* (remove p.Glu168Asp); *KMTC2* (remove p.Tyr816Ter); *ATM* (remove p.His1380Tyr); *FANCA* (remove p.Ser1088Phe and p.Ser858Arg); *NCOR1* (remove p.Arg190Ter); *AURKA* (remove p.Phe31Ile); *FOXA1* (ignore OncoKB filtering above and remove all synonymous mutations, p.Ala83Thr, p.Ser448Asn, and p.Leu148Val). Final filtered variants were classified as likely germline if they affected *HSD3B1* or any gene other than *AR*, *TP53*, *PTEN*, *ERF*, *PIK3CA*, or *CDK12* with a variant allele frequency between 0.45-0.55 or >0.95. Otherwise, filtered variants were classified as likely somatic.

## Identification of gene-specific copy number alterations

Read depths were calculated by mosdepth (v. 0.3.3)[64] using the duplicates-removed BAM file at every position targeted by the capture panel (i.e. gene exons or control regions) with no limit on the number of reads counted. Default parameters were used, except --fast-mode and --flag 1796, which discarded UNMAPPED, SECONDARY, QCFAIL, or DUPLICATE reads from the read depth sum at every position.

Copy number ratios were determined by comparing the normalized read depth from each sample against the average normalized read depth from 65 samples with tumor fraction ≤ 0.05, as determined by ichorCNA. For each gene, the mean read depth was calculated across the concatenated target regions (e.g. all exons). The within-sample normalized read depth was calculated differently for genes found on chrX versus autosomes because one of the five control (C) regions is

located on chrX: C1 (chr9:98258995-98264381), C2 (chrX:16153017-16159789), C3 (chr14:105249980-105255049), C4 (chr15:40514992-40520036) and C5 (chr15:67390102-67395049). For genes on chrX, norm_read_depth = mean read coverage of chrX gene / median of [mean read coverage (C1), mean read coverage (C2) * 2, mean read coverage (C3), mean read coverage (C4), mean read coverage (C5)]. For genes on autosomes: norm_read_depth = mean read coverage of autosome gene / median of [mean read coverage (C1) / 2, mean read coverage (C2), mean read coverage (C3), mean read coverage (C4) / 2, mean read coverage (C5) / 2]. Copy number ratios were calculated as: norm_read_depth for Gene X in sample Y / mean [norm_read_depth for Gene X in samples classified as low tumor fraction]. Genes were classified as copy number gain if the log2(copy number ratio) was > 0.3 and copy number loss if the log2(copy number ratio) was < −0.3 and the sample was classified as ctDNA-positive.

### Identification of *AR* gene structural rearrangements (*AR*-GSRs)

*AR* gene structural rearrangements (*AR*-GSRs) were called using Manta (v.1.5.0)[65], DELLY (v.0.8.1)[66], SvABA (v.1.1.0)[67], and LUMPY (v.0.3.0)[68]. The complete BAM (PCR duplicates removed) was supplied as input to Manta, DELLY, and SvABA; whereas, a discordant reads only BAM and split reads only BAM was supplied as input to LUMPY. Manta was used in tumor-only, exome mode with default parameters, except minEdgeObservations and minCandidateSpanningCount was changed from 3 to 2 reads. The DELLY call function was used with default parameters, except the insert size cutoff, s, was changed from 9 to 15 (median + s * MAD, deletions only). Variants in telomeres, centromeres, and other low complexity regions were excluded via a bed file provided by the software developer. Variants labeled "IMPRECISE" in the INFO column were removed and only variants labeled "PASS" in the FILTER column were retained. SvABA was used in tumor-only mode with default parameters and the unfiltered.sv.vcf was used for downstream analysis. LUMPY-based calling required a histogram of observed library sizes to be generated by the pairend_distro.py script using the complete BAM as input. The breakpoint probability distortion was returned along with the mean and standard deviation of the library, which were supplied to the LUMPY function as parameters.

VCFs from each caller were converted from symbolic to breakend format (see VCF specification 4.2, section 5.4, https://github.com/samtools/hts-specs) using the rtg-tools (v.3.11) svdecompose function. Representing all SVs in breakend format allowed for across-caller VCF comparison. SvABA VCFs only represent SVs in breakend format and do not predict SVTYPE for any calls. For SvABA VCFs, SVTYPE (INV, DEL, DUP, or BND) was predicted using a custom R script that examined SV breakend orientation and mate pair information. For the other callers, the original SVTYPE was preserved in the decomposed breakend format using a custom R script. Any SVs with breakends on different chromosomes were re-labeled as translocations (TRA). The rtg-tools bndeval function, with tolerance parameter set to 1000 bp, was used to compare any two VCFs and intersecting calls were retained. Variants from all intersecting and non-intersecting comparisons were imported into R (v. 4.2.0) where each was labeled, filtered, and counted using custom scripts. SV length and read support values were parsed from data contained within each VCF record. SV length was calculated as the base-pair difference between breakends (TRA variants given length -1). The number of split or discordant reads supporting an SV call was extracted from the following VCF fields, respectively: Manta (FORMAT: SR, PR), DELLY (FORMAT: RV, DV), SvABA (FORMAT: SR, DR), and LUMPY (INFO: SR, PE). SVs were removed from further analysis if either breakend overlapped with a list of known artifactual mapping regions (bed file) discovered by the ENCODE project. *AR*-GSRs were considered true positives if at least one breakend fell within the *AR* gene region, was identified by at least two of four SV callers, and had at least 3 supporting split reads and 3 supporting discordant paired-end reads.

Each AR-GSR was evaluated for whether it was likely to encode an AR variant with a truncated ligand binding domain (LBD). Criteria for inclusion in this list of likely LBD-truncating AR-GSRs were (1) a DEL, INV, or BND that fully or partially overlaps AR exon 4, 5, 6, 7, and/or the start of exon 8 through the stop codon of exon 8, but does not overlap the interval spanning the beginning of AR exon 1 through the end of AR exon 3; or (2) a DUP, INV, or BND that fully overlaps the interval from the start of AR exon 1 through the end of AR exon 3 with a breakpoint anywhere between the end of AR exon 3 and the stop codon in AR exon 8; or (3) a DUP with both start and end breakpoints within the AR gene and flanking one or more of AR exons 3-7; or (4) a TRA with a breakpoint anywhere between the end of AR exon 3 and the stop codon in AR exon 8 that retains the centomeric side of AR encompassing exons 1–3.

Samples with AR-GSRs were classified as likely containing AR extrachromosomal circular DNA (AR ecDNA) if that sample contained at least 2 AR-GSRs and displayed copy number gain of the AR gene body, defined as log2(copy number ratio) > 0.3.

### Genome-wide copy number analysis with CNVkit

Genome wide copy number alterations were inferred using CNVkit software (ver. 0.9.9)[53]. An accessible_regions.bed input file was prepared using the CNVkit *access* function with two parameters: the GRCh37/hg19 reference genome, and the ENCODE project exclude BED file (wgEncodeDacMapabilityConsensusExcludable.bed). This file includes only sequence-accessible regions of the genome to be considered by CNVkit. The targets.bed input file was prepared using the CNVkit *target* function with three parameters: the targeted DNA-seq SureSelect probes BED file, the GRCh37/hg19 Ensembl GTF (release 87) for gene annotation, and the "split" parameter was specified (to divide larger baited regions into regions closer to the default bin size, 267 bp). The antitargets.bed input file was prepared using the CNVkit *antitarget* function with two parameters: the targets.bed and accessible_regions.bed files. For each sample, the CNVkit *coverage* function was used with the inputs described above and the deduplicated BAM file to calculate read coverage levels across the targeted and off-target regions of the genome. This step required the hmmlearn software (ver. 0.2.7). A CNVkit reference was built from the coverage files of 65 samples that had tumor fraction ≤0.05, as determined by ichorCNA. The CNVkit *reference* function was performed with the "haploid-x-reference" and "sample-sex male" options to correctly estimate coverage levels in male samples. The CNVkit *fix* function was used to correct for known biases and calculate log2 copy number ratios. The CNVkit *segment* function was completed using the "drop-low-coverage" option and the "cbs" segmentation algorithm. Finally, the CNVkit *call* function was used to calculate copy number integers from the log2 ratios, with "haploid-x-reference", "sample-sex male", and "drop-low-coverage" parameters. For every sample, the inferred copy number ratios for each segment were plotted as a chromosomal heatmap using R and tidyverse functions. The log2 ratio color scale was winsorized to the range: mean +/− 5 SD (to remove spurious outlier values skewing the signal).

### Definition of ctDNA-positive cfDNA samples

The mean ctDNA aneuploidy fraction estimate of DNA-seq assays (n = 8) performed on control cfDNA isolated from a pool of plasma from healthy male donors (ZenBio, Inc., Catalog #SER-PDP-10) was 0.0698 with a standard deviation of 0.0241. Samples exceeding the mean + 3 SD ( > 0.1421) were classified as ctDNA-positive and belonging to ctDNA aneuploidy-high Group 1 based on the rationale that samples below the mean + 3 SD cutoff are 99.87% likely to lack ctDNA aneuploidy. Accordingly, samples below this mean + 3 SD cutoff were considered as ctDNA positive and belonging to ctDNA aneuploidy-low Group 2 only if they displayed at least one of the following features: a likely somatic SNV in any gene targeted by DNA-seq; *AR* log2(copy number ratio) of > 0.3; *AR* enhancer log2(copy number ratio) of >0.3;

positive for an *AR*-GSR; *MYC* log2(copy number ratio) of >0.3; *MYCN* log2(copy number ratio) of >0.3. Samples not classified in ctDNA aneuploidy-high Group 1 or ctDNA aneuploidy-low Group 2 were placed in ctDNA-negative Group 3.

## Statistics and reproducibility

Investigators at Duke University performed the statistical analyses to evaluate the prospective hypotheses that detection of *AR*-GSRs in ctDNA would be associated with worse OS, and that ctDNA positive patients would have adverse baseline clinical features, worse radiographic rPFS, and worse OS. Investigators at the University of Minnesota performed cfDNA-sequencing and data analysis to identify ctDNA-positive patient groups, and were blinded to clinical data. The comparisons were primarily focused on three groups: ctDNA-positive patients in ctDNA aneuploidy-high Group 1, ctDNA-positive patients in ctDNA aneuploidy-low Group 2, and ctDNA-negative Group 3. We focused our analysis to *AR*-GSRs as well as these ctDNA-positive and ctDNA-negative groups and minimized the testing of clinical correlations with other genomic features in ctDNA because this study is part of larger project to develop a clinical-genomic and predictive model in mCRPC patients, which will be reported separately.

First, disease characteristics were compared between patients in ctDNA aneuploidy-high Group 1 and ctDNA aneuploidy-low Group 2, relative to those in ctDNA-negative Group 3. Specifically, the levels of prostate-specific antigen (PSA), hemoglobin, alkaline phosphatase, lactate dehydrogenase, as well as the frequencies of bone and liver metastasis were compared. This analysis aimed to determine any discernible differences in disease characteristics among the groups. The association was determined between the ctDNA classification and the risk factor, which is based on a prognostic model of OS[14,31,54]. The Kaplan-Meier product-limit approach was used to estimate the rPFS and OS distributions by the different ctDNA classifications. The log-rank test was used to evaluate the statistical significance of differences between the two groups. The proportional hazards model was used to test for the prognostic significance in the groups predicting OS/rPFS adjusting for the ctDNA fraction. P-values are adjusted for multiplicity using the Benjamini-Hochberg method.

## Reporting summary

Further information on research design is available in the Nature Portfolio Reporting Summary linked to this article.

# Data availability

The raw DNA-seq data for this study are available in dbGaP under accession: phs003325.v1.p1 (https://www.ncbi.nlm.nih.gov/gap/). Processed DNA-seq data are in the source data file with the tab names of 'Supplementary Data file 2–6'. Additional information and results related to the phase 3 A031201 trial can be found at clinicaltrials.gov under accession: NCT01949337 (https://www.clinicaltrials.gov/). The A031201 clinical trial protocol has been published previously[31]. The A031201 clinical data are available under restricted access from the NCI NCTN Data Archive (https://nctn-data-22archive.nci.nih.gov/); access can be obtained by completing a NCI/NCTN data request form and a data use agreement. All other data are available in the main text, supplementary information or source data file. Source data are provided with this paper.

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

## Acknowledgements

Research reported in this publication was supported by the National Cancer Institute of the National Institutes of Health under Award Numbers U10CA180821 and U10CA180882 (to the Alliance for Clinical Trials in Oncology), UG1CA233180, UG1CA233253, UG1CA233290, R01CA256157 (to AJA, SH, and SMD), R01CA174777 (to SMD). This work was also supported in part by a 2018 John Black Charitable Foundation-Prostate Cancer Foundation Challenge Award (to RH and LG) and by Astellas and Pfizer. ESA is partially supported by NCI grant P30 CA077598. The content is solely the responsibility of the authors and does not necessarily represent the official views of the National Institutes of Health.

## Author contributions

A.J.A., S.H., and S.M.D. conceived the study. T.P.K., A.K., J.L., S.A.M., Y.L., and S.M.D. designed the cfDNA-sequencing assay and bioinformatics pipeline. T.P.K., B.L., J.L., S.G., E.S.A., A.J.A., S.H., and S.M.D. analyzed and interpreted data. M.J.M., H.B., C.J.R., and A.J.A. guided the access to A031201 plasma specimens. T.P.K. and S.M.D. prepared the manuscript. T.P.K., B.L., A.K., J.L., S.G., S.A.M., Y.L., R.H., L.G., M.J.M., H.B., C.J.R., E.S.A., A.J.A., S.H., and S.M.D. edited the manuscript.

## Competing interests

MJM has served as a consultant to Lantheus, AstraZeneca, Daiichi, Convergent Therapeutics, Pfizer, ITM Isotopes, Clarity Pharmaceuticals, Blue Earth Diagnostics, POINT Biopharma, Telix, Z-Alpha, AMBRX, Flare Therapeutics, Fusion Pharmaceuticals, Curium, Transtherabio, Celgene, Arvinas, and Exelixis. His institution receives royalty payments from Telix, and research funding from Novartis, Fusion, and Astellas. HB has served as consultant/advisory board member for Janssen, Astellas, Merck, Pfizer, Foundation Medicine, Blue Earth Diagnostics, Amgen, Bayer, Oncorus, LOXO, Daicchi Sankyo, Sanofi, Curie Therapeutics, Astra Zeneca, Novartis, and has received research funding (institution) from Janssen, AbbVie/Stemcentrx, Eli Lilly, Astellas, Millennium, Bristol Myers Squibb, Circle Pharma, Daicchi Sankyo, Novartis. CJR has served as a consultant to Oric, Pfizer, Bayer, and Sanofi. ESA is a paid consultant/advisor to Janssen, Astellas, Sanofi, Dendreon, Pfizer, Amgen, Eli Lilly, Bayer, AstraZeneca, Bristol Myers Squibb, ESSA, Clovis, Merck, Curium, Blue Earth Diagnostics, Foundation Medicine, Exact Sciences and Invitae; has received research funding to his institution from Janssen, Johnson & Johnson, Sanofi, Dendreon, Genentech, Novartis, Tokai, Bristol Myers Squibb, Constellation, Bayer, AstraZeneca, Clovis and Merck; and is the coinventor of a patented AR-V7 biomarker technology that has been licensed to Qiagen. AJA reports research support (to Duke) from the NIH/NCI, PCF/Movember, DOD, Astellas, Pfizer, Bayer, Janssen, Dendreon, BMS, AstraZeneca, Merck, Forma, Celgene, Amgen, Novartis.

AJA reports consulting or advising relationships with Astellas, Pfizer, Bayer, Janssen, BMS, AstraZeneca, Merck, Forma, Celgene, Myovant, Exelixis, GoodRx, Novartis, Medscape, MJH, Z Alpha, Telix. SH is a member of the Data Monitoring Committees (DMCs) for Aveo, Beigene, BMS, CG Oncology, J&J, Sanofi and was funded by grant awarded to Duke University by ASCO and Astellas. SMD has served as a paid consultant/advisor to Janssen, Bristol Myers Squibb, and Oncternal Therapeutics, and has served as principal investigator on grants awarded to the University of Minnesota by Janssen and Pfizer/Astellas. The remaining authors declare no competing interests.
