## [Peer Review File · Nature Communications]

REVIEWER COMMENTS

Reviewer #1 (CT, biostatistician):

Todd P. Knutson and colleagues analyzed specimens from the phase 3 A031201 trial and quantified ctDNA & AR-specific genomic alterations from cfDNA-seq assay. They grouped patients into 3 categories based on ctDNA threshold, somatic pathogenic mutation, AR-GSR, and/or copy number gain. In the current report, they described the methods and utility of their cfDNA-seq assay, and also interrogated its prognostic ability. The introduction section is lucid and well written. I provide general comments from a biostatistics (not bioinformatics) review of their work.

Comments:

1) Last sentence of the first paragraph of the Introduction section: The latter part of this sentence "...identify CRPC patients unlikely to benefit from treatment" is a feature of a predictive marker, not a prognostic marker.

2) Last paragraph of the Introduction section: provide manuscript reference.

3) What is the rationale for using mean+3SD as cutoff for ctDNA positive? Were other data-dependent thresholds assessed and could these have provided better discrimination between the 3 ctDNA groups, especially between Group 1 and Group 2?

4) Last sentence of second paragraph of the Introduction section: Authors argued that, and I quote "...there is unmet need for cfDNA assays able to prognosticate outcomes in mCRPC patients, particularly in the context of AR-targeted therapies." Comment on (or provide data showing) the novel classification of ctDNA status in the context of AR-specific mutations as done here being a significant improvement over the current method of ctDNA determination which is based on detection of aneuploidy and presence of mutations. The utility of the current work over current ctDNA assessment methods (based on aneuploidy and mutations) needs to be demonstrated before (or in conjunction with) the planned clinical-genomic and predictive model analysis.

5) Second before last paragraph of the Discussion section: I quote: "Another limitation is that our study was only approved to analyze 60% of the plasma specimens collected from patients treated in the phase 3 A031201 trial, because analysis of the remaining 40% would have exhausted them in the biorepository." Is this statement true? From the CONSORT diagram, you could have really only

analyzed 987 of 1311 because 252 and 72 did not consent and had no banked samples respectively. So, the percentage should be 80% (790/987).

Reviewer #2 (Prognostic marker, bioinformatics):

In this article, Knutson, Luo and colleagues report the development of an academic circulating tumor DNA (ctDNA) assay based on targeted cell-free DNA (cfDNA) sequencing for metastatic castration resistant prostate cancer (mCRPC). Intense computation and biostatistics are accurately reported in the methodology, including: (i) the estimation of ctDNA fraction from cfDNA calibrated on a pooled plasma from healthy donors; (ii) SNV/indel calling without patient-matched germline controls; (iii) CNA and rearrangement identification of androgen receptor and MYC/MYCN genes. The assays was applied to a large cohort of 776 patients from the Alliance A031201 randomized phase 3 trial receiving enzalutamide with or without abiraterone acetate and prednisone, using low input volumes of plasma (1-3 mL), and was proven capable of prognostication for radiographic progression-free survival (rPFS) and overall survival (OS).

From a clinical standpoint, the prognostic value of ctDNA detection in mCRPC is already well known, although mostly evaluated in later-line settings (Kohli et al, eBioMed 2020; Goodall et al, JCO 2020; Tukachinsky et al, CCR 2021).

From a technical standpoint, in the present work the authors report the adoption of specific computational techniques and analyze sequence mCRPC-distinct alterations such as AR alterations (including SNV, indels, CNA and structural rearrangements), AR enhancer amplifications, and MYC/MYCN gains. They demonstrate an association with worse prognosis that relies on alterations beyond the fraction of ctDNA, particularly relying on AR alterations. However, there are some technical concerns that should be addressed (see Major comments).

In summary, this work introduces the concept of a mCRPC-specific liquid-biopsy test that could overcome certain limitations of commercial assays, despite its applicability is more likely to be exploited in a research context rather than in the clinical setting. Indeed, ctDNA prognostication is not sufficient by itself to drive patient-tailored strategies in future clinical trials. Besides, there are technical points that need revision.

Major comments

1. The way the authors present group distinction into “ctDNA high” (Group 1) vs “ctDNA low harboring specific alterations” (Group 2) is not clinically relevant and frankly confounding. These two groups are in fact technical spectra within a broader “ctDNA positive” classification, based on detection of high aneuploidy (Group 1) or pathogenic genetic findings without high aneuploidy (Group 2). Both are distinct from “ctDNA negative” results (Group 3). Despite focusing throughout the dissertation on the difference between Group 1 and Group 2 from a technical standpoint, their individual impact on prognosis is limited as highlighted also in the abstract (“ctDNA high and ctDNA-low patients had similar rPFS and OS”, that was worse than “ctDNA negative” patients). Indeed, rPFS and OS of Group 1 and 2 are overlapping, and statistical tests are based on the difference with Group 1 vs Group 3 and Group 2 vs Group 3 (Figure 6). Therefore, from an utilitarian perspective targeting a broader audience also including physicians, I recommend to shift the focus on the prognostic value of “ctDNA positive” vs “ctDNA negative”, moving in the background the discussion on “ctDNA high” vs “ctDNA low”. In the results I suggest to: (1) first present how “ctDNA positive” samples were defined, therefore as “ctDNA high” vs “ctDNA low”; (2) then, when addressing prognostication, make the comparison of “ctDNA positive” vs “ctDNA negative” as the central point, then proving that worse prognosis was independent on how “ctDNA positive” samples were defined (Group 1 vs Group 2). The same considerations apply to the abstract. In this regard, also inverting Figure 6C and 6D with 6A and 6B would be more straightforward.

2. The rPFS/OS multivariable analysis is limited to the ctDNA fraction as the only additional variable, tuning down the conclusions of the study regarding the prognostic value of the liquid biopsy test. The claim that “there were many patients in the low- and intermediate-risk (Halabi) groups that were in ctDNA-high Group 1 and ctDNA-low Group 2” would be better supported by including these clinical-pathological features in the multivariable analysis. It would be beneficial for the relevance of the paper to first perform univariate analysis of these clinical-pathological features for rPFS/OS, and then include statistically significant findings in the multivariable analysis to prove that the prognostic value of liquid biopsy is maintained and independent from already known prognosticators.

3. In the manuscript, it is unclear whether there were ctDNA positive samples supported only by aneuploidy without any genetic alteration. Indeed, mutations and other genetic alterations are scatterly reported in Figure 2 to Figure 5. A clear summary figure showing overlapping between Group 1 and Group 2 is needed as the main figures, perhaps adapted from Supplementary S4. From this latter figure, it appears that all Group 1 samples would have been also identified by SNV/indel/CNA/fusion calling and this is relevant also to resize the importance of Group 1 vs Group 2.

4. The application of ichorCNA to off-targeted reads needs validation. Since the authors lack low-pass WGS data for which ichorCNA was designed, a proper validation of the methodology or, otherwise, a reference of previous works that validated it is necessary. Indeed, discarding targeted reads and leveraging only on off-targeted DNA is not a standard procedure with ichorCNA. I recommend to check whether any statistical difference is present in VAF-based ctDNA estimations between Group 1 and Group 2. This would help to validate the results. Besides, it is concerning that the lowest ctDNA estimate for healthy individuals (4.7%) is a log₁₀ higher compared to the minimum amount of ctDNA in cancer patients (0.4%). May the authors provide some explanation or comment on this?

5. The targeted-DNA panel of genes is unreported in the manuscript. Despite several mutations and other gene alterations are reported in Figures, a clear disclosure of the genes populating the sequenced panel is mandatory.

Minor comments

1. Figure 1 caption is missing.
2. The volume of available plasma per patient should be reported as average and standard deviation. Did the amount of plasma influence the results, i.e., less plasma is correlated with ctDNA negativity?
3. The “ctDNA-positive patients have poor clinical outcomes” paragraph is a bit redundant, i.e., the HRs for rPFS are reported twice.

Reviewer #3 (Prostate cancer genomics):

General comment: The manuscript is well written with high quality images and interesting results with regards to AR GSRs. However, two major concerns were noted, one with regards to the technical aspects of the manuscript and one with regards to novelty.

Technical aspects: The methodology applied in the manuscript (both lab and informatics) is of high quality. However, it is noteworthy is that ichorCNA, which was developed for low-pass whole genome sequencing data, is applied on the panel data to estimate the cancer DNA fraction. It is apparent from the data presented on healthy donors that this approach has a problem. In the healthy donors the ctDNA fraction was up to 12.2%. Considering that the median ctDNA fraction on ~4800 cfDNA prostate samples reported by Foundation Medicine (PMID: 36265119) was ~3% then an assay-specific threshold of 14.2% ctDNA is not good enough.

It is unclear whilst reading the manuscript why ichorCNA was applied and not other copy-number tools developed for panel data? The coverage is in general very low, due to the limited amount of plasma. Did this lead to even more noise when applying panel-based copy-number tools?

The poor ctDNA fraction estimation is likely affecting other aspects of the manuscript as it applied both in the multivariate analysis for AR GSRs and also to stratify the cohort into the three groups for prognostication. Prognostication using ctDNA fraction has been reported multiple times previously

with a much more distinct separation of the groups which highlights the problem with using ichorCNA (see PMID 33836265, 29367197, 38418825).

Novelty aspects: Many groups have presented ctDNA of advanced prostate cancer which includes in-depth interrogation of the androgen receptor and the common drivers of prostate cancer and also prognostication based on ctDNA. The presented manuscript does therefore not harbor novelty in relation to the currently published literature. The analysis of AR GSRs is potentially of interest, but it is affected by the poor cancer DNA fraction estimation.

Reviewer #4 (AR, prostate cancer diagnosis):

In this manuscript, the authors developed a targeted cell-free DNA (cfDNA) sequencing assay to evaluate 69 genes (including AR) frequently altered in metastatic castration-resistant prostate cancer (mCRPC), and they applied the assay to 776 banked EDTA baseline plasma samples (1-3 mL) from the Alliance A031201 phase 3 trial, which randomized men with mCRPC to treatment with first-line enzalutamide with or without abiraterone. The patients were stratified into 3 groups based on the level of ctDNA fraction and the presence of known pathogenic mutations, AR structural variant, or copy gain in AR, MYC, or MYCN. ctDNA-high patients had the worst rPFS and OS, followed by ctDNA-low patients with detectable gene alterations, compared to ctDNA-negative patients which had the most favorable prognosis.

Although the finding that ctDNA positive patients have worse prognosis than ctDNA negative mCRPC patients is not novel, the manuscript identifies an additional group of poor prognosis patients that have detectable molecular alterations in their cfDNA despite a low ctDNA fraction. This work has many strengths including large-scale ctDNA analysis of prospectively collected plasma specimens from a large, relatively uniform clinical cohort of mCRPC patients in the setting of a phase 3 clinical trial, and the use of a unique targeted cfDNA sequencing assay capable of comprehensive profiling of AR alterations including mutations, amplification, enhancer amplification, gene structural arrangements, and gains in MYC and MYCN. Detailed descriptions of cfDNA sequencing and bioinformatic analysis methods are provided. It is also a strength that the cfDNA sequencing and bioinformatics team was blinded to the clinical data, while a separate team at another institution performed the statistical analyses with the clinical data. Together, the manuscript provides a large body of genomic ctDNA data that may serve as a useful resource for the prostate cancer field.

The manuscript also has several important limitations which the authors appropriately discuss, including the use of a research-grade (rather than clinical-grade) ctDNA assay, the lack of germline controls, and the lack of analyses of matched tumor tissues. Several additional comments and suggestions are as follows:

Major comments:

- The finding that control plasma samples from a pool of healthy male donors had estimates of ctDNA fraction of 4.7% to 12.2% (Fig 1D) is rather odd. Can the authors provide more details regarding these “healthy” donors? What are their ages? How do the authors know that they do not harbor undiagnosed malignancies?
- In Figs. 2G and 2H, the authors have appropriately corrected for ctDNA fraction in their analyses of hazard ratios for rPFS and OS based on the presence of AR-GSR and AR ecDNA. Did the authors also correct for Halabi Risk score and/or other clinical factors in this analysis? In addition, it would be interesting to test for interaction by treatment group (enzalutamide vs. enzalutamide + abiraterone).
- It is odd that AR alterations would occur at a higher frequency in ctDNA-low Group 2 compared to ctDNA-high Group 1. Can the authors comment on this?
- Figure 4. The identification of TP53 and RB1 loss in the ctDNA of a subset of patients is interesting. Did these patients harbor any evidence of neuroendocrine differentiation based on greater incidence of visceral metastases, lower than expected serum PSA, or other characteristic features? Did these patients have worse clinical outcomes after treatment with the AR-targeting therapies enzalutamide or enzalutamide + abiraterone, as might be expected for patients with neuroendocrine prostate cancer?
- Page 14-15 and Suppl Table S2. The authors state in the text that patients in ctDNA-high Group 1 had higher PSA, lower HGB, higher Alk Phos, Higher LDH, higher frequency of bone and liver metastases, and higher Halabi Risk score compared to ctDNA-negative Group 3. Similarly, ctDNA-low Group 2 also had differences compared to Group 3. Please list the p-values for the differences in clinical features between Groups 1 and 3 and Groups 2 and 3 in Suppl Table S2.
- Suppl Table S2 seems to show that Groups 1 and 2 had a greater % of black patients compared to Group 3. Is this a statistically significant difference? If so, the authors should speculate in the Discussion on why there might be differences in ctDNA detection between races. Along those lines, were there differences in ctDNA gene alterations based on race?
- Figure 6. Because the Alliance A031201 trial did show a significant improvement in rPFS with the addition of abiraterone to enzalutamide, it would be interesting to test for any interaction by treatment group in the rPFS analyses (Fig 6A and 6C).

Minor comments:

- Figure legend for Fig. 1 is missing.

- Please make clear in the Methods section under “Patients and plasma samples” that these are baseline pre-treatment plasma specimens that were analyzed.
- Please label the Supplementary Data files. The version of these files downloaded from the submission site did not have informative file names.
- Page 13, “Leveraging detection of AR alterations...” There seems to be a typo with an extra “and” after the word “gains.”
- If the p-values were adjusted using the Benjamini-Hochberg method, the false discovery rate for this correction should be stated.

We thank the Reviewers for their fair and thorough evaluation of our manuscript. Our response to each reviewer comment is indicated in blue font.

Response to Reviewer #1 (CT, biostatistician):

Todd P. Knutson and colleagues analyzed specimens from the phase 3 A031201 trial and quantified ctDNA & AR-specific genomic alterations from cfDNA-seq assay. They grouped patients into 3 categories based on ctDNA threshold, somatic pathogenic mutation, AR-GSR, and/or copy number gain. In the current report, they described the methods and utility of their cfDNA-seq assay, and also interrogated its prognostic ability. The introduction section is lucid and well written. I provide general comments from a biostatistics (not bioinformatics) review of their work.

Response: We thank the reviewer for this summary and for the positive comment about our introduction section being lucid and well written.

1) Last sentence of the first paragraph of the Introduction section: The latter part of this sentence "...identify CRPC patients unlikely to benefit from treatment" is a feature of a predictive marker, not a prognostic marker.

Response: We agree with the reviewer, and **we deleted this part of the sentence in the Introduction**. The revised sentence now reads as follows:

"Optimizing treatment regimens with these and other AR-independent agents, as well as designing efficient clinical trials to test emerging new therapies, may be aided by prognostic biomarkers that improve on currently available clinical prognostic models in mCRPC patients^{14, 15}".

2) Last paragraph of the Introduction section: provide manuscript reference.

Response: This last paragraph was a summary of the current study (not a previous publication). To clarify the intent for this paragraph and also to adhere to Nature Communications formatting requirements **we edited the last paragraph of the Introduction:**

"In this work, we detect mCRPC-specific genomic alterations at high frequency in cfDNA specimens classified as having high ctDNA aneuploidy. These patients have a short duration of radiographic progression-free survival (rPFS) and overall survival (OS) compared to ctDNA-negative patients. In patients lacking detectable ctDNA aneuploidy, the presence of AR-GSRs, known pathogenic mutations, and/or copy gains in AR, MYC or MYCN can identify ctDNA-positivity that similarly predicts a shorter rPFS and OS relative to ctDNA-negative patients. These data validate the importance of evaluating these genomic alterations in patients having low ctDNA aneuploidy in a phase 3 trial context, as well as the prognostic utility of detecting ctDNA in mCRPC patients being treated with contemporary AR-targeted therapies."

3) What is the rationale for using mean+3SD as cutoff for ctDNA positive? Were other data-dependent thresholds assessed and could these have provided better discrimination between the 3 ctDNA groups, especially between Group 1 and Group 2?

Response: In the current study, we did not adjust thresholds for detecting ctDNA fraction or other genomic features iteratively to understand how these changes would affect prognostication of clinical outcomes in Figure 6. The hypotheses of the study, data analysis methods, and cutoffs were pre-defined at the beginning of the study to ensure rigor in testing the cfDNA-sequencing assay findings against clinical outcomes. The rationale for using mean + 3 standard deviations as a cutoff is that this is a well-known approach to identify outliers based on the normal distribution. By definition, all values in a normal distribution are nearly certain (99.73% certain) to fall within the mean + 3 SD. Thus, our approach is a conservative way to ensure samples identified as ctDNA aneuploidy-high are true outliers relative to a normal distribution. For instance, if our set of 776 samples all lacked ctDNA and we used a cutoff of 3 standard deviations, only 0.13% of the samples would lie above the normal distribution by chance (meaning this approach would be predicted to only mis-classify one sample as ctDNA-positive). However, if we used a cutoff of 2 standard deviations, 2.27% of the 776 samples could lie outside of the normal distribution by chance (resulting in 17 mis-classified samples).

Based on comments from the other reviewers, **we updated the terminology in our paper** to more accurately reflect that the modified ichorCNA method we used to calculate “ctDNA fraction” is measuring genome-wide aneuploidy. Therefore, we now refer to the “ctDNA aneuploidy fraction” throughout the paper.

To address this question directly, **we edited the Methods section under the “Definition of ctDNA-positive cfDNA samples” heading** to provide more discussion on the rationale for our mean + 3 SD cutoff, and also edited this and other sections of the manuscript to reflect this cutoff distinguishing between high vs. low ctDNA aneuploidy:

“The mean ctDNA aneuploidy fraction estimate of DNA-seq assays (n = 8) performed on control cfDNA isolated from a pool of plasma from healthy male donors (ZenBio, Inc., Catalog #SER-PDP-10) was 0.0698 with a standard deviation of 0.0241. Samples exceeding the mean + 3SD (> 0.1421) were classified as ctDNA-positive and belonging to ctDNA aneuploidy-high Group 1 based on the rationale that samples below the mean + 3SD cutoff are 99.87% likely to lack ctDNA aneuploidy.”

4) Last sentence of second paragraph of the Introduction section: Authors argued that, and I quote “...there is unmet need for cfDNA assays able to prognosticate outcomes in mCRPC patients, particularly in the context of AR-targeted therapies.” Comment on (or provide data showing) the novel classification of ctDNA status in the context of AR-specific mutations as done here being a significant improvement over the current method of ctDNA determination which is based on detection of aneuploidy and presence of mutations. The utility of the current work over current ctDNA assessment methods (based on aneuploidy and mutations) needs to be demonstrated before (or in conjunction with) the planned clinical-genomic and predictive model analysis.

Response: To address this comment, we focused on the 479 patients that lacked detection of aneuploidy or pathogenic somatic mutations (these 479 patients are denoted by gray bars at the top of Figures 2A and 3A). In these 479 patients, those lacking detection of AR, MYC, or MYCN alterations (n = 320) had superior rPFS and OS compared with those patients where these

alterations were detected (n = 159). These data are included in the revised manuscript as **new Supplementary Figure S4**, and strongly support the utility of the current work over current ctDNA assessment methods based on aneuploidy and/or mutations only. **We edited the text in the last paragraph of the “AR alterations and MYC/MYCN gains identify ctDNA-positive patients” section of the Results** to reflect these new data as follows:

“we evaluated the clinical significance of incorporating AR, MYC or MYCN alterations into ctDNA detection. Across the 479 patients that were ctDNA aneuploidy-low and lacked a pathogenic mutation, OS and rPFS were longer in the 320 patients that lacked detectable alterations in AR, MYC or MYCN compared with the 159 patients where these alterations were detected (Supplementary Figure S4A&B).”

5) Second before last paragraph of the Discussion section: I quote: "Another limitation is that our study was only approved to analyze 60% of the plasma specimens collected from patients treated in the phase 3 A031201 trial, because analysis of the remaining 40% would have exhausted them in the biorepository." Is this statement true? From the CONSORT diagram, you could have really only analyzed 987 of 1311 because 252 and 72 did not consent and had no banked samples respectively. So, the percentage should be 80% (790/987).

Response: We agree with the reviewer. To address this comment, **we edited this section of the Discussion** as follows:

“Another limitation is that out of the 1,311 patients treated in the phase 3 A031201 trial, plasma specimens were only available from 987 of these patients and our study was only approved to analyze 790 of them. This is because analysis of the remaining 197 plasma specimens would have exhausted them in the biorepository.”

Response to Reviewer #2 (Prognostic marker, bioinformatics):

In this article, Knutson, Luo and colleagues report the development of an academic circulating tumor DNA (ctDNA) assay based on targeted cell-free DNA (cfDNA) sequencing for metastatic castration resistant prostate cancer (mCRPC). Intense computation and biostatistics are accurately reported in the methodology, including: (i) the estimation of ctDNA fraction from cfDNA calibrated on a pooled plasma from healthy donors; (ii) SNV/indel calling without patient-matched germline controls; (iii) CNA and rearrangement identification of androgen receptor and MYC/MYCN genes. The assays was applied to a large cohort of 776 patients from the Alliance A031201 randomized phase 3 trial receiving enzalutamide with or without abiraterone acetate and prednisone, using low input volumes of plasma (1-3 mL), and was proven capable of prognostication for radiographic progression-free survival (rPFS) and overall survival (OS).

Response: We thank the reviewer for this summary and for the positive comment about the intense computation and biostatistics being accurately reported in the methodology.

From a clinical standpoint, the prognostic value of ctDNA detection in mCRPC is already well known, although mostly evaluated in later-line settings (Kohli et al, eBioMed 2020; Goodall et al, JCO 2020; Tukachinsky et al, CCR 2021).

Response: We agree with the reviewer and emphasize that the Kohli 2020 and Tukachinsky 2021 references were properly cited and discussed in our manuscript along with several other previous studies investigating ctDNA in the mCRPC setting. We are unaware of the Goodall JCO 2020 reference, and were unable to locate it on PubMed (although we did find a Sumanasuriya et al., Eur Urol 2021 reference related to ctDNA in mCRPC with Jane Goodall as 10th author; PMID: 34103179)

We would like to emphasize three key differences between the indicated publications and our study:

- 1) The raw DNA-seq data from our study are publicly available in dbGaP for other groups to validate and reproduce our findings, as well as to develop/test new bioinformatics tools and analytical approaches. The Kohli 2020 and Tukachinsky 2021 studies are based on commercial DNA-seq assays (Predicine and Foundation, respectively) for which the raw data are unavailable. The Sumanasuriya 2021 study is based on low-pass whole genome sequencing funded by Sanofi, for which the raw data are unavailable. Thus, although the prognostic value of ctDNA detection in mCRPC has been reported in these studies, these conclusions cannot be verified or expanded upon from their data.
- 2) Our data were generated using samples from a large, randomized phase 3 trial. The Tukachinsky 2021 and Sumanasuriya 2021 studies pooled samples from multiple trials and/or used data from samples analyzed outside of formal trials. The Kohli 2020 study analyzed samples from a single institution that were collected outside of a formal trial.
- 3) The commercial assays underpinning the Kohli 2020 and Tukachinsky 2021 studies have minimum assay requirements that are incompatible with the low volume of plasma or mass of cfDNA in many of the A031201 specimens we analyzed in our study (Figure 1C, Supplementary Figure S2, and Supplementary Table S2). For example, in Kohli 2020, the input for the Predicine assay is 5-20 ng cfDNA (approximately 1500-6000 haploid genome equivalents) for library preparation prior to targeted DNA-seq. In Tukachinsky 2021, the minimum assay requirement of the commercial Foundation Medicine assay is 18-20 mL of blood and 20-100 ng input cfDNA (approximately 6000-30,000 haploid genome equivalents) for library construction prior to targeted DNA-seq. The Sumanasuriya 2021 study analyzed volumes of plasma that were in-line with our study, but their assay used low-pass whole genome sequencing only. They did not distinguish between ctDNA-positive and ctDNA-negative samples and instead stratified by median of ctDNA fraction estimate by ichorCNA.

From a technical standpoint, in the present work the authors report the adoption of specific computational techniques and analyze sequence mCRPC-distinct alterations such as AR alterations (including SNV, indels, CNA and structural rearrangements), AR enhancer amplifications, and MYC/MYCN gains. They demonstrate an association with worse prognosis that relies on alterations beyond the fraction of ctDNA, particularly relying on AR alterations. However, there are some technical concerns that should be addressed (see Major comments).

In summary, this work introduces the concept of a mCRPC-specific liquid-biopsy test that could overcome certain limitations of commercial assays, despite its applicability is more likely to be exploited in a research context rather than in the clinical setting. Indeed, ctDNA prognostication

is not sufficient by itself to drive patient-tailored strategies in future clinical trials. Besides, there are technical points that need revision.

Response: We agree with the reviewer that knowledge of ctDNA fraction or concentration alone either by traditional methods or by also identifying *AR* gain, *AR*-GSRs, or *MYC/MYCN* gain will not change medical practice despite the clear prognostic utility. Predictive utility will require demonstration of specific genetic alterations and their predictive association in a randomized controlled clinical trial of a novel therapy designed to target these alterations. We have constructed a multivariable clinical-genetic model using the present assay and this work will be presented orally at ASCO 2024 and has been submitted as a separate manuscript (the abstract and separate manuscript are under embargo until ASCO 2024). The current manuscript establishes the assay and data analysis, the importance of ctDNA fraction and positivity status determined using this assay, and the prognostic association of some of the novel components like *AR*-GSRs, *AR* ecDNA amplification signature, and *MYC/MYCN* gain. We plan to evaluate predictive associations of specific *AR* and non-*AR* based genetic alterations in the present A031201 phase 3 trial testing enzalutamide +/- abiraterone, but this will be reported separately because it is beyond the scope of the current work.

Major comments

1. The way the authors present group distinction into “ctDNA high” (Group 1) vs “ctDNA low harboring specific alterations” (Group 2) is not clinically relevant and frankly confounding. These two groups are in fact technical spectra within a broader “ctDNA positive” classification, based on detection of high aneuploidy (Group 1) or pathogenic genetic findings without high aneuploidy (Group 2). Both are distinct from “ctDNA negative” results (Group3). Despite focusing throughout the dissertation on the difference between Group 1 and Group 2 from a technical standpoint, their individual impact on prognosis is limited as highlighted also in the abstract (“ctDNA high and ctDNA-low patients had similar rPFS and OS”, that was worse than “ctDNA negative” patients). Indeed, rPFS and OS of Group 1 and 2 are overlapping, and statistical tests are based on the difference with Group 1 vs Group 3 and Group 2 vs Group 3 (Figure 6). Therefore, from an utilitarian perspective targeting a broader audience also including physicians, I recommend to shift the focus on the prognostic value of “ctDNA positive” vs “ctDNA negative”, moving in the background the discussion on “ctDNA high” vs “ctDNA low”. In the results I suggest to: (1) first present how “ctDNA positive” samples were defined, therefore as “ctDNA high” vs “ctDNA low”; (2) then, when addressing prognostication, make the comparison of “ctDNA positive” vs “ctDNA negative” as the central point, then proving that worse prognosis was independent on how “ctDNA positive” samples were defined (Group 1 vs Group 2). The same considerations apply to the abstract. In this regard, also inverting Figure 6C and 6D with 6A and 6B would be more straightforward.

Response: We agree that the ctDNA high Group 1 and ctDNA low Group 2 in our original submission are technical spectra within a broader “ctDNA positive” classification. The reviewer is correct that Group 1 can be described as having high aneuploidy and that Group 2 can be described as having pathogenic genetic findings without high aneuploidy. **We revised the entire manuscript to re-define Group 1 as having high ctDNA aneuploidy and Group 2 as having low ctDNA aneuploidy**, which we hope makes this important point clearer.

Leveraging *AR*-GSRs and/or a copy gain in *AR*, *MYC*, or *MYCN* to classify samples in Group 2 as “ctDNA positive” is a unique aspect of our work that, to the best of our knowledge, has not been used in previous studies to define the presence of ctDNA in mCRPC. Because this was a

unique classification, and the majority of samples in Group 2 were ctDNA-positive as a result of this unique classification (159/256, or 62% of samples in Group 2), we think it is important to benchmark and test clinical outcomes of Groups 1 and 2 separately (in addition to evaluating them collectively as the “ctDNA-positive” samples in the study). To make this benchmarking more rigorous, we:

- 1) Added **new Supplementary Figure S4**, which focused on the 479 patients that lacked detection of aneuploidy or pathogenic somatic mutations (these 479 patients are denoted by gray bars at the top of Figures 2a and 3a). In these 479 patients, those lacking detection of *AR*, *MYC*, or *MYCN* alterations (n = 320) had superior rPFS and OS compared to those patients where these alterations were detected (n = 159). These new data strongly support the utility of the current work over current ctDNA assessment methods based on aneuploidy and/or mutations only.
- 2) Added **new Supplementary Figure S6A**, which illustrates that variant allele fractions of mutations detected in samples belonging to ctDNA aneuploidy-high Group 1 were higher than the variant allele fractions of mutations detected in samples belonging to ctDNA aneuploidy-low Group 2.
- 3) Added new data comparing hazard ratios for rPFS and OS in Group 1 vs. Group 2, and found that Group 1 had shorter OS (but not rPFS) compared to Group 2 and higher risk of death (the **new hazard ratios with 95% confidence intervals and P-values are in the text of the Results, under the heading “ctDNA-positive patients have poor clinical outcomes”**):

“There was no statistical difference in rPFS between Groups 1 and 2 (HR = 1.1, 95% CI: 0.9-1.4, P = 0.21)”

“The HR for OS was higher in Group 1 compared with Group 2 (HR = 1.3, 95% CI: 1.0-1.6, P = 0.03)”

Finally, we agree with the reviewer that from an utilitarian perspective targeting a broader audience also including physicians, the focus should shift to the prognostic value of “ctDNA positive” vs “ctDNA negative”. Therefore, **we revised Figure 6 by swapping panels A/B and C/D** to present the clinical outcomes of “ctDNA positive” vs “ctDNA negative” patients first, and then present the stratification of the ctDNA-positive patients into Groups 1 and 2 second. Accordingly, **the text of the Results section has been completely revised under the heading “ctDNA-positive patients have poor clinical outcomes”**.

2. The rPFS/OS multivariable analysis is limited to the ctDNA fraction as the only additional variable, tuning down the conclusions of the study regarding the prognostic value of the liquid biopsy test. The claim that “there were many patients in the low- and intermediate-risk (Halabi) groups that were in ctDNA-high Group 1 and ctDNA-low Group 2” would be better supported by including these clinical-pathological features in the multivariable analysis. It would be beneficial for the relevance of the paper to first perform univariate analysis of these clinical-pathological features for rPFS/OS, and then include statistically significant findings in the multivariable analysis to prove that the prognostic value of liquid biopsy is maintained and independent from already known prognosticators.

Response: We agree that a true multivariable clinical-genetic model is preferable to fully prognosticate survival and perhaps predict for future targeted therapies and treatment interactions. In fact, this full clinical-genetic model of overall survival will be presented as an oral abstract at ASCO 2024 and has been submitted for a separate publication (both are

currently under embargo). The present manuscript is focused on establishing the *AR*-ctDETECT assay, testing prospectively-defined hypotheses regarding associations with outcome for ctDNA content as well as selected assay components such as *AR*-GSRs and *AR* ecDNA amplification signature, and the utility of leveraging these alterations along with gain of *AR* and/or *MYC/MYCN* to detect ctDNA. We have provided additional prognostic information in response to Reviewer #1 in our revision by demonstrating the additional prognostic utility in defining ctDNA positive specimens by these features (**new Supplementary Figure S4**).

3. In the manuscript, it is unclear whether there were ctDNA positive samples supported only by aneuploidy without any genetic alteration. Indeed, mutations and other genetic alterations are scatterly reported in Figure 2 to Figure 5. A clear summary figure showing overlapping between Group 1 and Group 2 is needed as the main figures, perhaps adapted from Supplementary S4. From this latter figure, it appears that all Group 1 samples would have been also identified by SNV/indel/CNA/fusion calling and this is relevant also to resize the importance of Group 1 vs Group 2.

Response: We updated the UpSet plot that was originally presented as Supplementary Figure 4A to include the category “ctDNA aneuploidy”. Inclusion of this new category in the UpSet plot demonstrates that 54 specimens in ctDNA aneuploidy-high group 1 were classified as ctDNA-positive only by aneuploidy without any pathogenic mutation or alteration in *AR*, *MYC*, or *MYCN*. As suggested by the reviewer, we made this information more prominent in the manuscript by moving the revised UpSet plots (formerly Supplementary Figures 4A and 4B) to the beginning of Figure 4 (**revised Figures 4A and 4B**).

4. The application of ichorCNA to off-targeted reads needs validation. Since the authors lack low-pass WGS data for which ichorCNA was designed, a proper validation of the methodology or, otherwise, a reference of previous works that validated it is necessary. Indeed, discarding targeted reads and leveraging only on off-targeted DNA is not a standard procedure with ichorCNA. I recommend to check whether any statistical difference is present in VAF-based ctDNA estimations between Group 1 and Group 2. This would help to validate the results. Besides, it is concerning that the lowest ctDNA estimate for healthy individuals (4.7%) is a log₁₀ higher compared to the minimum amount of ctDNA in cancer patients (0.4%). May the authors provide some explanation or comment on this?

Response: Based on comments from the other reviewers, **we updated the terminology in our paper** to more accurately reflect that the modified ichorCNA method we used to calculate “ctDNA fraction” is in fact measuring genome-wide aneuploidy. Therefore, we now refer to the “ctDNA aneuploidy fraction” throughout the paper.

The strategy of using off-target reads as input for ichorCNA has been published previously (PMID: 34353895). The developer of ichorCNA, Dr. Gavin Ha, has also described this method (https://github.com/GavinHaLab/ichorCNA_offtarget). A similar approach of using off-target reads to infer ctDNA fraction from a targeted cfDNA-seq assay was published and validated by TEMPUS, which they call OTTER (Off-Target Tumor Estimation Routine, PMID: 34215841). This ctDNA fraction calculation is part of the TEMPUS xM Monitor assay, which was released November 2023. We neglected to cite these papers in our original submission, and have **revised the Patients and Methods section under the “Estimation of ctDNA aneuploidy fraction” section** as follows:

“The circulating tumor DNA (ctDNA) aneuploidy fraction was estimated by the ichorCNA R package (v.0.2.0)³² with R (v.4.2.0) using off-target reads as input for ichorCNA as described previously³⁴ (https://github.com/GavinHaLab/ichorCNA_offtarget). Specifically...”

We also revised the Results under the **“Baseline plasma cfDNA correlates with ctDNA aneuploidy fraction”** section as follows:

“The ichorCNA algorithm was developed to estimate circulating tumor DNA (ctDNA) fraction from aneuploidy detected in low-pass whole genome sequencing of cfDNA³². Previous studies demonstrated the off-target reads from targeted DNA-seq assays can be used to approximate low-pass whole genome sequencing and be leveraged for estimating ctDNA fraction^{33, 34}. Therefore, we removed the on-target reads from each sample and used the remaining off-target reads as input in ichorCNA.”

To address the comment that the approach needs validation, we **added data in new Supplementary Figure S6** to benchmark our application of ichorCNA to off-target reads against alternative inputs that are used to calculate ctDNA fraction, including variant allele fraction of tumor-derived pathogenic mutations, copy number alterations in individual genes, and a separate algorithm (CNVkit). Overall, the correlations we observed validate the application of ichorCNA to off-targeted reads. **We revised the Results under the heading “Alterations in common mCRPC genes are detectable in ctDNA”** as follows:

“We used the genetic alterations detected in ctDNA to benchmark the ctDNA aneuploidy fraction estimates derived from using off-target reads as input for ichorCNA. For instance, the variant allele fraction (VAF) of somatic tumor-derived mutations has been used for calculating ctDNA fraction in studies where high plasma volumes yielded high cfDNA concentrations, enabling high sequencing depth and sensitive mutation detection^{17, 28}. We found that the VAFs of mutations detected across 183 cfDNA specimens were higher in samples having high ctDNA aneuploidy than in samples having low ctDNA aneuploidy (Supplementary Figs. S6A&B). Further, the maximum VAF detected in each of these 183 cfDNA specimens displayed a positive correlation with the ctDNA aneuploidy fraction (Supplementary Figure S6C). This positive correlation between maximum VAF and ctDNA aneuploidy fraction was highest for TP53, which was the most frequently-mutated gene in our study (Supplementary Figure S6C). Another input that has been used to calculate ctDNA fraction is the magnitude of tumor-derived somatic copy number gains or losses of targeted genes^{21, 25, 36}. We observed positive correlations between the ctDNA aneuploidy fraction and magnitude of copy number gain detected in the AR gene body or enhancer in cfDNA samples where these gains occurred (Supplementary Figures S6D&E), and a negative correlation between the ctDNA aneuploidy fraction and magnitude of copy number loss in TP53, PTEN, or RB1 in cfDNA samples where these losses occurred (Supplementary Figures S6F-H). Finally, CNVkit is an algorithm to infer and visualize tumor-derived copy number alterations from targeted DNA-seq data⁵³. Visual inspection of the CNVkit output for all 776 plasma specimens confirmed a high degree of genome-wide copy number gains and losses in samples predicted by ichorCNA to have high ctDNA aneuploidy, and a relative paucity of genome-wide copy number gains and losses in samples below the ichorCNA ctDNA aneuploidy threshold (Supplementary Figure S6I). Collectively, these benchmarking data demonstrate that the ctDNA aneuploidy

fraction estimated by ichorCNA correlates with inputs used to calculate ctDNA fraction in cfDNA.”

To address the concern related to cutoffs between ctDNA estimates for cases and controls, it is important to note that aneuploidy-based ctDNA fraction estimates below 10% are “noisy” regardless of the assay platform. Therefore, our finding that the lowest ctDNA aneuploidy fraction estimate for healthy individuals (4.7%) is a log₁₀ higher compared to the minimum amount of ctDNA in cancer patients (0.4%) should not be concerning because these values fall within this range where data are known to be noisy. For instance, Husain 2023 and other reports from Foundation Medicine use an aneuploidy-based strategy similar to ours for calculating ctDNA fraction. For cfDNA samples with $\geq 10\%$ ctDNA, this aneuploidy-based calculation is the reported ctDNA fraction (note this cutoff of 10% ctDNA is comparable to our cutoff of 14.2% ctDNA). For cfDNA samples with $< 10\%$ ctDNA, they detect ctDNA by identifying somatic mutations, and use the variant allele fraction of those mutations to determine ctDNA fraction. In our study, we showed utility of combining detection of AR-GSRs and/or gains in *AR/MYC/MYCN* with detection of pathogenic mutations for detecting ctDNA in samples falling below the threshold of detection by aneuploidy-based methods.

5. The targeted-DNA panel of genes is unreported in the manuscript. Despite several mutations and other gene alterations are reported in Figures, a clear disclosure of the genes populating the sequenced panel is mandatory.

Response: The targeted-DNA panel of genes was reported in Figure 1b, but may have been unclear because the Figure 1 legend was mistakenly omitted (see next comment). **The Figure 1 legend has been added.** Supplementary Data S1 also provided detailed information of the targeted genomic coordinate details in .bed format so others can replicate the exact bait panel design used in our study. We **added a descriptive title to Supplementary Data S1** to make this clearer.

Minor comments

1. Figure 1 caption is missing.

Response: Thank you for bringing this to our attention. We are disappointed this escaped our notice and **we have added the Figure 1 Legend** to the revised manuscript.

2. The volume of available plasma per patient should be reported as average and standard deviation. Did the amount of plasma influence the results, i.e., less plasma is correlated with ctDNA negativity?

Response: We **added the average and standard deviation of available plasma per patient to the Patients and Methods, under the “Cell-free DNA (cfDNA) isolation and yield” section** as requested:

“Cell-free DNA (cfDNA) was isolated from 1-3 mL (mean = 2.12 mL, standard deviation = 0.49 mL) of input plasma using a Circulating Nucleic Acid kit (Qiagen catalog 55114) according to the manufacturer’s recommendation.”

We explored whether plasma input volume influenced the ctDNA detection results, and found no pattern as illustrated in **Reviewer Figure 1**.

3. The “ctDNA-positive patients have poor clinical outcomes” paragraph is a bit redundant, i.e., the HRs for rPFS are reported twice.

Response: Based on this Reviewer’s Major Comment #1, the text of the Results section has been completely revised under the heading “ctDNA-positive patients have poor clinical outcomes” (see above for our response to that comment). In these revisions, we deleted the redundancy noted in this Minor Comment.

Response to Reviewer #3 (Prostate cancer genomics):

General comment: The manuscript is well written with high quality images and interesting results with regards to AR GSRs. However, two major concerns were noted, one with regards to the technical aspects of the manuscript and one with regards to novelty.

Response: We thank the reviewer for the positive comments about our manuscript being well written with high quality images and interesting results with regards to AR GSRs.

Technical aspects: The methodology applied in the manuscript (both lab and informatics) is of high quality. However, it is noteworthy is that ichorCNA, which was developed for low-pass whole genome sequencing data, is applied on the panel data to estimate the cancer DNA fraction. It is apparent from the data presented on healthy donors that this approach has a problem. In the healthy donors the ctDNA fraction was up to 12.2%. Considering that the median ctDNA fraction on ~4800 cfDNA prostate samples reported by Foundation Medicine

(PMID: 36265119) was ~3% then an assay-specific threshold of 14.2% ctDNA is not good enough.

Response: We thank the reviewer for the positive comments about our methodology being of high quality.

The strategy of using off-target reads as input for ichorCNA has been published previously (PMID: 34353895). The developer of ichorCNA, Dr. Gavin Ha, has also described this method (https://github.com/GavinHaLab/ichorCNA_offtarget). A similar approach of using off-target reads to infer ctDNA fraction from a targeted cfDNA-seq assay was published and validated by TEMPUS, which they call OTTER (Off-Target Tumor Estimation Routine) (PMID: 34215841). This ctDNA fraction calculation is part of their TEMPUS xM Monitor assay, which was released November 2023. We neglected to cite these papers in our original submission, and have **revised the Patients and Methods section under the “Estimation of ctDNA aneuploidy fraction” section** as follows:

“The circulating tumor DNA (ctDNA) aneuploidy fraction was estimated by the ichorCNA R package (v.0.2.0)³² with R (v.4.2.0) using off-target reads as input for ichorCNA as described previously³⁴ (https://github.com/GavinHaLab/ichorCNA_offtarget). Specifically...”

We also **revised the Results under the “Baseline plasma cfDNA correlates with ctDNA aneuploidy fraction” section** as follows:

“The ichorCNA algorithm was developed to estimate circulating tumor DNA (ctDNA) fraction from aneuploidy detected in low-pass whole genome sequencing of cfDNA³². Previous studies demonstrated the off-target reads from targeted DNA-seq assays can be used to approximate low-pass whole genome sequencing and be leveraged for estimating ctDNA fraction^{33,34}. Therefore, we removed the on-target reads from each sample and used the remaining off-target reads as input in ichorCNA.”

To address the comment that the approach needs validation, we **added data in new Supplementary Figure S6** to benchmark our application of ichorCNA to off-target reads against alternative inputs that are used to calculate ctDNA fraction, including variant allele fraction of tumor-derived pathogenic mutations, copy number alterations in individual genes, and a separate algorithm (CNVkit). Overall, the correlations we observed validate the application of ichorCNA to off-targeted reads. **We revised the Results under the heading “Alterations in common mCRPC genes are detectable in ctDNA”** as follows:

“We used the genetic alterations detected in ctDNA to benchmark the ctDNA aneuploidy fraction estimates derived from using off-target reads as input for ichorCNA. For instance, the variant allele fraction (VAF) of somatic tumor-derived mutations has been used for calculating ctDNA fraction in studies where high plasma volumes yielded high cfDNA concentrations, enabling high sequencing depth and sensitive mutation detection^{17,28}. We found that the VAFs of mutations detected across 183 cfDNA specimens were higher in samples having high ctDNA aneuploidy than in samples having low ctDNA aneuploidy (Supplementary Figs. S6A&B). Further, the maximum VAF detected in each of these 183 cfDNA specimens displayed a positive correlation with the ctDNA aneuploidy fraction (Supplementary Figure S6C). This positive correlation between maximum VAF and ctDNA aneuploidy fraction was highest for TP53, which was the most frequently-mutated gene in our

study (Supplementary Figure S6C). Another input that has been used to calculate ctDNA fraction is the magnitude of tumor-derived somatic copy number gains or losses of targeted genes^{21, 25, 36}. We observed positive correlations between the ctDNA aneuploidy fraction and magnitude of copy number gain detected in the AR gene body or enhancer in cfDNA samples where these gains occurred (Supplementary Figures S6D&E), and a negative correlation between the ctDNA aneuploidy fraction and magnitude of copy number loss in TP53, PTEN, or RB1 in cfDNA samples where these losses occurred (Supplementary Figures S6F-H). Finally, CNVkit is an algorithm to infer and visualize tumor-derived copy number alterations from targeted DNA-seq data⁵³. Visual inspection of the CNVkit output for all 776 plasma specimens confirmed a high degree of genome-wide copy number gains and losses in samples predicted by ichorCNA to have high ctDNA aneuploidy, and a relative paucity of genome-wide copy number gains and losses in samples below the ichorCNA ctDNA aneuploidy threshold (Supplementary Figure S6I). Collectively, these benchmarking data demonstrate that the ctDNA aneuploidy fraction estimated by ichorCNA correlates with inputs used to calculate ctDNA fraction in cfDNA.”

The commercial Foundation Medicine assay used in the reference cited by the reviewer (Husain 2023; JCO Precis Oncol) has a minimum input cfDNA mass requirement of >20 ng, which is incompatible with over half of the A031201 specimens we analyzed in our study (Figure 1C, Supplementary Figure S2, and Supplementary Table S2).

In Husain 2023 and other reports from Foundation Medicine (such as Tukachinsky 2021), the assay requirement is 18-20 mL of blood and 20-100 ng input cfDNA (approximately 6000-30,000 haploid genome equivalents) for library construction prior to targeted DNA-seq. This enables Foundation Medicine to take a 2-pronged approach to calculating ctDNA fraction. First, Foundation Medicine uses an aneuploidy-based strategy similar to ours for calculating ctDNA fraction. For cfDNA samples with $\geq 10\%$ ctDNA, this aneuploidy-based calculation is the reported ctDNA fraction (note this cutoff of 10% ctDNA is comparable to our cutoff of 14.2% ctDNA). However, because of known inaccuracies of detecting aneuploidy in samples with $<10\%$ ctDNA, when cfDNA samples are below this cutoff, Foundation Medicine defers to calculating ctDNA fraction using the variant allele fraction of tumor-derived somatic mutations. Because of the high cfDNA input used for their assay (containing a high number of haploid genome equivalents), Foundation Medicine is able to perform DNA-seq to $>5000\times$ depth, which has excellent sensitivity for detecting tumor-derived somatic mutations present at very low variant allele fraction. In contrast, the low volumes of plasma available for many of the A031201 samples in our study (yielding low cfDNA mass and a low number of genome equivalents for input) precludes sequencing at this depth. Therefore, for the A031201 samples reported in our study, we had much lower sensitivity for detecting these tumor-derived somatic mutations in samples with low levels of ctDNA. Indeed, only 183/776 specimens in our study displayed a pathogenic mutation in a panel gene (Figures 4A&B). To overcome this, we leveraged AR-GSRs and/or a copy gain in AR, MYC, or MYCN to classify samples below our aneuploidy threshold as “ctDNA positive”. This is a unique aspect of our work that, to the best of our knowledge, has not been used in previous studies to define the presence of ctDNA in mCRPC.

It is unclear whilst reading the manuscript why ichorCNA was applied and not other copy-number tools developed for panel data? The coverage is in general very low, due to the limited amount of plasma. Did this lead to even more noise when applying panel-based copy-number tools?

Response: Very few tools estimate tumor fraction. ichorCNA makes this estimation, but other copy number tools developed for panel data (like CNVkit) do not. As described above, we **added new data in new Supplementary Figure S6** to benchmark our application of ichorCNA to off-target reads against alternative inputs that are used to calculate ctDNA fraction, including CNVkit as well as the variant allele fraction of tumor-derived gene mutations and the magnitude of tumor-derived gene gains/losses. The correlations we observed demonstrate that the ctDNA aneuploidy fraction estimated by ichorCNA reflects these other inputs commonly used to calculate ctDNA fraction in cfDNA.

The poor ctDNA fraction estimation is likely affecting other aspects of the manuscript as it applied both in the multivariate analysis for AR GSRs and also to stratify the cohort into the three groups for prognostication. Prognostication using ctDNA fraction has been reported multiple times previously with a much more distinct separation of the groups which highlights the problem with using ichorCNA (see PMID 33836265, 29367197, 38418825).

Response: We thank the reviewer for pointing out these excellent studies published by Drs. Annala, Chi, Wyatt, and colleagues. Similar to the Foundation Medicine assay, the targeted cfDNA-sequencing assay developed by this team at Vancouver Prostate Centre has very high sensitivity for detecting ctDNA and calculating ctDNA fraction by leveraging variant allele fractions of tumor-derived somatic mutations. As reported in the methods sections of these publications, the input requirement for this “Vancouver assay” is 10-100 ng cfDNA (or 3000-30,000 haploid genome equivalents) for library preparation prior to targeted DNA-seq, with a companion whole blood germline assay to enhance sensitivity for disambiguating mutations in ctDNA from germline mutations or clonal hematopoiesis (CHIP). These requirements of the Vancouver assay are incompatible with 1) the low mass of cfDNA in many of the A031201 specimens (Figure 1C, Supplementary Figure S2, and Supplementary Table S2), and 2) the absence of a whole blood germline control (our study was only approved for plasma).

The Vancouver assay uses the variant allele fraction of tumor-derived somatic mutations for calculating ctDNA fraction. The high number of haploid genome equivalents used as input for the Vancouver assay (3,000-30,000) allows for high DNA-sequencing depth (>3,000) and high sensitivity for detecting these tumor-derived somatic mutations. In our study, because of the low plasma and cfDNA input available for many of the A031201 samples, we only detected pathogenic alterations in a panel gene for 183/776 specimens (Figures 4A&B). Therefore, we used an aneuploidy-based method for calculating ctDNA fraction estimates across all 776 samples. As outlined in our response to this reviewer’s previous comment, we benchmarked the ichorCNA-based calculation method against other metrics reflecting ctDNA fraction, and observed consistent positive correlations.

Finally, we recognize that the studies published by Drs. Annala, Chi, Wyatt, and colleagues have established the prognostic significance of detecting ctDNA by the Vancouver assay. We recognize the more distinct separation for OS and rPFS they have reported for high/medium/low ctDNA groups in their publications (using ctDNA cutoffs of 2% and 30%). However, an important distinction between these published reports and our current study is that in our study the team performing cfDNA-seq and analysis of ctDNA (Knutson, Kobilka, Lyman, Munro, Li, and Dehm) were blinded to the patient outcomes analyzed by the biostatistics team (Luo, Guo, and Halabi), meaning our ctDNA definitions were not influenced by knowing which cutoffs could provide the most distinct separation for OS and rPFS. We argue that our use of samples from a Phase 3 trial coupled with a blinded approach to ctDNA analysis and testing clinical associations

provides a high degree of rigor that validates and extends these and other excellent reports of the prognostic significance of ctDNA detection.

In response to the comment that “The poor ctDNA fraction estimation is likely affecting other aspects of the manuscript as it applied both in the multivariate analysis”, as described above, we **added new data in new Supplementary Figure S6** to benchmark our application of ichorCNA to off-target reads against alternative inputs that are used in other studies to calculate ctDNA fraction, including variant allele fraction of tumor-derived gene mutations, the magnitude of tumor-derived copy number gains/losses, and a separate algorithm for targeted panel data (CNVkit).

Novelty aspects: Many groups have presented ctDNA of advanced prostate cancer which includes in-depth interrogation of the androgen receptor and the common drivers of prostate cancer and also prognostication based on ctDNA. The presented manuscript does therefore not harbor novelty in relation to the currently published literature. The analysis of AR GSRs is potentially of interest, but it is affected by the poor cancer DNA fraction estimation.

Response: we would like to emphasize these novel aspects of our study that were not appreciated by the reviewer:

- 1) We leveraged *AR*-GSRs and copy number gains in frequently-altered oncogenes (*AR*, *MYC*, *MYCN*) to identify ctDNA in plasma specimens with low abundance, low cfDNA mass, low aneuploidy, and low power for detecting oncogenic mutations. This was appreciated by Reviewer 4 in their comment: “Although the finding that ctDNA positive patients have worse prognosis than ctDNA negative mCRPC patients is not novel, the manuscript identifies an additional group of poor prognosis patients that have detectable molecular alterations in their cfDNA despite a low ctDNA fraction”
- 2) We reported novel genomics findings including: a) gain of *MYCN* in the context of *MYC* gain; b) associations of *AR*-GSRs with rPFS and OS; and c) demonstration that an “*AR* ecDNA” signal enhances associations between *AR*-GSRs and rPFS/OS
- 3) Ours is the first report of cfDNA-seq in specimens from a randomized phase 3 trial of first-line mCRPC patients with linked to clinical outcomes and raw data made publicly available in dbGaP for the field to validate and build on our work.
- 4) A blinded study design where investigators performing cfDNA-sequencing analysis and defining ctDNA were blinded to patient clinical data, and investigators at a separate institution performed the statistical analyses with the clinical data.
- 5) The tumor fraction values estimated by ichorCNA in this study approximate a normal distribution in 776 samples with a long tail skew at higher values. Based on these results, we calculated an assay-dependent threshold for high ctDNA tumor fraction. We have revised the entire manuscript to make clear that our tumor fraction estimation is aneuploidy-based and the threshold value should not be directly compared against variant allele fraction-based methods. Our method provides a new way to classify ctDNA from panel sequencing of patient cfDNA samples.

Response to Reviewer #4 (AR, prostate cancer diagnosis):

In this manuscript, the authors developed a targeted cell-free DNA (cfDNA) sequencing assay to evaluate 69 genes (including AR) frequently altered in metastatic castration-resistant prostate cancer (mCRPC), and they applied the assay to 776 banked EDTA baseline plasma samples (1-3 mL) from the Alliance A031201 phase 3 trial, which randomized men with mCRPC to treatment with first-line enzalutamide with or without abiraterone. The patients were stratified into 3 groups based on the level of ctDNA fraction and the presence of known pathogenic mutations, AR structural variant, or copy gain in AR, MYC, or MYCN. ctDNA-high patients had the worst rPFS and OS, followed by ctDNA-low patients with detectable gene alterations, compared to ctDNA-negative patients which had the most favorable prognosis.

Although the finding that ctDNA positive patients have worse prognosis than ctDNA negative mCRPC patients is not novel, the manuscript identifies an additional group of poor prognosis patients that have detectable molecular alterations in their cfDNA despite a low ctDNA fraction. This work has many strengths including large-scale ctDNA analysis of prospectively collected plasma specimens from a large, relatively uniform clinical cohort of mCRPC patients in the setting of a phase 3 clinical trial, and the use of a unique targeted cfDNA sequencing assay capable of comprehensive profiling of AR alterations including mutations, amplification, enhancer amplification, gene structural arrangements, and gains in MYC and MYCN. Detailed descriptions of cfDNA sequencing and bioinformatic analysis methods are provided. It is also a strength that the cfDNA sequencing and bioinformatics team was blinded to the clinical data, while a separate team at another institution performed the statistical analyses with the clinical data. Together, the manuscript provides a large body of genomic ctDNA data that may serve as a useful resource for the prostate cancer field.

Response: We thank the reviewer for the excellent summary of these strengths of our study.

The manuscript also has several important limitations which the authors appropriately discuss, including the use of a research-grade (rather than clinical-grade) ctDNA assay, the lack of germline controls, and the lack of analyses of matched tumor tissues. Several additional comments and suggestions are as follows:

Major comments:

- The finding that control plasma samples from a pool of healthy male donors had estimates of ctDNA fraction of 4.7% to 12.2% (Fig 1D) is rather odd. Can the authors provide more details regarding these “healthy” donors? What are their ages? How do the authors know that they do not harbor undiagnosed malignancies?

Response: To address this question directly, we edited the Methods section under the “Definition of ctDNA-positive cfDNA samples” heading to provide information on the source of control plasma samples (note that in response to Reviewer 1, we also edited this section to provide better rationale for using a cutoff of mean + 3 SD):

“The mean ctDNA aneuploidy fraction estimate of DNA-seq assays (n = 8) performed on control cfDNA isolated from a pool of plasma from healthy male donors (ZenBio, Inc., Catalog #SER-PDP-10) was 0.0698 with a standard deviation of 0.0241. Samples exceeding the mean + 3SD (> 0.1421) were classified as ctDNA-positive

and belonging to ctDNA aneuploidy-high Group 1 based on the rationale that samples below the mean + 3SD cutoff are 99.87% likely to lack ctDNA aneuploidy.”

We do not have access to donor information, so we are unable to know their ages or status of undiagnosed malignancies.

As described in our response to Reviewer #3, aneuploidy-based ctDNA fraction estimates below 10% are “noisy” regardless of the assay platform, so our finding that normal cfDNA falls within this noisy range is not odd. For instance, In Husain 2023 and other reports from Foundation Medicine (such as Tukachinsky 2021), the assay requirement is 18-20 mL of blood and 20-100 ng input cfDNA (approximately 6000-30,000 haploid genome equivalents) for library construction prior to targeted DNA-seq. This enables Foundation Medicine to take a 2-pronged approach to calculating ctDNA fraction. First, Foundation Medicine uses an aneuploidy-based strategy similar to ours for calculating ctDNA fraction. For cfDNA samples with $\geq 10\%$ ctDNA, this aneuploidy-based calculation is the reported ctDNA fraction (note this cutoff of 10% ctDNA is comparable to our cutoff of 14.2% ctDNA). However, because of known inaccuracies of detecting aneuploidy in samples with $<10\%$ ctDNA, when cfDNA samples are below this cutoff, Foundation Medicine defers to calculating ctDNA fraction using the variant allele fraction of tumor-derived somatic mutations. Because of the high cfDNA input used for their assay (containing a high number of haploid genome equivalents), Foundation Medicine is able to perform DNA-seq to $>5000X$ depth, which has excellent sensitivity for detecting tumor-derived somatic mutations present at very low variant allele fraction. In contrast, the low volumes of plasma available for many of the A031201 samples in our study (yielding low cfDNA mass and a low number of genome equivalents for input) precludes sequencing at this depth. Therefore, for the A031201 samples reported in our study, we had much lower sensitivity for detecting these tumor-derived somatic mutations in samples with low levels of ctDNA. Indeed, only 183/776 specimens in our study displayed a pathogenic mutation in a panel gene (Figures 4A&B). To overcome this, we leveraged AR-GSRs and/or a copy gain in AR, MYC, or MYCN to classify samples below our aneuploidy threshold as “ctDNA positive”. This is a unique aspect of our work that, to the best of our knowledge, has not been used in previous studies to define the presence of ctDNA in mCRPC.

- In Figs. 2G and 2H, the authors have appropriately corrected for ctDNA fraction in their analyses of hazard ratios for rPFS and OS based on the presence of AR-GSR and AR ecDNA. Did the authors also correct for Halabi Risk score and/or other clinical factors in this analysis? In addition, it would be interesting to test for interaction by treatment group (enzalutamide vs. enzalutamide + abiraterone).

Response: The hazard ratios for rPFS and OS are corrected for ctDNA fraction only in the multivariable analysis. We did not correct for Halabi Risk score or other clinical factors.

We have constructed a multivariable clinical-genetic model using the present assay and this work will be presented orally at ASCO 2024 and will be submitted as a separate manuscript (this abstract and separate manuscript are under embargo until ASCO 2024). The current manuscript establishes the assay and data analysis, the importance of ctDNA fraction and positivity status determined using this assay, and the prognostic association of some of the novel components like AR-GSRs, AR ecDNA amplification signature, and MYC/MYCN gain. We plan to evaluate predictive associations of specific AR and non-AR based genetic alterations in the present A031201 phase 3 trial based on treatment group, but this will be reported separately because it is beyond the scope of the current work.

- It is odd that AR alterations would occur at a higher frequency in ctDNA-low Group 2 compared to ctDNA-high Group 1. Can the authors comment on this?

Response: We revised and moved UpSet plots to Figures 4A and 4B to more clearly illustrate the alterations detected in samples belonging to Group 1 or Group 2. From Figure 4B, it is evident that the majority of samples in ctDNA-low Group 2 are classified as ctDNA-positive because AR alterations were detected. We believe this bias exists because Group 2 status is mainly based on detection of an AR alteration.

- Figure 4. The identification of TP53 and RB1 loss in the ctDNA of a subset of patients is interesting. Did these patients harbor any evidence of neuroendocrine differentiation based on greater incidence of visceral metastases, lower than expected serum PSA, or other characteristic features? Did these patients have worse clinical outcomes after treatment with the AR-targeting therapies enzalutamide or enzalutamide + abiraterone, as might be expected for patients with neuroendocrine prostate cancer?

Response: Patients with NEPC at baseline were ineligible for the A031201 trial, so there is no evidence for NEPC as a function of *TP53/PTEN/RB1* status (Figure 4N). Adverse clinical outcomes based on *TP53* and/or *RB1* status (detected in tissues or ctDNA) is well-established in clinical practice and was not considered as a prospective hypothesis in the statistical plan for this study. Confining the reporting of P-values to our predefined hypotheses keeps the study integrity and strengthens the rigor and the reliability of scientific conclusions. However, **we performed this post-hoc analysis requested of the reviewer**, which supports the concept that *TP53* and *RB1* alterations, alone or in combination, are associated with increased risk of rPFS and OS:

Reviewer Table 1: Hazard Ratios (univariate) for alterations in *TP53* and/or *RB1*

Gene	HR OS (95%CI)	Median OS (95%CI)	HR rPFS (95%CI)	Median rPFS (95%CI)
RB1=0 (n=633)	Ref	38.2 (35.5,41.7)	Ref	16.4 (23.0,29.0)
RB1=1 (n=143)	2.8 (2.3,3.4)	19.3 (16.7,21.7)	2.4 (1.9,2.9)	13.6 (11.1,15.3)
TP53=0 (n=660)	Ref	37.3 (34.8,40.2)	Ref	25.3 (22.5,27.6)
TP53=1 (n=116)	2.8 (2.3,3.5)	17.0 (15.2,18.9)	2.5(2.0,3.0)	11.2 (8.7,14.6)
TP53=0 or RB1=0 (n=702)	Ref	36.2 (33.4,38.8)	Ref	24.8 (22.3,27.3)
TP53=1&RB1=1 (n=74)	3.1 (2.4,4.1)	16.4 (14.6,19.4)	2.5 (2.0,3.3)	11.4 (8.9,14.8)

Reviewer Table 2: Hazard Ratios from the proportional hazard model of OS &rPFS for alterations in *TP53* and/or *RB1* adjusting for the ctDNA aneuploidy fraction

Gene	HR OS (95%CI)	HR rPFS (95%CI)
RB1=0 (n=633)	Ref	Ref
RB1=1 (n=143)	2.2 (1.7,2.7)	2.0 (1.6,2.5)
TP53=0 (n=660)	Ref	Ref
TP53=1 (n=116)	2.1 (1.6,2.7)	2.1 (1.6,2.7)
TP53=0 or RB1=0 (n=702)	Ref	Ref
TP53=1&RB1=1 (n=74)	2.1 (1.6,2.9)	1.9 (1.4,2.6)

- Page 14-15 and Suppl Table S2. The authors state in the text that patients in ctDNA-high Group 1 had higher PSA, lower HGB, higher Alk Phos, Higher LDH, higher frequency of bone and liver metastases, and higher Halabi Risk score compared to ctDNA-negative Group 3. Similarly, ctDNA-low Group 2 also had differences compared to Group 3. Please list the p-values for the differences in clinical features between Groups 1 and 3 and Groups 2 and 3 in Suppl Table S2.

Response: Supplementary Table S2 has been updated to include P-values for risk scores and risk groups. In this study, we only evaluated our prospective hypothesis that ctDNA-positive patients in Groups 1 and 2 will exhibit higher risk scores and consequently belong to poorer risk groups compared to ctDNA negative patients. Confining the reporting of P-values to our predefined hypotheses keeps the study integrity and strengthens the rigor and the reliability of scientific conclusions. Accordingly, we have modified the text of the Results section to only reflect this prospective hypothesis:

“A prospective hypothesis of this study was that ctDNA-positive patients would have poor prognosis. To test this prospective hypothesis, we evaluated relationships with prognostic risk at baseline. Compared to patients in ctDNA-negative Group 3, patients in ctDNA aneuploidy-high Group 1 or ctDNA aneuploidy-low Group 2 were more likely to be classified as high risk and less likely to be classified as low risk, and had higher Halabi Risk scores, which are based on a multivariate prognostic model of OS (Supplementary Table S2)^{14, 31, 54}.”

The risk scores and risk groups incorporate and reflect 8 of the clinical variables reported in Supplementary Table S2, including site of metastasis. We elected to keep these clinical variables in Supplementary Table S2, but eliminated any discussion of comparisons of these individual clinical variables between groups because any differences noted are reflected by the differences between risk scores and risk groups.

- Suppl Table S2 seems to show that Groups 1 and 2 had a greater % of black patients compared to Group 3. Is this a statistically significant difference? If so, the authors should speculate in the Discussion on why there might be differences in ctDNA detection between races. Along those lines, were there differences in ctDNA gene alterations based on race?

Response: Differences in ctDNA detection as a function of race, ethnicity, or genetic ancestry were not considered as prospective hypotheses in the statistical plan for this study. Confining the reporting of P-values to our predefined hypotheses keeps the study integrity and strengthens the rigor and the reliability of scientific conclusions. However, **we performed this post-hoc analysis requested of the reviewer:** The % of black patients in Group 1 vs. Group 3 was not significant (P = 0.0623), and the % of black patients in Group 2 vs. Group 3 was not significant (P = 0.0556). We point out that these p-values are not reliable because they are based on a hypothesis that was not pre-defined and was not adjusted for multiplicity.

- Figure 6. Because the Alliance A031201 trial did show a significant improvement in rPFS with the addition of abiraterone to enzalutamide, it would be interesting to test for any interaction by treatment group in the rPFS analyses (Fig 6A and 6C).

Response: Comparing differences in clinical outcome by treatment group was not considered as a prospective hypothesis in the statistical plan for this study. Such additional analyses focused on developing predictive biomarkers are coming in future manuscripts that will build off the data reported in this study.

Minor comments:

- Figure legend for Fig. 1 is missing.

Response: Thank you for bringing this to our attention. We are disappointed this escaped our notice and **we have added the Figure 1 Legend** to the revised manuscript.

- Please make clear in the Methods section under “Patients and plasma samples” that these are baseline pre-treatment plasma specimens that were analyzed.

Response: We edited the Patients and Methods section under the “Patients and plasma samples” heading to reflect this important detail as follows:

“Approval was received by the NCTN for correlative science proposal CSC0159 on June 16, 2021, enabling 2 mL of banked pre-treatment EDTA plasma specimens collected from patients at baseline to be shipped on dry ice from the Alliance biorepository at Ohio State University to the University of Minnesota, only for those patients where at least 2 mL of plasma was available.”

- Please label the Supplementary Data files. The version of these files downloaded from the submission site did not have informative file names.

Response: We added informative headers to each of the Supplementary Data files:

*“Supplementary Data S1. Details of Agilent SureSelect Target Panel
Supplementary Data S2. cfDNA yield, tumor aneuploidy fraction, and ctDNA detection in A031201 cfDNA
Supplementary Data S3. Pathogenic mutations detected in A031201 ctDNA
Supplementary Data S4. DNA-seq coverage and log2 copy number ratios of targeted genes relative to control regions
Supplementary Data S5. Counts of AR-GSRs detected in A031201 cfDNA
Supplementary Data S6. Breakpoint details of AR-GSRs detected in A031201 cfDNA”*

- Page 13, “Leveraging detection of AR alterations...” There seems to be a typo with an extra “and” after the word “gains.”

Response: We fixed this typo.

- If the p-values were adjusted using the Benjamini-Hochberg method, the false discovery rate for this correction should be stated.

Response: The adjusted p-values are equal to the FDR. We revised the Figure 2 legend to reflect this:

“(g&h) Forest plots illustrating hazard ratio and 95% confidence intervals for (g) radiographic progression (rPFS) and (h) overall survival (OS) in patients demonstrating indicated cfDNA features. Multivariable analysis is adjusted for ctDNA aneuploidy fraction. P-values are adjusted for multiplicity using the Benjamini-Hochberg method (false discovery rate). A FDR <0.05 is considered statistically significant.”

REVIEWER COMMENTS

Reviewer #1:

The authors adequately addressed my concerns.

Reviewer #2:

The Authors have provided detailed responses to our technical and methodological comments, addressing thoroughly most of our concerns.

However, they have indicated that the multivariable analysis, which should include clinicopathological predictors alongside ctDNA status according to our opinion, is part of another manuscript currently under embargo for presentation at ASCO. Consequently, we are unable to evaluate this aspect of their work at this time.

Reviewer #3:

Revisiting the manuscript the authors have made great effort to improve the manuscript.

However, the limitations of the biobanked material (low plasma and cfDNA input) causes poor coverage which in turn leads to poor sensitivity to detect somatic alterations and prevent robust estimation of ctDNA fraction for many samples. Based on the data presented many samples are likely false negatives, that is, samples that would be ctDNA+, if more plasma was available. Additionally, the ichorCNA aneuploidy ctDNA fraction estimation is applied but is limited to detect ≥ 0.14 ctDNA fraction. The “aneuploidy low group” therefore contains a wide range of ctDNA fractions with different prognosis.

With certainty, this data contains a lot of variability in the ctDNA fraction estimates which is problematic as the ctDNA fraction is the key parameter used throughout the manuscript.

This is demonstrated in supplementary figure 6A. The ctDNA fraction estimation using mutations (e.g. in PTEN) will be more accurate than use of ichorCNA. In group 2 (aneuploidy low), the median mutation VAF is ~0.25, this is a high number. Any prostate cancer genome containing variants with this high VAF will harbor clearly detectable copy-number alterations. This means that the LOD for ctDNA fraction detection using ichorCNA in this dataset is a lot poorer than the threshold applied from healthy donors. It likely randomly varies depending on how much plasma/cfDNA that was available for library prep.

Additionally, the authors argue that the maximum VAF detected is positively correlated to the aneuploidy fraction, however, as seen in Supplementary Figure S6C, this association is driven by outlier cases. The y-axis spread in <0.20 aneuploidy ctDNA fractions is very large indicating that the ctDNA fraction estimation is completely off if ichorCNA is used.

My concern is that the noise will lead to samples with e.g. high/low ctDNA fraction in reality has low/high ctDNA fraction (or even undetectable) which might affect one of the main results of the manuscript, the association between AR-GSRs with poor PFS and OS. In this comparison aneuploidy ctDNA fraction is used. The authors need to perform a subgroup analysis by including only cases ctDNA+ by mutation detection. This is the set that is ctDNA+ almost with certainty for which the ctDNA fraction estimates are robust. As a “true” ctDNA- group in this comparison, the authors should include ctDNA- cases with as high coverage as possible to get a reasonable size cohort (e.g. cutoff of 750x?).

This concern also affects figure 6 with the prognostic groups. The authors should perform the same subgroup analysis to investigate if the prognostic groups are robust.

General comments:

Fraction ctDNA+:

In figure 2A the authors detect ctDNA by aneuploidy in 200 samples, 97 by somatic alterations whereas 479 are negative. Overall, the ctDNA+ rate is therefore $297/(297+479) = 38\%$ (not including amplification-only detection which was used to expand group 2). This is very low. Research groups cfDNA/gDNA profiling have ~60% cfDNA+ rate in comparable cohorts (pmid 29367197, 30458854). In addition, the data presented in the manuscript likely has false positive ctDNA-cases due to the detection of CHIP (clonal hematopoiesis of indeterminate potential) variants, coming from clonal expansions in the white blood cells as gDNA sequencing was not done in parallel. Supplementary Figure S5 the authors show that TP53 is the most common mutation to classify samples as ctDNA+ which very commonly is also occurring in CHIP. This also affects other genes such as ATM which

was also applied to classify a case as ctDNA+. The authors should mention this as a limitation in the discussion.

The authors write:

“Therefore, we classified 60 cfDNA samples with gain of the AR gene body and 2 or more AR-GSRs as likely harboring AR ecDNA (Figures 2D&F, Supplementary Data S5). “

Is this anchored in any data in which the authors have correlated ecDNA with the AR gain level and number of GSRs? Otherwise, the authors should change the definition of the cases with "with gain of the AR gene body and 2 or more AR-GSRs" to "AR alteration high" or something similar.

Figure 4, g-h:

The authors report copy loss of TP53, PTEN and RB1, but are these variants really relevant? If one copy remain the gene is still functional, homozygous loss is required for inactivation or one-copy loss with a mutation of the other allele. E.g. RB1 is very close to BRCA2, often a large segment is detected that cover both genes are but RB1 is still functional if BRCA is homozygously lost and is therefore an irrelevant passenger event. The authors need to motivate the presented data and reason if it is really relevant variants that are displayed.

Reviewer #4:

The authors have responded adequately to my comments. They have also adequately responded to the other reviewer comments, in my opinion.

We thank the Reviewers for their evaluation of our revised manuscript. Our response to each reviewer comment is indicated in blue font.

Response to Reviewer #1:

The authors adequately addressed my concerns.

Response: We thank the reviewer for their time in evaluating our manuscript and guiding improvements during revision.

Response to Reviewer #2:

The Authors have provided detailed responses to our technical and methodological comments, addressing thoroughly most of our concerns.

Response: We thank the reviewer for their time in evaluating our manuscript and guiding improvements during revision.

However, they have indicated that the multivariable analysis, which should include clinicopathological predictors alongside ctDNA status according to our opinion, is part of another manuscript currently under embargo for presentation at ASCO. Consequently, we are unable to evaluate this aspect of their work at this time.

Response: Dr. Halabi's ASCO Abstract #5007 was presented orally on June 1, 2024. The title of the abstract is "*A clinical-genetic (CG) circulating tumor DNA (ctDNA)-based prognostic model for predicting overall survival (OS) in men with metastatic castrate-resistant prostate cancer (mCRPC) treated with potent androgen receptor inhibition (Alliance)*". The abstract text is below:

Authors: Susan Halabi, Bin Luo, Siyuan S Guo, Todd Knutson, Jacqueline Lyman, Anna Kobilka, Himisha Beltran, Emmanuel S. Antonarakis, Jonathan E. Rosenberg, Matt D. Galsky, Charles J. Ryan, William Kevin Kelly, Eric J. Small, Michael J. Morris, Scott M. Dehm, Andrew J. Armstrong

Organizations: Department of Biostatistics and Bioinformatics, Duke Cancer Institute Center for Prostate and Urologic Cancers, Duke University School of Medicine, Durham, NC, Duke University, Durham, NC, Duke University Department of Biostatistics and Bioinformatics, Durham, NC, University of Minnesota, Minneapolis, MN, Dana-Farber Cancer Institute, Boston, MA, University of Minnesota Masonic Cancer Center, Minneapolis, MN, Department of Medicine, Genitourinary Oncology Service, Memorial Sloan Kettering Cancer Center, New York, NY, Mount Sinai School of Medicine, New York, NY, Prostate Cancer Foundation, Santa Monica, CA, Department of Medical Oncology and Urology, Sidney Kimmel Medical College, Thomas Jefferson University, Philadelphia, PA, Helen Diller Family Comprehensive Cancer Center, University of California, San Francisco, San Francisco, CA, Memorial Sloan Kettering Cancer Center, New York, NY, Cancer Institute Center for Prostate and Urologic Cancer, Duke University Medical Center, Durham, NC

Background: We have previously developed and validated a clinical prognostic model of OS in mCRPC men that included these variables: performance status, disease site, opioid analgesic use, lactate dehydrogenase, albumin, hemoglobin, prostate specific antigen, and alkaline phosphatase. The goal of this analysis is to improve upon the clinical model of OS by incorporating ctDNA pathogenic genetic alterations (PGAs).

Methods: Data from the A031201 phase 3 trial of enzalutamide+/- abiraterone were used to develop and validate the CG model of OS. Cell-free DNA was isolated from plasma and analyzed using a 69-gene targeted DNA-sequencing assay for detection of ctDNA PGAs. Genetic features were identified based on feature importance using a random survival forest and the final CG model was trained including clinical and selected genetic factors. Model discrimination was assessed using time-dependent area under the receiver operating characteristic curve (tAUC).

Results: Data were available on 776 patients. In addition to clinical variables, the model included in this order: gains in *AR* and the *AR* enhancer, *MYC*, *RSPO2*, and losses and/or PGAs of *ZBTB16*, *PTEN*, *MSH6*, *PPP2R2A*, *NKX3-1*, *TP53*, *FANCA*, *RB1*, *APC*, *CHD1*, and *BRCA2*, and ichorCNA tumor fraction. tAUCs in clinical and CG models were 0.72 (95% CI=0.72-0.73) and 0.77 (95% CI= 0.76-0.77). Median OS and the hazard ratios by the three- and four- prognostic risk groups are presented in the table.

Conclusions: CG model identified novel ctDNA PGAs prognostic of OS and can be utilized to classify patients into risk groups useful in selecting patients in future trials of mCRPC. Clinical trial information: NCT01949337.

Median OS and hazard ratios by the three- and four-prognostic risk groups.		
Prognostic Risk Groups*	Median Overall Survival (95% Confidence Interval (CI)), months	Hazard Ratio (95% CI)
3-Prognostic Risk Groups		
Low	58.9 (50.1- NR)	0.22 (0.17- 0.27)
Intermediate	35.5 (32.3- 40.2)	0.42 (0.35- 0.51)
Poor	19.3 (17.3- 21.5)	Reference
4-Prognostic Risk Groups		
Low	64.2 (52.8- NR)	0.15 (0.11- 0.19)
Low Intermediate	43.6 (38.2- 48.9)	0.26 (0.20- 0.33)
Intermediate Poor	31.1 (28.3- 33.6)	0.43 (0.34- 0.53)
Poor	17.0 (16.0- 18.9)	Reference

*Patients classified into prognostic risk groups based on the tertile or the quartile of the predicted risk.

This work has been described in a manuscript that was reviewed at *Annals of Oncology*, and a revision has been invited. This revised manuscript entitled “*Clinical-Genetic Circulating Tumor DNA Prognostic Model for Overall Survival in Metastatic Castrate-Resistant Prostate Cancer*” is attached for the Reviewers to evaluate.

Response to Reviewer #3:

Revisiting the manuscript the authors have made great effort to improve the manuscript.

Response: We thank the reviewer for this positive comment.

However, the limitations of the biobanked material (low plasma and cfDNA input) causes poor coverage which in turn leads to poor sensitivity to detect somatic alterations and prevent robust estimation of ctDNA fraction for many samples. Based on the data presented many samples are likely false negatives, that is, samples that would be ctDNA+, if more plasma was available. Additionally, the ichorCNA aneuploidy ctDNA fraction estimation is applied but is limited to detect ≥ 0.14 cDNA fraction. The “aneuploidy low group” therefore contains a wide range of ctDNA fractions with different prognosis.

With certainty, this data contains a lot of variability in the ctDNA fraction estimates which is problematic as the ctDNA fraction is the key parameter used throughout the manuscript.

This is demonstrated in supplementary figure 6A. The ctDNA fraction estimation using mutations (e.g. in PTEN) will be more accurate than use of ichorCNA. In group 2 (aneuploidy low), the median mutation VAF is ~ 0.25 , this is a high number. Any prostate cancer genome containing variants with this high VAF will harbor clearly detectable copy-number alterations. This means that the LOD for ctDNA fraction detection using ichorCNA in this dataset is a lot poorer than the threshold applied from healthy donors. It likely randomly varies depending on how much plasma/cfDNA that was available for library prep.

Additionally, the authors argue that the maximum VAF detected is positively correlated to the aneuploidy fraction, however, as seen in Supplementary Figure S6C, this association is driven by outlier cases. The y-axis spread in < 0.20 aneuploidy ctDNA fractions is very large indicating that the ctDNA fraction estimation is completely off if ichorCNA is used.

My concern is that the noise will lead to samples with e.g. high/low ctDNA fraction in reality has low/high ctDNA fraction (or even undetectable) which might affect one of the main results of the manuscript, the association between AR-GSRs with poor PFS and OS. In this comparison aneuploidy ctDNA fraction is used. The authors need to perform a subgroup analysis by including only cases ctDNA+ by mutation detection. This is the set that is ctDNA+ almost with certainty for which the ctDNA fraction estimates are robust. As a “true” ctDNA- group in this comparison, the authors should include ctDNA- cases with as high coverage as possible to get a reasonable size cohort (e.g. cutoff of 750x?).

This concern also affects figure 6 with the prognostic groups. The authors should perform the same subgroup analysis to investigate if the prognostic groups are robust.

Response: We agree with the reviewer about the limitations of the biobanked material (low plasma and cfDNA input) resulting in poor coverage, which in turn leads to poor sensitivity to detect somatic alterations and prevent robust estimation of ctDNA fraction for many samples. To better understand these factors, we evaluated the associations between cfDNA yield (defined as the mass of cfDNA isolated from a unit volume of plasma, as shown in Fig. 1C) and the detection of somatic mutations. As expected, the samples in which somatic mutations were detected had higher average cfDNA yields than the samples in which somatic mutations were not detected (**new Supplementary Figure S2**). This finding was added to the Results as follows:

Lines 160-165: “We noted that the patients with detected pathogenic mutations had a higher average yield of cfDNA isolated from their plasma specimens compared to those without detected pathogenic mutations (Supplementary Figure S2). This finding may indicate a higher sensitivity for detecting ctDNA-specific mutations in plasma with greater cfDNA amounts, or higher levels of ctDNA in cfDNA with detected pathogenic mutations.”

We also evaluated the associations between cfDNA yield and the classification of patients into ctDNA aneuploidy-high Group 1, ctDNA aneuploidy-low Group 2, or ctDNA-negative Group 3. As expected, the samples in ctDNA aneuploidy-high Group 1 displayed the highest average cfDNA yield, the samples in ctDNA-negative Group 3 had the lowest average cfDNA yield, and patients in ctDNA aneuploidy-low Group 2 had an intermediate average cfDNA yield (**new Supplementary Figure S7**). This finding was added to the Results as follows:

Lines 311-317: “A prospective hypothesis of this study was that ctDNA-positive patients would have poor prognosis. High plasma concentrations of cfDNA are associated with worse survival outcomes in mCRPC patients treated with taxane chemotherapy¹⁶, which is consistent with our observation that mCRPC patients in ctDNA aneuploidy-high Group 1 had the highest average plasma cfDNA concentration, patients in ctDNA-negative Group 3 had the lowest average plasma cfDNA concentration, and patients in ctDNA aneuploidy-low Group 2 had an intermediate average plasma cfDNA concentration (Supplementary Figure S7).”

We re-emphasize that a prospective hypothesis of our study was that ctDNA-positive patients would have poor prognosis, which we tested in a rigorous and blinded fashion using ctDNA aneuploidy fraction as one of several parameters to define ctDNA-positive specimens. The fact that ctDNA aneuploidy-high Group 1 (defined only by the ichorCNA ctDNA aneuploidy fraction cutoff) displayed worse OS than ctDNA aneuploidy-low Group 2 (defined exclusively by genomic alterations and agnostic to ichorCNA ctDNA aneuploidy fraction data because these values were below the cutoff) and ctDNA-negative Group 3 demonstrates that our use of ichorCNA with appropriate thresholds had value for identifying patients with poor prognosis.

We also re-emphasize that the manuscript only refers to ctDNA aneuploidy fraction as the output of ichorCNA. We use this term to make sure our data and conclusions remain distinct from other studies that calculate “ctDNA fraction” using other methods. We agree that ctDNA aneuploidy fraction is not a precise measure of true ctDNA fraction, and we are not trying to suggest these measures are equivalent. For example, in Supplementary Figure S6 of the current manuscript (also Supplementary Figure S6 of the previous version), we benchmarked the ctDNA aneuploidy fraction against other inputs used to measure ctDNA fraction, including the maximum VAF preferred by the reviewer. The benchmarking data demonstrate that ctDNA aneuploidy fraction positively correlates with alternative inputs that have been used in other studies to calculate ctDNA fraction (maximum VAF, magnitude of copy number gains and losses). The reviewer indicates the positive correlations are only driven by outlier cases, which is incorrect as illustrated by the data cutoffs tested in **Reviewer Figure 1**. As expected, these cutoffs demonstrate that the positive correlations do drop off when ctDNA aneuploidy fraction reaches the experimentally-derived cutoff (14.2%). We recognize throughout Supplementary Figure S6 (and Reviewer Figure 1) that the positive correlations are not strong/linear, which clearly shows they are not equivalent. To make sure these important points remain unambiguous, **we added them to the limitations section of the Discussion:**

Lines 452-455: “the low volumes of plasma available for analysis led to low yields of cfDNA in many samples, reducing the sensitivity for detecting ctDNA-specific alterations. These technical limitations likely contributed to false negatives among the patients categorized as ctDNA-negative.”

Lines 462-467: “Accurately detecting tumor-associated mutations in cfDNA and measuring their corresponding VAFs is important for determining ctDNA fractions in cfDNA^{17,28}. Due to the lower sensitivity for detecting these mutations in our study, we calculated a ctDNA aneuploidy fraction for all samples using the ichorCNA algorithm. Although this ctDNA aneuploidy fraction correlated with other inputs used to calculate ctDNA fraction, it is important to note that these metrics are distinct and should not be directly compared.”

Reviewer Figure 1. Scatterplots of ichorCNA ctDNA aneuploidy fraction vs. maximum variant allele fraction for cfDNA specimens harboring a pathogenic mutation. The 4 scatterplots represent the full set of cfDNA specimens harboring a pathogenic mutation ($n = 183$), or subsets derived by cutoffs imposed at 0.50, 0.25, and 0.125 ctDNA aneuploidy fraction. The most frequently-mutated gene (TP53) is denoted by red dots. Trendlines and Pearson correlations coefficients are shown for all genes (grey line) or for TP53 only (red line).

Another prospective hypothesis of our study was that patients with AR-GSRs detected in cfDNA would have shorter survival compared with patients where AR-GSRs were not detected. We tested this prospective hypothesis in a rigorous and blinded fashion (Figures 2G&H). However, the reviewer raised concern about the lack of equivalency between ichorCNA ctDNA aneuploidy fraction and the “true” ctDNA fraction measured by the maximum VAF method affecting the multivariable corrections. The reviewer suggested we perform a subset analysis restricted to the 183 samples in which likely somatic pathogenic mutations were detected, because this represents a subset in which we could use the maximum VAF method to calculate ctDNA fraction for multivariable correction. Although this is a *post-hoc* analysis that was not prospectively-defined, and the sample sizes and numbers of events were small for some of the groups, we performed this subset analysis as required by the reviewer (**Reviewer Table 1**). As demonstrated by the short median OS and rPFS in both the AR-GSR-positive and -negative groups, this subset analysis extracts a patient population with poor survival characteristics reflecting those that we had originally observed in ctDNA aneuploidy-high Group 1 and ctDNA aneuploidy-low Group 2. This is expected because all samples with likely somatic pathogenic mutations are defined as ctDNA-positive in our study. Accordingly, in univariate analysis of this

subset of 183 patients that have poor survival characteristics, the patients that were AR-GSR-positive did not have worse OS or rPFS when compared with the patients that were AR-GSR-negative. Because the univariate test with this subgroup was negative, there is no requirement for multivariable correction using the VAFs from likely somatic pathogenic mutations as a “true” measure of ctDNA fraction.

Reviewer Table 1. Hazard Ratios for Associations of AR-GSR Status with OS and rPFS in the Subset of 183 Patients Harboring Likely Somatic Pathogenic Mutations

Univariate analysis n=183	# of events OS n=154	HR OS (95% CI)	Median OS (95% CI)	# of events rPFS n=159	HR rPFS (95% CI)	Median rPFS (95%CI)
AR-GSR subset No n=131	107	Reference	19.1 (15.4,24.7)	110	Reference	14.9 (11.4,16.9)
AR-GSR subset Yes n=52	47	1.1 (0.8,1.5)	17.9 (16.7,24.6)	49	1.3 (0.9,1.9)	10.6 (8.64,17.12)
LBD-truncating subset No n=178	149	Reference	18.5 (16.8,21.7)	154	Reference	13.7 (11.1,16.7)
LBD-truncating subset Yes n=5	5	1.9 (0.8,4.7)	16.4 (8.8,NA)	5	2.3 (0.9,5.6)	8.8 (3.7,NA)
AR ecDNA subset No n=158	131	Reference	18.6 (17.0,21.9)	135	Reference	15.1 (11.6,17.3)
AR ecDNA subset Yes n=25	23	1.4 (0.9,2.2)	14.8 (10.3,26.6)	24	1.8 (1.1,2.7)	8.64 (5.7,14.6)

The reviewer also raised concern about the lack of equivalency between ichorCNA ctDNA aneuploidy fraction and the “true” ctDNA fraction measured by the maximum VAF method affecting the prognostic groups shown in Figure 6. The reviewer is correct that the ctDNA aneuploidy fraction provided by ichorCNA was important for defining ctDNA aneuploidy-high Group 1 (i.e. above the mean + 3SD cutoff), but the ctDNA aneuploidy-low Group 2 and the ctDNA negative Group 3 were derived using measures that were independent of ichorCNA ctDNA aneuploidy fraction. Nevertheless, as required by the reviewer, we also used this subset of 183 patients harboring likely somatic pathogenic mutations in ctDNA aneuploidy-high Group 1 and ctDNA aneuploidy-low Group 2 to perform a *post-hoc* analysis of whether they had a higher risk for progression or death compared with a subset of the patients from ctDNA-negative Group 3. The subset of patients in ctDNA-negative Group 3 that we selected for this comparison were those with plasma cfDNA concentrations above the mean value of the mutation-positive group (above the mean of 3.96 log₂ ng cfDNA per mL of plasma as illustrated in new Supplementary Figure S2). As indicted by the reviewer, this may represent a “true negative” subgroup because there were no genomic alterations identified despite having a reasonably high amount of cfDNA in which to detect them). As shown in **Reviewer Table 2**, this subset analysis revealed the same patterns we reported in Figure 6, where ctDNA-positive patients had worse outcomes, especially those in ctDNA aneuploidy-high Group 1.

Reviewer Table 2. Hazard Ratios for Associations of ctDNA Classification with OS and rPFS in the Subset of 183 ctDNA-Positive Patients Harboring Likely Somatic Pathogenic Mutations and the Subset of 43 ctDNA-Negative Patients with log₂ cfDNA Concentration > 3.96

Subgroup analysis n=226	# of events OS n=185	HR OS (95% CI)	Median OS (95% CI)	# of events rPFS n=191	HR rPFS (95% CI)	Median rPFS (95%CI)
Group 3 subset n=43	31	Reference	35.8 (27.7, 50.6)	32	Reference	25.9 (16.8, 39.7)
Group 2 subset n=94	75	1.4 (0.9, 2.1)	23.6 (19.3, 33.5)	80	1.5 (1.0, 2.3)	16.1 (13.6, 20.9)
Group 1 subset n=89	79	2.4 (1.5, 3.6)	16.3 (14.6, 18.0)	79	2.2 (1.5, 3.4)	10.9 (8.1, 15.1)
ctDNA-Neg subset n=43	31	Reference	35.8 (27.7, 50.6)	32	Reference	25.9 (16.8, 39.7)
ctDNA-Pos subset n=183	154	1.7 (1.2, 2.6)	18.4 (16.7, 21.5)	159	1.8 (1.2, 2.7)	13.6 (11.0, 16.4)

General comments:

Fraction ctDNA+:

In figure 2A the authors detect ctDNA by aneuploidy in 200 samples, 97 by somatic alterations whereas 479 are negative. Overall, the ctDNA+ rate is therefore 297/(297+479) = 38% (not including amplification-only detection which was used to expand group 2). This is very low. Research groups cfDNA/gDNA profiling have ~60% cfDNA+ rate in comparable cohorts (pmid 29367197, 30458854). In addition, the data presented in the manuscript likely has false positive ctDNA-cases due to the detection of CHIP (clonal hematopoiesis of indeterminate potential) variants, coming from clonal expansions in the white blood cells as gDNA sequencing was not done in parallel. Supplementary Figure S5 the authors show that TP53 is the most common mutation to classify samples as ctDNA+ which very commonly is also occurring in CHIP. This also affects other genes such as ATM which was also applied to classify a case as ctDNA+. The authors should mention this as a limitation in the discussion.

Response: We agree that the ctDNA+ rate excluding detection of *AR/MYC/MYCN* alterations/gain (297/776 or 38%) is low relative to the cited studies. In our previous rebuttal, we emphasized that the study published by Drs. Annala, Chi, Wyatt, and colleagues (PMID 29367197) at the Vancouver Prostate Centre has very high sensitivity for detecting ctDNA and calculating ctDNA fraction by leveraging variant allele fractions of tumor-derived somatic mutations. As reported in the methods sections of their publication, the input requirement for this “Vancouver assay” is 10-100 ng cfDNA (or 3000-30,000 haploid genome equivalents) for library preparation prior to targeted DNA-seq. This Vancouver assay also has a companion whole blood germline assay to enhance sensitivity for disambiguating mutations in ctDNA from germline mutations or clonal hematopoiesis (CHIP). Similarly, the study published by Mayrhofer, Lindberg, and colleagues (PMID 30458854) performed 3 separate assays: 1) low-pass whole-genome cfDNA-seq; 2) targeted capture cfDNA-seq; 3) targeted capture DNA-seq of whole blood germline DNA, which (like the Vancouver assay) enhances sensitivity for disambiguating mutations in ctDNA from germline mutations or clonal hematopoiesis (CHIP). We would like to re-emphasize that our study was only approved for analysis of plasma DNA, which meant we did not have a germline DNA control for enhancing sensitivity of mutation detection. **This was mentioned as a limitation in the Discussion of the previous versions of the manuscript,**

and remains unchanged in the Discussion of the current version of the manuscript:

Lines 440-443: “A final limitation was that targeted DNA-seq was performed on cfDNA without patient-matched germline controls, blood, or tumor tissue. This limits the ability to accurately distinguish between somatic or germline alterations in target genes, or between tumor-associated mutations and clonal hematopoiesis. To address this, we only considered high confidence mutations with well-annotated pathogenicity (from OncoKB⁵⁷). This may have resulted in under-reporting of mutations in some genes targeted by the AR-ctDETECT assay and also restricts our ability to determine tumor mutational burden.”

The authors write:

“Therefore, we classified 60 cfDNA samples with gain of the AR gene body and 2 or more AR-GSRs as likely harboring AR ecDNA (Figures 2D&F, Supplementary Data S5).”

Is this anchored in any data in which the authors have correlated ecDNA with the AR gain level and number of GSRs? Otherwise, the authors should change the definition of the cases with "with gain of the AR gene body and 2 or more AR-GSRs" to "AR alteration high" or something similar.

Response: Yes, the statement is supported by a study we published last year (reference 40). That publication investigated the underlying mechanism for the consistent finding across CRPC biopsy, CRPC autopsy, and CRPC ctDNA studies that a significant number of CRPC patients displayed AR copy number gain concurrent with accumulation of multiple AR-GSRs. In that study, we discovered that amplified AR gene structures were multiply-rearranged in many samples, due to AR residing on ecDNA. Thus, a signature of AR amplification on ecDNA is AR gain concurrent with multiple (2+) AR-GSRs. That study was properly referenced in the sentence preceding the one highlighted by the reviewer:

Lines 190-194: “Additionally, AR being amplified on extrachromosomal circular DNA (ecDNA) was recently identified as a mechanism that promotes high AR copy number and accumulation of complex AR-GSRs in individual CRPC cells⁴⁰. Therefore, we classified 60 cfDNA samples with gain of the AR gene body and 2 or more AR-GSRs as likely harboring AR ecDNA (Figures 2D&F, Supplementary Data S5).”

Figure 4, g-h:

The authors report copy loss of TP53, PTEN and RB1, but are these variants really relevant? If one copy remain the gene is still functional, homozygous loss is required for inactivation or one-copy loss with a mutation of the other allele. E.g. RB1 is very close to BRCA2, often a large segment is detected that cover both genes are but RB1 is still functional if BRCA is homozygously lost and is therefore an irrelevant passenger event. The authors need to motivate the presented data and reason if it is really relevant variants that are displayed.

Response: We agree with the reviewer about the two-hit definition of functional tumor suppressor gene loss. In cfDNA sequencing data, it is challenging to accurately distinguish between 1- or 2-copy losses. Therefore, **we edited the Results** to better reflect this fact:

Lines 271-273: “In addition to mutations, substantial copy number loss was detectable for TP53, PTEN, and RB1 in ctDNA-positive samples, although we were

not able to distinguish whether these were 1- or 2-copy losses (Figures 4F-H, Supplementary Data S4)."

We think that keeping the tumor suppressor gene copy loss information is important in light of Dr. Halabi's ASCO Abstract #5007, which was presented orally on June 1, 2024. The title of the abstract is "*A clinical-genetic (CG) circulating tumor DNA (ctDNA)-based prognostic model for predicting overall survival (OS) in men with metastatic castrate-resistant prostate cancer (mCRPC) treated with potent androgen receptor inhibition (Alliance)*". In this work, Dr. Halabi used the AR-ctDETECT data we generated in the current manuscript to identify genetic features based on feature importance using a random survival forest. She used these features to develop a clinical-genetic model that improved upon the validated clinical prognostic model of OS she developed (and validated) previously in mCRPC men that included the variables: performance status, disease site, opioid analgesic use, lactate dehydrogenase, albumin, hemoglobin, prostate specific antigen, and alkaline phosphatase. Important genetic features extracted from AR-ctDETECT data included losses and/or mutations in many genes, including *PTEN*, *TP53*, and *RB1*.

This work has been described in a manuscript that was reviewed at *Annals of Oncology*, and a revision has been invited. This revised manuscript entitled "*Clinical-Genetic Circulating Tumor DNA Prognostic Model for Overall Survival in Metastatic Castrate-Resistant Prostate Cancer*" is attached for the Reviewers to evaluate.

Response to Reviewer #4:

The authors have responded adequately to my comments. They have also adequately responded to the other reviewer comments, in my opinion.

Response: We thank the reviewer for their time in evaluating our manuscript and guiding improvements during revision.

Reviewers' comments:

Reviewer #2:

The Authors have responded to my observation adequately by submitting the embargoed ASCO abs.

My concerns have been answered .

Reviewer #3:

This reviewer thank the authors again for addressing the concerns raised. However, the replies strengthen my concerns with regards to the limitations of the biobanked material and how it causes low coverage which in turn leads to poor sensitivity to detect somatic alterations and prevent robust estimation of ctDNA fraction. This will in turn affect the all aspects of the manuscript where ctDNA fraction is applied in various ways for outcomes analysis.

The issue is highlighted by the low ctDNA+ rate (38%) which I commented on previously and which was acknowledged by the authors. Assuming a similar positivity rate as others have reported, $\geq 22\%$ are missclassified as ctDNA- whilst in reality have a ctDNA fraction $\geq 2\%$.

Additionally, I pointed out previously that the ctDNA-fraction estimation concordance between "ctDNA aneuploidy fraction" and "maximum VAF" appeared to be driven by outliers. The authors replied that this is not the case and provided "Reviewer Figure 1". However, in the the subplots " ≤ 0.25 " and " ≤ 0.125 " with "all genes" the correlation is very low especially as it approaches the LOD of "ctDNA aneuploidy fraction".

My concern is that the noise will lead to samples with e.g. high/low/undetectable ctDNA are wrongly classified which will affect the PFS/OS analysis for GSRs which to me is the main novel aspect of the manuscript.

Therefore I requested an analysis using only the cases ctDNA+ by maximum VAF. As this analysis did not show any significant association between AR-GSRs and outcome it is unclear at this state, if the significant results using the entire cohort is due to noise or true signal. Unfortunately, and due to

the limitations of the biobanked material, I do not think that the authors can address this with the available data.

The editors need to decide if the current work is acceptable for Nature Communications despite the discussed limitations.

We thank the Reviewers for their re-evaluation of our revised manuscript. Our response to each reviewer comment is indicated in blue font.

Response to Reviewer #2:

The Authors have responded to my observation adequately by submitting the embargoed ASCO abs.

My concerns have been answered.

Response: We thank the reviewer for their time in re-evaluating our manuscript and guiding improvements during revision.

Response to Reviewer #3:

This reviewer thank the authors again for addressing the concerns raised. However, the replies strengthen my concerns with regards to the limitations of the biobanked material and how it causes low coverage which in turn leads to poor sensitivity to detect somatic alterations and prevent robust estimation of ctDNA fraction. This will in turn affect the all aspects of the manuscript where ctDNA fraction is applied in various ways for outcomes analysis.

The issue is highlighted by the low ctDNA+ rate (38%) which I commented on previously and which was acknowledged by the authors. Assuming a similar positivity rate as others have reported, $\geq 22\%$ are missclassified as ctDNA- whilst in reality have a ctDNA fraction $\geq 2\%$.

Additionally, I pointed out previously that the ctDNA-fraction estimation concordance between “ctDNA aneuploidy fraction” and “maximum VAF” appeared to be driven by outliers. The authors replied that this is not the case and provided “Reviewer Figure 1”. However, in the the subplots “ ≤ 0.25 ” and “ ≤ 0.125 ” with “all genes” the correlation is very low especially as it approaches the LOD of “ctDNA aneuploidy fraction”.

My concern is that the noise will lead to samples with e.g. high/low/undetectable ctDNA are wrongly classified which will affect the PFS/OS analysis for GSRs which to me is the main novel aspect of the manuscript.

Therefore I requested an analysis using only the cases ctDNA+ by maximum VAF. As this analysis did not show any significant association between AR-GSRs and outcome it is unclear at this state, if the significant results using the entire cohort is due to noise or true signal. Unfortunately, and due to the limitations of the biobanked material, I do not think that the authors can address this with the available data.

The editors need to decide if the current work is acceptable for Nature Communications despite the discussed limitations.

Response: We thank Reviewer 3 for their time in evaluating our revised manuscript and responses to their critiques. We recognize that the reviewer has outstanding concerns with the signal vs. noise of the ctDNA aneuploidy fraction measurement, and understand they are concerned this issue could impact “all aspects of the manuscript where ctDNA aneuploidy fraction is applied in various ways for outcomes analysis” (namely: 1) rate of detection of ctDNA-

positive samples and 2) the strength of multivariable corrections in tests of associations between AR-GSRs and rPFS/OS).

We strongly disagree with the statement made by Reviewer 3 that “it is unclear at this state, if the significant results using the entire [sic] cohort is due to noise or true signal. Unfortunately, and due to the limitations of the biobanked material, I do not think that the authors can address this with the available data”. To demonstrate this, we **added the hazard ratios for the test of association between ctDNA aneuploidy fraction and clinical outcomes to the Results section under the “Baseline cfDNA correlates with ctDNA aneuploidy fraction” heading:**

Lines 143-146: “A test of the association of ctDNA aneuploidy fraction with clinical outcomes revealed that the hazard ratio (HR) for rPFS was 1.3 (95% CI: 1.2-1.4, $P < 0.0001$) and the HR for OS was 1.5 (95% CI: 1.4-1.6, $P < 0.0001$) for each 0.1 unit (10%) increase in ctDNA aneuploidy fraction.”

The fact that each 0.1 unit (10%) increase in ctDNA aneuploidy fraction is associated with increased risk of radiographic progression and death demonstrates that the ctDNA aneuploidy fraction provides a signal (not noise) that is associated with clinical outcomes. This further demonstrates that ctDNA aneuploidy fraction serves as a valid variable for multivariable correction in tests of associations between other genetic features (such as AR-GSRs) and rPFS/OS in our study.

The statement made by Reviewer 3 that our study found a “low ctDNA+ rate (38%)” is false. As reported everywhere in our manuscript (including the abstract), the ctDNA+ rate in our study was 59%, with 26% having high ctDNA aneuploidy and 33% having undetectable ctDNA aneuploidy but displaying AR gain or structural rearrangement, MYC/MYCN gain, or a pathogenic mutation. Because this incorrect ctDNA+ rate (38%) was used by Reviewer 3 to estimate a false negative rate of 22% (60% minus 38%), the statement made by Reviewer 3 that “Assuming a similar positivity [sic] rate as others have reported, $\geq 22\%$ are misclassified [sic] as ctDNA- whilst in reality have a ctDNA fraction $\geq 2\%$.”, is also false. The key point of our paper is that we used the data in Figures 2 and 3 to build an argument that additional signals detected by the AR-ctDETECT assay, namely alterations in AR (including AR-GSRs), MYC, and MYCN, can be leveraged to detect these remaining ctDNA-positive cases, resulting in a ctDNA positive rate of 59%. This ctDNA-positive rate is almost identical to the 60% positivity rate in other studies noted by the Reviewer. The original submission of our manuscript, and all revisions since, have dedicated a paragraph of the Discussion (lines 380-399) to making these comparisons.

We appreciate that Reviewer 3 prefers detection of tumor-derived somatic mutations in cfDNA as a method for calculating ctDNA fraction. As we have articulated repeatedly, high plasma volumes and cfDNA yields along with germline DNA measurements are necessary to detect tumor-derived somatic mutations in ctDNA with high sensitivity. The importance of our work is that it demonstrates (and clinically validates) an alternative way to detect ctDNA in plasma cfDNA from mCRPC patients, which could be applied to many other retrospective studies with low volumes of banked plasma. As we emphasized in our previous revision (lines 469-471 of the Discussion), “Although this ctDNA aneuploidy fraction correlated with other inputs used to calculate ctDNA fraction, it is important to note that these metrics are distinct and should not be directly compared.” It will be interesting in future studies to perform a comparative analysis of the AR ctDETECT assay with other assay(s) that require high input plasma/cfDNA along with a companion germline measurement to detect tumor-derived somatic mutations for calculating ctDNA fraction. However, that would require a separate and appropriately powered patient

cohort for which higher amounts of plasma are available. This is clearly outside of the scope of the current study.

We also appreciated the suggestion provided by Reviewer 3 in their previous review to perform a subset analysis restricted to the 183 samples in which likely somatic pathogenic mutations were detected, because this represents a subset in which we could use the maximum VAF method preferred by Reviewer 3 to calculate ctDNA fraction for multivariable correction. As we emphasized in the previous rebuttal letter (and supported by the data in Reviewer Table 1 that we had provided in that previous rebuttal letter), this subset analysis idea would not work for the envisioned multivariable correction because it erased any signal from AR-GSRs in univariate analysis (the univariate analysis was never raised by Reviewer 3 as an issue; it was the multivariable correction method using ctDNA aneuploidy fraction with which they had concerns). Inappropriately, the reviewer used the new results we had presented to incorrectly dismiss the original association found between AR-GSRs with OS and rPFS in the full cohort (Figs. 2g and 2h) as “noise”.

REVIEWER COMMENTS

Reviewer #2:

I carefully reviewed the authors further reply (the third one) and believe that they have satisfactorily responded to this last round of revision.

Reviewer #3:

First, I thank the authors for addressing my comments and I apologize if the last review was a bit blunt. The manuscript is quite complicated to read due to the limitation in the biobanked material, therefore I was a bit stressed due to other pressing tasks. In my comments I will refer to:

200 cases with aneuploidy high.

97 cases with a detected pathogenic mutation.

159 cases with detected amplification in AR/MYC/MYCN.

320 completely negative cases.

With regards to my skepticism to the use of the "ctDNA aneuploidy fraction estimate".

Based on data from healthy donors the "ctDNA aneuploidy fraction estimate" noise limit of detection is a ctDNA fraction of 0.142. Therefore, cases with "ctDNA aneuploidy fraction" below this number should be considered to be below the limit of detection which equals noise. It is equivalent to detecting a point mutation with a VAF of 0.00001, in this scenario it might as well be present due to sequencing errors/mapping errors etc.

Furthermore, the authors argue that the "ctDNA aneuploidy fraction estimate" may be used in the manuscript as they in line 143-146 test the association of "ctDNA aneuploidy fraction estimate" with clinical outcomes. In my previous comments I was not suggesting that the "ctDNA aneuploidy fraction estimate" estimate is just noise, the cases with signal above the LOD ctDNA fraction of 0.142 for sure have true signal (but with discrepancy vs maximum VAF as discussed in previous communications). However, to motivate use of the "ctDNA aneuploidy fraction estimate" for samples with signal below the limit of detection (<0.142), the authors should test the association of

"ctDNA aneuploidy fraction estimate" with clinical outcomes by excluding the 200 aneuploidy high cases and the 97 cases with a detected mutation in a second step.

With regards to the ctDNA+ rate of 38%. This is based on the fraction of patients in this cohort where it is possible for the authors to assess the ctDNA-fraction. Either through aneuploidy analysis or through detected mutations. Being able to assess the ctDNA fraction accurately is of course critical if ctDNA fraction estimation are to be used for any outcome analysis.

The authors claim that additionally 159 cases should be added to the ctDNA+ cohort based on detection of amplifications of AR/MYC/MYCN. However, these cases have "ctDNA aneuploidy fraction" below the noise level of the healthy donors, which means that the ctDNA fraction estimates are unknown. Likely they have higher ctDNA fractions than the completely negative cases which is demonstrated by rPFS and OS analysis compared to the completely negative cases. However, I am not convinced by the motivation to create a joint group with the 97 cases with a detected pathogenic mutation. If keeping this group at all then I would prefer to keep them as a separate group in the plots investigating associations with outcomes.

As for the associations of AR-GSRs and rPFS/OS. As I understand it, here the authors again apply the "ctDNA aneuploidy fraction estimate" using all samples despite that only 200/776 have a signal higher than the noise level of the healthy donors. I do of course believe that AR-GSRs are a resistance mechanism to ARPIs and associated with poor response, however, I am uncertain based on how the analysis is done if it is demonstrated in a fair way.

Reviewer #4:

The authors have further improved the manuscript in their revisions.

However, there appears to be a lingering misunderstanding of the ctDNA positivity rate, as evidenced by the exchange in comments between one reviewer and the authors. Upon re-reading the manuscript, this reviewer appreciates that the authors calculated a 59% ctDNA positivity rate (26% high ctDNA aneuploidy + 33% pathogenic mutations or AR/MYC/MYCN alteration). This additional use of alterations in AR/MYC/MYCN to call the detection of ctDNA is a key practical innovation of this work, as it can be applied to cases with low plasma input, such as banked samples from prior clinical trials. This 59% positivity is comparable to the ~60% ctDNA positivity previously reported in similar cohorts using higher volumes of plasma input (e.g. PMID 29367197, PMID 34083234).

Nevertheless, this reviewer also appreciates how one could come to the conclusion that the ctDNA positivity is 38%, based on data presented in Figs 2A and 3A using somatic pathogenic mutations plus ichorCNA ctDNA aneuploidy to calculate ctDNA positivity $[(200+97)/(200+97+479)]$. The concept of including alterations in AR/MYC/MYCN to call additional ctDNA positivity is not introduced until Fig 4B-C, rather late in the manuscript. Although this reviewer appreciates that the authors presented the story in a gradual, step-wise manner to be rigorous, the point may be lost on some readers. To address this source of potential confusion for a broader audience, I recommend the authors be more upfront in providing a detailed explanation of their overall Group 1 vs 2 vs 3 classification scheme much earlier in the text, perhaps at the end of the Introduction section.

We thank the Reviewers for their re-evaluation of our revised manuscript. Our response to each reviewer comment is indicated in blue font.

Response to Reviewer #2:

I carefully reviewed the authors further reply (the third one) and believe that they have satisfactorily responded to this last round of revision.

Response: We thank the reviewer for their time in re-evaluating our manuscript and responses to Reviewer #3.

Response to Reviewer #3:

First, I thank the authors for addressing my comments and I apologize if the last review was a bit blunt. The manuscript is quite complicated to read due to the limitation in the biobanked material, therefore I was a bit stressed due to other pressing tasks. In my comments I will refer to:

200 cases with aneuploidy high.

97 cases with a detected pathogenic mutation.

159 cases with detected amplification in AR/MYC/MYCN.

320 completely negative cases.

With regards to my skepticism to the use of the "ctDNA aneuploidy fraction estimate".

Based on data from healthy donors the "ctDNA aneuploidy fraction estimate" noise limit of detection is a ctDNA fraction of 0.142. Therefore, cases with "ctDNA aneuploidy fraction" below this number should be considered to be below the limit of detection which equals noise. It is equivalent to detecting a point mutation with a VAF of 0.00001, in this scenario it might as well be present due to sequencing errors/mapping errors etc.

Furthermore, the authors argue that the "ctDNA aneuploidy fraction estimate" may be used in the manuscript as they in line 143-146 test the association of "ctDNA aneuploidy fraction estimate" with clinical outcomes. In my previous comments I was not suggesting that the "ctDNA aneuploidy fraction estimate" estimate is just noise, the cases with signal above the LOD ctDNA fraction of 0.142 for sure have true signal (but with discrepancy vs maximum VAF as discussed in previous communications). However, to motivate use of the "ctDNA aneuploidy fraction estimate" for samples with signal below the limit of detection (<0.142), the authors should test the association of "ctDNA aneuploidy fraction estimate" with clinical outcomes by excluding the 200 aneuploidy high cases and the 97 cases with a detected mutation in a second step.

With regards to the ctDNA+ rate of 38%. This is based on the fraction of patients in this cohort where it is possible for the authors to assess the ctDNA-fraction. Either through aneuploidy analysis or through detected mutations. Being able to assess the ctDNA fraction accurately is of course critical if ctDNA fraction estimation are to be used for any outcome analysis.

The authors claim that additionally 159 cases should be added to the ctDNA+ cohort based on detection of amplifications of AR/MYC/MYCN. However, these cases have "ctDNA aneuploidy fraction" below the noise level of the healthy donors, which means that the ctDNA fraction

estimates are unknown. Likely they have higher ctDNA fractions than the completely negative cases which is demonstrated by rPFS and OS analysis compared to the completely negative cases. However, I am not convinced by the motivation to create a joint group with the 97 cases with a detected pathogenic mutation. If keeping this group at all then I would prefer to keep them as a separate group in the plots investigating associations with outcomes.

As for the associations of AR-GSRs and rPFS/OS. As I understand it, here the authors again apply the "ctDNA aneuploidy fraction estimate" using all samples despite that only 200/776 have a signal higher than the noise level of the healthy donors. I do of course believe that AR-GSRs are a resistance mechanism to ARPIs and associated with poor response, however, I am uncertain based on how the analysis is done if it is demonstrated in a fair way.

Response: We have performed the requested subgroup analyses, the results of which continue to support the hypotheses and conclusions that were presented in earlier versions of the paper.

As requested by Reviewer 3, we tested the association of ctDNA aneuploidy fraction with clinical outcomes in the subgroup of 479 patients with low ctDNA aneuploidy and lacking a detectable pathogenic mutation. In this sub-group, ctDNA aneuploidy fraction remained prognostic for rPFS and OS. This new finding (which further substantiates the accuracy of our conclusions), and the rationale for the subset analysis, **were added to the Results section of the manuscript under the “AR-GSRs in baseline cfDNA associates with poor clinical outcomes” heading.**

Lines 212-219: “We noted that many of the samples with AR-GSRs were below the ctDNA aneuploidy fraction cutoff, which could impact the accuracy of using ctDNA aneuploidy fraction for multivariable correction. Therefore, we tested the association of ctDNA aneuploidy fraction as a continuous variable with clinical outcomes in the 479 patients with low ctDNA aneuploidy and lacking a detectable pathogenic mutation. Remarkably, even in this subgroup, ctDNA aneuploidy fraction remained prognostic for rPFS (HR = 1.05 for each 1% increase in ctDNA aneuploidy fraction, 95% CI: 1.01-1.09) and OS (HR = 1.05 for each 1% increase in ctDNA aneuploidy fraction, 95% CI: 1.01-1.09).”

These findings further demonstrate that the multivariable correction for ctDNA aneuploidy fraction is appropriate and that we are testing the association between AR-GSRs and clinical outcomes in a fair way. To reflect these new results, **we made small edits to the Abstract and Discussion** as follows:

Lines 46-47: “undetectable ctDNA aneuploidy” was changed to “low ctDNA aneuploidy”:

Lines 488-491: “Although this ctDNA aneuploidy fraction calculation was prognostic for rPFS and OS, and correlated with other inputs used to calculate ctDNA fraction, it is important to note that these metrics are distinct and should not be directly compared.”

As requested by Reviewer 3, we also separated the 256 samples in Group 2 into those harboring a detectable pathogenic mutation (n = 97) and those lacking a detectable pathogenic mutation but harboring a detectable AR-GSR or gain in AR, MYC, or MYCN (n = 159) (**new Supplementary Figure S8**). The 97 samples harboring a detectable pathogenic mutation had clinical outcomes (rPFS and OS) that could not be distinguished from ctDNA-positive Group 1.

The 159 samples harboring a detectable AR-GSR or gain in AR, MYC, or MYCN had better OS than ctDNA-positive Group 1, but rPFS that could not be distinguished from ctDNA-positive Group 1. Both of these subgroups had worse clinical outcomes (rPFS and OS) compared to ctDNA-negative Group 3. **We edited the Results to reflect these new findings:**

Lines 361-368: "When the patients in ctDNA aneuploidy-low Group 2 were analyzed as two subgroups, the 97 patients in Group 2 that harbored detectable pathogenic mutations had worse rPFS and OS than patients in ctDNA-negative Group 3 (Supplementary Figures S8A-D), but rPFS and OS that were indistinguishable from ctDNA aneuploidy-high Group 1 (Supplementary Figures S8E&F). The 159 patients in Group 2 that lacked detectable pathogenic mutations but harbored AR-GSRs and/or a copy gain in AR, MYC, or MYCN had worse rPFS and OS than patients in ctDNA-negative Group 3 (Supplementary Figures S8A-D), and better rPFS (but not OS) than patients in ctDNA aneuploidy-high Group 1 (Supplementary Figures S8E&F)."

Response to Reviewer #4:

The authors have further improved the manuscript in their revisions.

However, there appears to be a lingering misunderstanding of the ctDNA positivity rate, as evidenced by the exchange in comments between one reviewer and the authors. Upon re-reading the manuscript, this reviewer appreciates that the authors calculated a 59% ctDNA positivity rate (26% high ctDNA aneuploidy + 33% pathogenic mutations or AR/MYC/MYCN alteration). This additional use of alterations in AR/MYC/MYCN to call the detection of ctDNA is a key practical innovation of this work, as it can be applied to cases with low plasma input, such as banked samples from prior clinical trials. This 59% positivity is comparable to the ~60% ctDNA positivity previously reported in similar cohorts using higher volumes of plasma input (e.g. PMID 29367197, PMID 34083234).

Nevertheless, this reviewer also appreciates how one could come to the conclusion that the ctDNA positivity is 38%, based on data presented in Figs 2A and 3A using somatic pathogenic mutations plus ichorCNA ctDNA aneuploidy to calculate ctDNA positivity $[(200+97)/(200+97+479)]$. The concept of including alterations in AR/MYC/MYCN to call additional ctDNA positivity is not introduced until Fig 4B-C, rather late in the manuscript. Although this reviewer appreciates that the authors presented the story in a gradual, step-wise manner to be rigorous, the point may be lost on some readers. To address this source of potential confusion for a broader audience, I recommend the authors be more upfront in providing a detailed explanation of their overall Group 1 vs 2 vs 3 classification scheme much earlier in the text, perhaps at the end of the Introduction section.

Response: We thank the reviewer for their time in re-evaluating our manuscript and responses to Reviewer #3. We appreciate their suggestion to enhance the clarity of our study, and have **edited the last paragraph of the Introduction:**

Lines 98-108: "In this work, we detect mCRPC-specific genomic alterations at high frequency in cfDNA specimens classified as having high ctDNA aneuploidy. We assign these patients to a ctDNA aneuploidy-high Group 1 and demonstrate they have a short duration of radiographic progression-free survival (rPFS) and

overall survival (OS) compared with patients assigned to a ctDNA-negative Group 3. In patients with low ctDNA aneuploidy, we identify a second ctDNA-positive group on the basis of detectable pathogenic tumor-derived mutations, AR-GSRs, and/or copy gains in AR, MYC or MYCN. We assign these patients to a ctDNA aneuploidy-low Group 2, and demonstrate they also have shorter rPFS and OS compared with patients in ctDNA-negative Group 3. These data validate the importance of evaluating these genomic alterations in patients having low ctDNA aneuploidy in a phase 3 trial context, as well as the prognostic utility of detecting ctDNA in mCRPC patients being treated with contemporary AR-targeted therapies.”

REVIEWERS' COMMENTS

Reviewer #4 (Remarks to the Author):

In my opinion the authors have satisfactorily responded to all reviewer comments. The additional subgroup analysis associating ctDNA aneuploidy fraction with outcomes in the low ctDNA aneuploidy subgroup provides additional evidence supporting their conclusions.